# Human de novo mutation rates from a four-generation pedigree reference

David Porubsky[1], Harriet Dashnow[2,3,22], Thomas A. Sasani[2,22], Glennis A. Logsdon[1,20,22], Pille Hallast[4,22], Michelle D. Noyes[1,22], Zev N. Kronenberg[5,22], Tom Mokveld[5,22], Nidhi Koundinya[1], Cillian Nolan[5], Cody J. Steely[2,6], Andrea Guarracino[7], Egor Dolzhenko[5], William T. Harvey[1], William J. Rowell[5], Kirill Grigorev[8,9], Thomas J. Nicholas[2], Michael E. Goldberg[2], Keisuke K. Oshima[10], Jiadong Lin[1], Peter Ebert[11,12], W. Scott Watkins[2], Tiffany Y. Leung[13], Vincent C. T. Hanlon[13], Sean McGee[1], Brent S. Pedersen[2], Hannah C. Happ[2], Hyeonsoo Jeong[1,21], Katherine M. Munson[1], Kendra Hoekzema[1], Daniel D. Chan[13], Yanni Wang[13], Jordan Knuth[1], Gage H. Garcia[1], Cairbre Fanslow[5], Christine Lambert[5], Charles Lee[4], Joshua D. Smith[1], Shawn Levy[14], Christopher E. Mason[15,16,17], Erik Garrison[7], Peter M. Lansdorp[13,18], Deborah W. Neklason[2], Lynn B. Jorde[2], Aaron R. Quinlan[2], Michael A. Eberle[5] & Evan E. Eichler[1,19✉]

Understanding the human de novo mutation (DNM) rate requires complete sequence information[1]. Here using five complementary short-read and long-read sequencing technologies, we phased and assembled more than 95% of each diploid human genome in a four-generation, twenty-eight-member family (CEPH 1463). We estimate 98–206 DNMs per transmission, including 74.5 de novo single-nucleotide variants, 7.4 non-tandem repeat indels, 65.3 de novo indels or structural variants originating from tandem repeats, and 4.4 centromeric DNMs. Among male individuals, we find 12.4 de novo Y chromosome events per generation. Short tandem repeats and variable-number tandem repeats are the most mutable, with 32 loci exhibiting recurrent mutation through the generations. We accurately assemble 288 centromeres and six Y chromosomes across the generations and demonstrate that the DNM rate varies by an order of magnitude depending on repeat content, length and sequence identity. We show a strong paternal bias (75–81%) for all forms of germline DNM, yet we estimate that 16% of de novo single-nucleotide variants are postzygotic in origin with no paternal bias, including early germline mosaic mutations. We place all this variation in the context of a high-resolution recombination map (~3.4 kb breakpoint resolution) and find no correlation between meiotic crossover and de novo structural variants. These near-telomere-to-telomere familial genomes provide a truth set to understand the most fundamental processes underlying human genetic variation.

The telomere-to-telomere (T2T) assembly of a human genome[1] added an estimated 8% of the most repeat-rich DNA, including regions typically excluded from studies of human genetic variation, such as centromeres[2], segmental duplications (SDs)[3] and acrocentric regions[1,4]. Long-read sequencing (LRS) of many phased human diploid genomes has already begun to offer insights into mutational mechanisms[5–7], opening up the discovery of all forms of variation irrespective of class or complexity[8,9]. Direct comparison of parental genomes to their offspring increases the power to identify DNMs as opposed to mapping reads to an intermediate reference, such as GRCh38 or T2T-CHM13 (ref. 10).

The goal of this study was to construct a high-quality human pedigree resource whereby chromosomes were fully assembled and phased, and their transmission was studied intergenerationally to enhance our understanding of both recombination and DNM processes. We sought to eliminate three ascertainment biases with respect to discovery, including biases to specific genomic regions, classes of genetic

[1]Department of Genome Sciences, University of Washington School of Medicine, Seattle, WA, USA. [2]Department of Human Genetics, University of Utah, Salt Lake City, UT, USA. [3]Department of Biomedical Informatics, University of Colorado Anschutz Medical Campus, Aurora, CO, USA. [4]The Jackson Laboratory for Genomic Medicine, Farmington, CT, USA. [5]PacBio, Menlo Park, CA, USA. [6]Department of Internal Medicine, University of Kentucky College of Medicine, Lexington, KY, USA. [7]Genetics, Genomics and Informatics, University of Tennessee Health Science Center, Memphis, TN, USA. [8]Space Biosciences Research Branch, NASA Ames Research Center, Moffett Field, CA, USA. [9]Blue Marble Space Institute of Science, Seattle, WA, USA. [10]Department of Genetics, Epigenetics Institute, Perelman School of Medicine, University of Pennsylvania, Philadelphia, PA, USA. [11]Core Unit Bioinformatics, Medical Faculty and University Hospital Düsseldorf, Heinrich Heine University, Düsseldorf, Germany. [12]Center for Digital Medicine, Heinrich Heine University, Düsseldorf, Germany. [13]Terry Fox Laboratory, BC Cancer Agency, Vancouver, British Columbia, Canada. [14]Element Biosciences, San Diego, CA, USA. [15]Department of Physiology and Biophysics, Weill Cornell Medicine, New York, NY, USA. [16]The HRH Prince Alwaleed Bin Talal Bin Abdulaziz Alsaud Institute for Computational Biomedicine, Weill Cornell Medicine, New York, NY, USA. [17]The WorldQuant Initiative for Quantitative Prediction, Weill Cornell Medicine, New York, NY, USA. [18]Department of Medical Genetics, University of British Columbia, Vancouver, British Columbia, Canada. [19]Howard Hughes Medical Institute, University of Washington, Seattle, WA, USA. [20]Present address: Department of Genetics, Epigenetics Institute, Perelman School of Medicine, University of Pennsylvania, Philadelphia, PA, USA. [21]Present address: Altos Labs, San Diego, CA, USA. [22]These authors contributed equally: Harriet Dashnow, Thomas A. Sasani, Glennis A. Logsdon, Pille Hallast, Michelle D. Noyes, Zev N. Kronenberg, Tom Mokveld. ✉e-mail: ee3@uw.edu

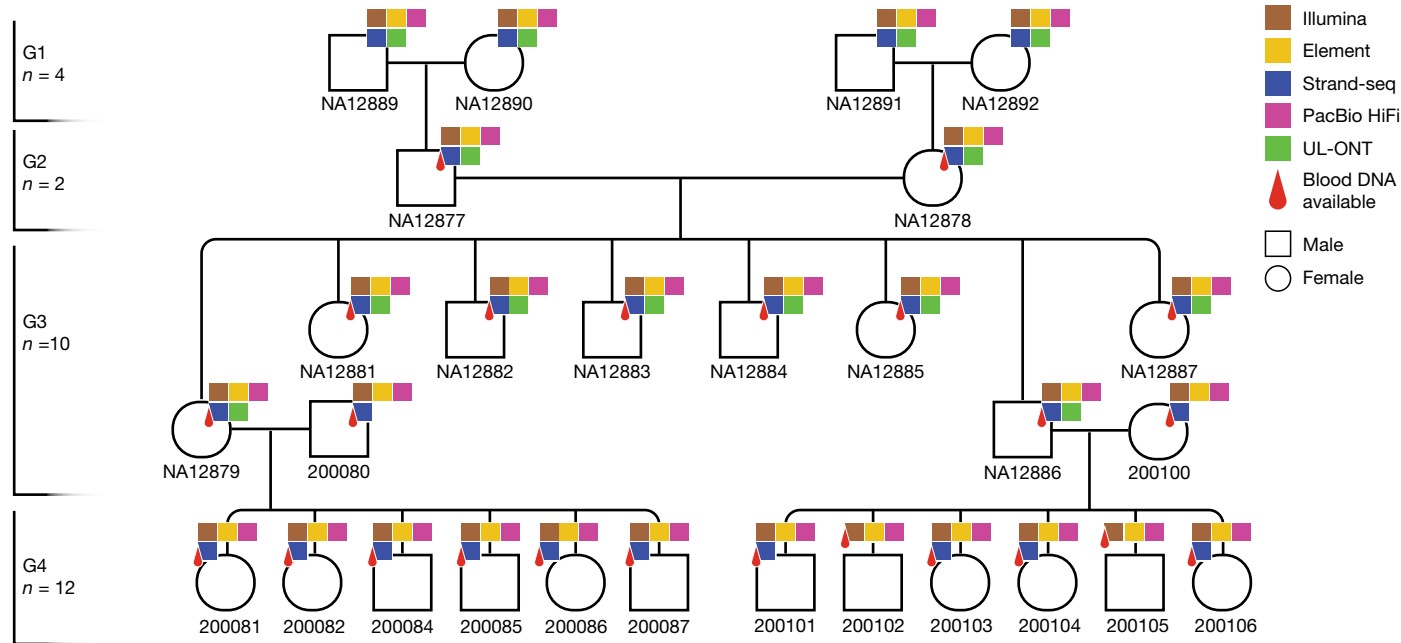

**Fig. 1 | Sequencing the CEPH 1463 pedigree with five technologies.** Twenty-eight members of the four-generation pedigree CEPH 1463 were sequenced using five orthogonal next-generation and LRS platforms: HiFi sequencing, Illumina and Element sequencing were performed on peripheral blood for G2–G4, and UL-ONT and Strand-seq data were generated on available lymphoblastoid cell lines for G1–G3. The pedigree dataset has been expanded to include the fourth generation and G3 spouses (200080 and 200100).

variation and reference genome effects. To achieve this, we focused on the four-generation, 28-member family CEPH 1463. This family has been intensively studied over the past three decades[11], and we sequenced the family members using five sequencing technologies with distinct and complementary error modalities. This particular pedigree has served as a benchmark for early linkage mapping studies[11,12] using short-read sequencing (SRS)[13] and continues to serve as reference for understanding human variation, including patterns of mosaicism[14,15].

Just as the initial T2T genome[1] served as a reference for understanding all regions of the genome, our objective was to create a reference truth set for both inherited and de novo variation.

## Genome sequence and assembly

We generated PacBio high-fidelity (HiFi), ultra-long Oxford Nanopore Technologies (UL-ONT), Strand-seq, Illumina and Element AVITI Biosciences (Element) whole-genome sequencing (WGS) data for most of the 28 members from a four-generation family (CEPH 1463 pedigree) (Fig. 1 and Supplementary Table 1).

For the purpose of variant discovery, we focused on generating long-read PacBio, short-read Illumina and Element data from blood-derived DNA to avoid cell-line-specific artefacts. We also used the corresponding cell lines to generate UL-ONT reads to construct near-T2T assemblies as well as Strand-seq data to detect large polymorphic inversions and evaluate assembly accuracy (Methods and Supplementary Table 2). In brief, we generated deep WGS data from multiple orthogonal sequencing platforms, focusing primarily on the first three generations (G1–G3) (Extended Data Fig. 1a), and used the fourth generation (G4) to validate de novo germline variants. We applied two hybrid genome assembly pipelines, Verkko[16] and hifiasm[17], to generate highly contiguous, phased genome assemblies for G1–G3, while G4 members were assembled using HiFi data only (Methods).

In summary, Verkko assemblies are the most contiguous (AuN (similar to average contig length measure): 102 Mb) (Extended Data Fig. 1b, Supplementary Figs. 1 and 2 and Supplementary Note 8) and we estimate that 63.3% (319 out of 504) of chromosomes across G1–G3 are near-T2T (Extended Data Fig. 1c). Moreover, 42.3% (213 out of 504)

of non-acrocentric chromosomes are spanned in a single contig with canonical telomere repeats at each end (Methods, Extended Data Fig. 1d, Supplementary Fig. 3 and Supplementary Table 3). We sequenced and assembled 288 centromeres (44.7%, 288 out of 644) across G1–G3 and note that different assemblers preferentially assembled different human centromeres (Methods, Extended Data Fig. 1e and Supplementary Fig. 4). Both the sequence (QV range, 47–58) and phasing accuracy are high (Methods, Supplementary Figs. 5–12 and Supplementary Table 4).

## A multigenerational variant callset

This data resource enables us to track the inheritance of any genomic segment and associated variants across all four generations (Extended Data Fig. 2a). We identified a total of 5.95 million single-nucleotide variants (SNVs) and indels and 35,662 structural variants (SVs)—all of which are Mendelian consistent across the second and third generations (Methods, Supplementary Table 5, Supplementary Fig. 13 and Data availability)[18]. Of the 5.95 million, 77% of small variants are supported by all three technologies, with variant calling from primary material helping to eliminate DNMs arising from cell line artefacts (Supplementary Note 1). LRS provides access to an additional approximately 260 Mb of the human genome (2.77 Gb) in contrast to the Genome in a Bottle (GIAB) (2.51 Gb)[19] or Illumina WGS (2.58 Gb)[13] data, including 201 Mb not present in either study. Some of the largest gains occur among SDs and their associated genes. We classified 85.5% (6,883 out of 8,048 merged SDs) of the SDs (coverage, >95%) as high confidence in comparison to 25.6% (2,060 out of 8,048 merged SDs) in the previous GIAB analysis, a major improvement for these highly copy-number variable regions[20]. We find that the majority (>91%) of known copy-number variable regions were stably transmitted in this pedigree, while the remaining 9% were often flagged as potentially misassembled (Supplementary Note 2). Similarly, we provide a comprehensive census of mobile element insertions (Methods, Supplementary Table 6, Supplementary Fig. 14 and Supplementary Note 3) and identify 120 inversions segregating in a Mendelian manner (21 were ambiguous) (Supplementary Table 7 and Supplementary Figs. 15–18). The latter

includes a rare inversion (~703 kb) overlapping a disease-associated copy-number variable region at chromosome 15q25.2 (ref. 21) (Supplementary Fig. 19) and an inverted duplication (~295 kb) at chromosome 16q11.2 (Supplementary Fig. 20).

## Sequence-resolved recombination map

Using three different approaches[13,22] (Methods and Extended Data Fig. 2b), we identify 539 meiotic breakpoints in G3 ($n = 8$) with respect to T2T-CHM13, with 99.8% (538 out of 539) supported by more than one approach (Supplementary Table 8 and Supplementary Fig. 21). From an initial resolution of around 3.4 kb, we further refined 90.4% (487 out of 539) of the breakpoints to a median size of about 2.5 kb based on direct genome comparisons between parent and a child (Methods and Supplementary Fig. 22). Notably, 191 breakpoints actually increase in size as a result of reference biases in T2T-CHM13 (Supplementary Fig. 23). We distinguish recombination breakpoints with very sharp transition between parental haplotypes from those with an extended region of homology at both parental haplotypes (Extended Data Fig. 2b and Supplementary Fig. 24). We also characterize 78 smaller haplotype segment 'switches' in G3 (median size of ~1 kb)[23–25] that would be consistent with either a double crossover or an allelic gene conversion event, although this is probably an underestimate due to our strict filtering criteria (Methods, Supplementary Table 9 and Supplementary Fig. 25). Extending recombination mapping to G4 chromosomes, we add 964 breakpoints for a total of 1,503 meiotic breakpoints across 22 transmissions (Supplementary Fig. 26). This includes 16 recombination hotspots, 11 of which are consistent with previously reported increased recombination rates[26] (Supplementary Table 8 and Supplementary Fig. 27).

Overall, 15–20% of paternal and maternal homologues are transmitted without a detectable meiotic breakpoint (that is, non-recombinant chromosomes) (Supplementary Fig. 28). We observe a significant excess (Wilcoxon signed-rank test, $P = 6.4 \times 10^{-5}$) of maternal recombination events with expected maternal to paternal breakpoint ratio of 1.4 (ref. 27) (Supplementary Fig. 29). Paternal recombination is significantly biased towards the ends of human chromosomes with 55 paternal recombination events mapping to within 2 Mb of the telomere in comparison to 1 event in female individuals[27–29] (Methods, Extended Data Fig. 2c and Supplementary Fig. 30). In G2–G3, we observed a decrease in crossover events with advancing parental age for both male and female germlines (Extended Data Fig. 2d). We modelled this observation across G1–G4 using a Poisson generalized linear model (GLM) with a log link and continued to observe a significant decrease in recombination breakpoints as a function of parental age and sex ($P = 7.17 \times 10^{-3}$ and $1.22 \times 10^{-9}$ for parental age and sex, respectively; Poisson GLM with a log link, AIC = 284.2) (Supplementary Fig. 31). Although there is no known biological mechanism that would lead to a decrease in both parental germlines, this observation runs counter to a population-level analysis based on SRS data[25,30,31]. We consider this observation to be preliminary until a larger number of families is analysed.

## De novo SNVs and small indels

To discover small variants, we examined HiFi reads aligned to T2T-CHM13, then used orthogonal ONT and Illumina data to confirm that a variant is in fact present in a sample and absent from parents (Methods). This strategy reduces platform bias but restricts DNM discovery to G2 ($n = 2$) and G3 ($n = 8$) individuals, as ONT data were not generated for G4. Our de novo callset included 755 SNVs and 73 indels across the autosomes (Fig. 2a), and 27 SNVs and 1 indel on the X chromosome. We used flanking SNVs to construct haplotypes, phase variants and trace a mutation back either to a parental gamete or the early embryo. We determined that a mutation occurred somatically, and probably early in embryonic development, if it met one of two criteria: it was

incompletely linked to a parental haplotype ($n = 122$) or, if it could not be phased, it had an allele balance significantly less than 0.5 across all three sequencing platforms ($n = 7$) (Fig. 2b), which was further confirmed using Element data (Supplementary Fig. 32). Moreover, we validated each postzygotic mutation (PZM) by tracing its haplotype backwards across generations and forwards for the four individuals with sequenced offspring (Supplementary Note 4).

Of the 62 PZMs in these four samples, 64.5% ($n = 40$) are transmitted to the next generation, compared with 97.2% of germline SNVs (242 out of 249) and 100% of indels (Extended Data Fig. 3). We found that 10 PZMs failed these haplotype-based validations, resulting in a final callset of 119 PZMs, accounting for 16% of total autosomal SNVs (745 de novo SNVs). Previous Illumina-based analysis of this family[14] identified 605 de novo SNVs of either germline (G2 and G3) or postzygotic (only G2) origin, 92.4% ($n = 559$) of which were represented in our final callset, while all but four of the absent variants failed validation with long-read data. We were able to identify an additional 72 PZMs in G3 for the first time, including a total of 186 novel DNMs, a 6.1% and 21% increase in germline SNV and indel discovery, respectively.

In total, 81.4% of germline small DNMs originate on paternal haplotypes (4.38:1 paternal:maternal ratio, Wilcoxon signed-rank test, $P < 2 \times 10^{-16}$), with a significant parental age effect of 1.55 germline DNMs per additional year of paternal age when fitting with linear regression (two-sided $t$-test, $P = 0.013$). By contrast, PZMs show no significant difference with respect to parental origin (1.38:1 paternal:maternal ratio, Wilcoxon signed-rank test, $P = 0.09$) and no parental age effects (Fig. 2c). Although our small sample size does not provide sufficient power to detect significant differences between the de novo and postzygotic mutational spectra (Supplementary Fig. 33a), we do observe a novel depletion of CpG>TpG PZMs ($\chi^2$ test, $P = 0.17$) and an enrichment of postzygotic T>A substitutions ($\chi^2$ test, $P = 0.268$) that has been previously observed[14].

We successfully assayed 91.9% of the autosomal genome (2.66 Gb) (Supplementary Fig. 33b and Supplementary Note 4). Excluding all variants classified as postzygotic, we find that the parental germline contributes $1.17 \times 10^{-8}$ SNVs per bp per generation (95% confidence interval (CI) = $1.08 \times 10^{-8}$–$1.27 \times 10^{-8}$). De novo SNVs are significantly enriched in repetitive sequences, as much as 2.8-fold in centromeres (95% CI = $1.79 \times 10^{-8}$–$5.51 \times 10^{-8}$ SNVs per bp per generation, two-sided $t$-test, $P = 0.017$) and 1.9-fold in SDs (95% CI = $1.64 \times 10^{-8}$–$2.88 \times 10^{-8}$ SNVs per bp per generation, two-sided $t$-test, $P = 0.0066$) (Fig. 2d, Supplementary Fig. 33c and Supplementary Table 10). We observed a lower PZM rate of $2.04 \times 10^{-9}$ SNVs per bp per generation (95% CI = $1.68 \times 10^{-9}$–$2.47 \times 10^{-9}$) across the autosomes, yet we see a 3.9-fold enrichment of PZMs in SDs (95% CI = $4.84 \times 10^{-9}$–$1.25 \times 10^{-8}$ SNVs per bp per generation, two-sided $t$-test, $P = 0.049$). Among PZMs transmitted to the next generation ($n = 33$ PZMs across four samples), we observe a 2.69-fold enrichment in SDs (95% CI = $1.15 \times 10^{-9}$–$1.08 \times 10^{-8}$ SNVs per bp per generation) that does not reach significance owing to the small sample size (two-sided $t$-test, $P = 0.218$).

## De novo TRs

Here we investigate tandem repeats (TRs), including short TRs (STRs, 1–6 bp motifs) and variable-number TRs (VNTRs, 7–1,000 bp motifs). We successfully genotyped 7.68 million out of 7.82 million TR loci (Methods) on HiFi data using the Tandem Repeat Genotyping Tool (TRGT)[32], across all members of the pedigree. Of those, 7.17 million (93.4%) loci were completely Mendelian concordant across all trios. We used TRGT-denovo to identify candidate DNMs at loci that were covered by at least 10 HiFi reads across all members of a given trio; on average, 6.88 million TR loci met this criterion[33]. We refined these putative DNMs through orthogonal sequencing and transmission (Methods). Element sequencing, generated from blood DNA, exhibits substantially lower error rates following homopolymer tracts[34], so we tested whether it

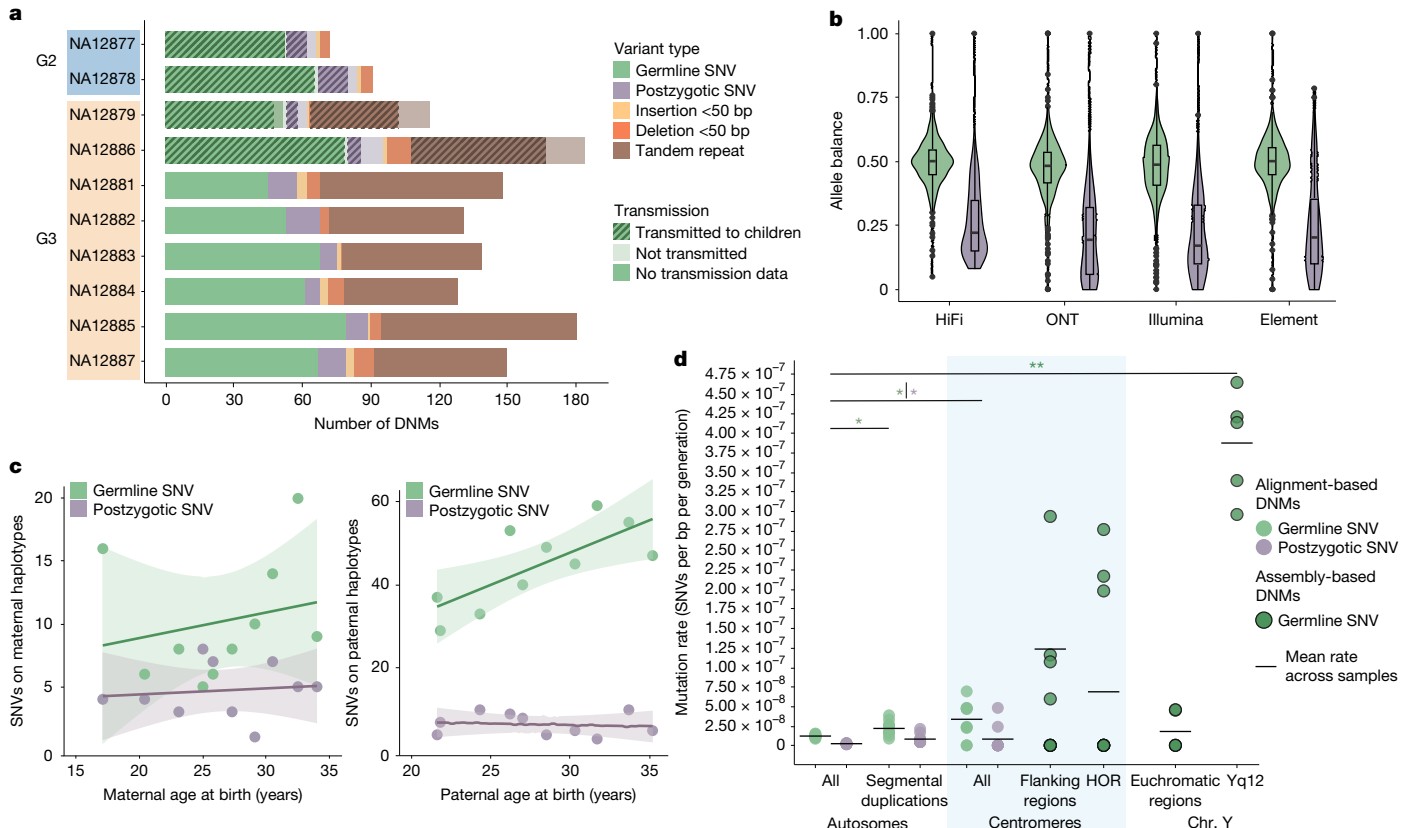

**Fig. 2 | Summary of DNM rates. a**, The number of de novo germline mutations, PZMs and indels (<50 bp) for the parents (G2) and eight children in CEPH1463. TR DNMs (<50 bp) are shown for G3 only because they have greater parental sequencing depth and we can assess transmission (Methods). The hatched bars show the number of SNVs confirmed as transmitting to the next generation. **b**, Germline SNVs (*n* = 626) have a mean allele balance of near 0.50 across the sequencing platforms, while the mean postzygotic SNV (*n* = 119) allele balance is less than 0.25. The box plots show the median (centre line), the interquartile range (IQR) (box limits) and the whiskers extend to 25% − 1.5 × IQR and 75% + 1.5 × IQR; outliers are shown as dots. **c**, A strong paternal age effect is observed for germline de novo SNVs (+1.55 DNMs per year; two-sided *t*-test, *P* = 0.013) but not for PZMs (*P* = 0.72). We observe no significant maternal age effect for DNMs (+0.20 DNMs per year, *P* = 0.54) or PZMs (*P* = 0.74). The solid lines are regression

lines that were fitted using a linear model function; the surrounding shaded areas represent their 95% confidence intervals. **d**, The estimated SNV DNM rate by region of the genome shows a significant excess of DNM for large repeat regions, including centromeres and SDs. Assembly-based DNM calls on the centromeres and Y chromosome (chr.) show an excess of DNM in the satellite DNA. A significant difference from the autosomal DNM or PZM rate was determined using two-sided *t*-tests; *$P$ < 0.05, **$P$ < 0.001. *P* values for each comparison are as follows: 0.0066 (alignment-based DNMs in SDs), 0.049 (alignment-based PZMs in SDs), 0.017 (alignment-based DNMs in centromeres), 0.34 (alignment-based PZMs in centromeres), 0.13 (assembly-based DNMs in centromeric flanking regions), 0.14 (assembly-based DNMs in centromeric HORs), 0.59 (assembly-based DNMs in chromosome Y euchromatic regions) and 0.00025 (assembly-based DNMs in Yq12).

could more accurately measure the length of homopolymers and other TR alleles. We observed low stutter in the Element data at homopolymers; across a random sample of 1,000 homozygous homopolymer loci called by TRGT, an average of 99.5% of Element reads perfectly support the TRGT-genotyped allele size in GRCh38, compared to 93.5% of Illumina sequencing reads (Supplementary Figs. 34 and 35).

We used the Element data to further validate de novo TR alleles called by TRGT-denovo. Owing to the short read length of Element data, we could assess only 80 out of 613 (13.1%) de novo STR alleles (average of 10 STRs per sample). We considered a DNM validated if Element reads supported the TRGT allele size in the child and did not support it in either parent (allowing for off-by-one base-pair errors; Methods). Of the 80 de novo STRs that we could assess, 56 (70%) passed our strict consistency criteria. The validation rate was lower at homopolymers (3 out of 20; 15%) than at non-homopolymers (53 out of 60; 88.3%), indicating that our estimates of mutation rates at homopolymers may be less precise. Using pedigree information, we required that candidate de novo TR alleles observed in the two G3 individuals with sequenced children (NA12879 and NA12886) be transmitted to at least one child in the subsequent generation (G4). Of the 128 de novo TR alleles observed in the two G3 individuals, 96 (75%) were transmitted to the next generation,

which is significantly lower than de novo SNVs reflecting the challenges that still remain in accurately characterizing de novo TRs.

After Element and transmission validation, we found an average of 65.3 TR DNMs (including STRs, VNTRs and complex loci) per sample and estimated a TR DNM rate of 4.74 × 10⁻⁶ per locus per haplotype per generation (95% CI = 4.06 × 10⁻⁶–5.43 × 10⁻⁶), with substantial variation across repeat motif sizes (Fig. 3a). Collectively, TR DNMs inserted or deleted a mean of 978 bp per sample or 15.0 bp per event (Supplementary Table 10). An average of 54.9 mutations were expansions or contractions of STR motifs, 2.6 affected VNTR motifs and 7.8 affected 'complex' loci comprising both STR and VNTR motifs. The STR mutation rate was 5.50 × 10⁻⁶ DNMs per locus per haplotype per generation (95% CI = 5.0 × 10⁻⁶–6.04 × 10⁻⁶). The VNTR mutation rate was 0.83 × 10⁻⁶ (95% CI = 0.51 × 10⁻⁶–1.27 × 10⁻⁶), predominantly comprising loci that could not be assessed in SRS studies. Several previous estimates of the genome-wide STR mutation rate considered only polymorphic STR loci; when we limited our analysis to STR loci that were polymorphic in the CEPH 1463 pedigree, we found 5.98 × 10⁻⁵ de novo STR events per locus per generation (95% CI = 5.43–6.57 × 10⁻⁵), which is broadly consistent with previous estimates of 4.95 × 10⁻⁵–5.6 × 10⁻⁵ (refs. 35–37). Overall, 75.0% of phased de novo TR alleles were paternal in origin

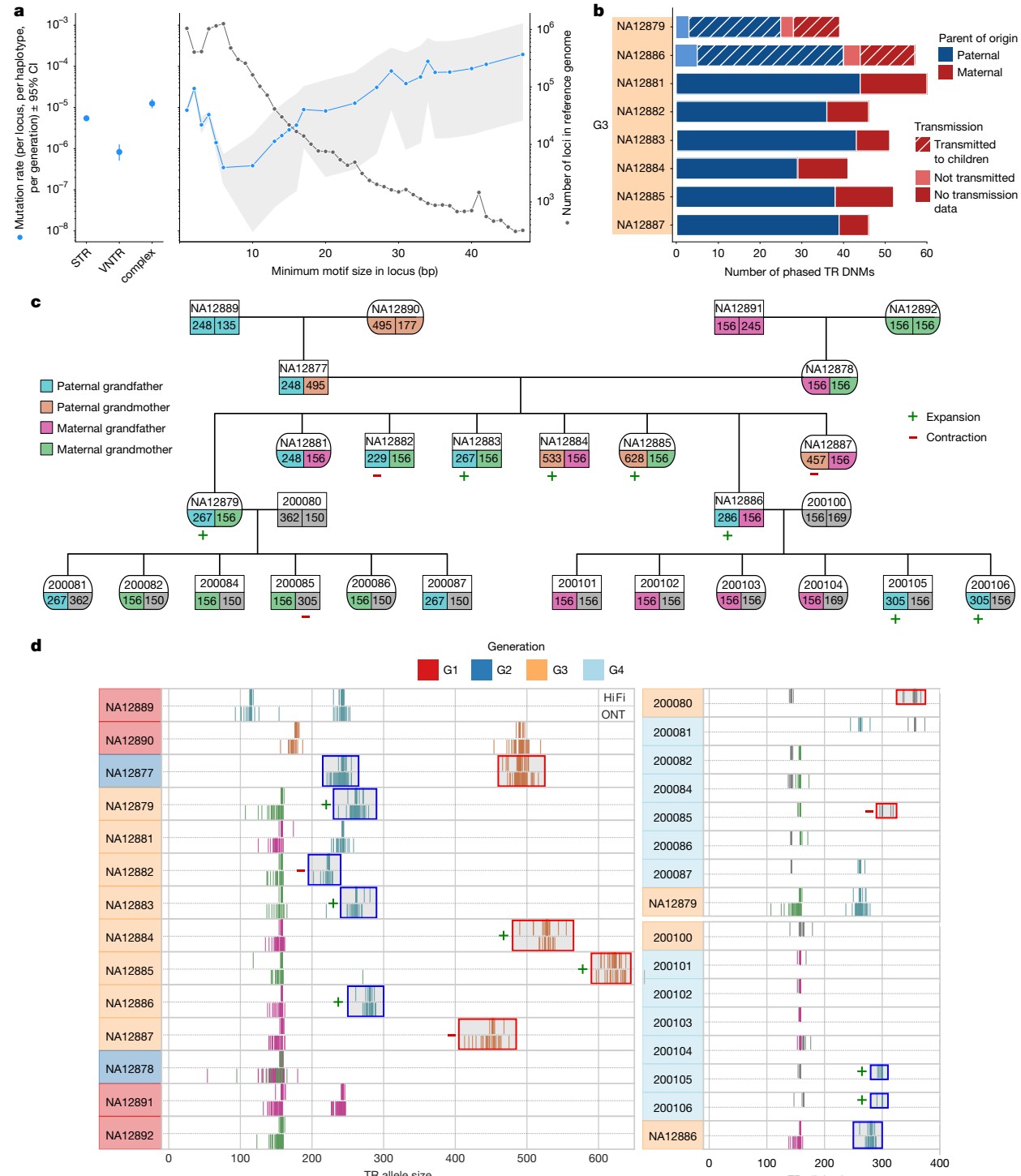

**Fig. 3 | TR DNMs show motif-size-dependent mutation rates, paternal bias and are highly recurrent at specific loci. a**, TR DNM rates (mutations per haplotype per locus per generation) are displayed for each TR class (STR, VNTR or complex) as a function of the minimum motif size observed at each TR locus ($n$ = 522) in the T2T-CHM13 reference genome (blue; left $y$ axis). The average number of loci of each motif size that passed filtering criteria in each individual are displayed in grey (right $y$ axis). The error bars denote the 95% Poisson CIs (computed using a $\chi^2$ distribution) around the mean mutation rate estimate. The mutation rates include all non-recurrent calls that pass TRGT-denovo filtering criteria and Element consistency analysis. **b**, The inferred parent-of-origin for confidently phased TR DNMs in G3. The hatching indicates transmission to at least one G4 child, where available. **c**, Pedigree overview of a recurrent VNTR locus at chromosome 8: 2376919–2377075 (T2T-CHM13) with motif composition GAGGCGCCAGGAGAGAGCGCT($n$)ACGGG($n$). Allele colouring

indicates inheritance patterns as determined by inheritance vectors, with grey representing unavailable data. The symbols denote the inheritance type relative to the inherited parental allele: plus (+) for de novo expansion and minus (−) for de novo contraction, shown only for the mutating alleles; the numbers indicate allele lengths in bp. De novo TR alleles are present in seven out of eight G3 individuals and transmit to four G4 individuals, with two expanding further after transmission. The spouse of a G3 individual (200080) carries a distinct TR allele that undergoes a de novo contraction in subsequent transmissions. **d**, Read-level evidence for the recurrent DNM in **c**, represented as vertical lines, obtained from individual sequencing reads, shown per sample. Where available, both HiFi (top) and ONT (bottom) sequencing reads are displayed. Colouring is consistent with the inheritance patterns in **c**; the outlined boxes with plus or minus markers highlight DNMs.

(Fig. 3b). The mutation rate for dinucleotide motifs was higher than for homopolymers, and we observed an increasing mutation rate with motif size for motifs greater than 6 bp in length (Fig. 3a). As reported in previous studies[35], larger TR loci (defined as the total length of the TR locus in the reference genome sequence) exhibited higher mutation rates (Supplementary Fig. 36a). We did not observe a significant bias towards expansions or contractions (two-sided binomial test, $P = 0.19$) (Supplementary Fig. 36b).

We identified a subset of TR loci that were recurrently mutated among members of the pedigree. We identified a high-confidence set of 32 loci (Methods and Supplementary Table 11): five showing intragenerational recurrence (observed DNMs in at least two G3 individuals) and 27 loci with intergenerational recurrence (observed DNMs in at least two generations). As they are observed only in a single generation, the five intragenerational DNMs may represent mosaicism in the parental germline, rather than recurrent mutational events. Notably, we observed three or more distinct de novo expansions or contractions at 16 of the loci that exhibited recurrence (Extended Data Table 1). As an example, we highlight an intergenerational recurrently mutated TR locus with ten unique de novo expansions and/or contractions (Fig. 3c,d). All allele transmissions are fully consistent with the inheritance vectors (Supplementary Note 5) and are supported by both HiFi and ONT reads.

### Centromere transmission and de novo SVs

Among the 288 completely assembled centromeres, we assessed 150 intergenerational transmissions (Fig. 4a). Comparing these assembled centromeres between parent and child, we identify 18 (12%) de novo SVs validated by both ONT and HiFi data with roughly equivalent numbers of insertions and deletions (Fig. 4b and Methods). All de novo SVs ($n = 8$) that had a child sequenced as part of this study confirmed transmission to the next generation (Supplementary Table 10). We find that 72.2% (13 out of 18) of SVs map to α-satellite higher-order repeat (HOR) arrays (Extended Data Fig. 4a) with the remainder (5 out of 18, 27.8%) corresponding to various pericentromeric flanking sequences but not flanking monomeric α-satellites. All α-satellite HOR de novo SV events involve integer changes in the basic α-satellite HOR cassettes specific to each centromere and range in size from 680 bp (one 4-mer α-satellite HOR on chromosome 9) to 12,228 bp (four 18-mer α-satellite HORs on chromosome 6) (Fig. 4c and Extended Data Fig. 4b). One transmission from chromosome 9 involves both a gain of 2,052 bp (six dimer α-satellite HOR units) and a loss of 1,710 bp (one 4-mer α-satellite HOR and three α-satellite dimer units) in a single G2-to-G3 transmission (Fig. 4d–f). The chromosome 6 centromere has the most recurrent structural events, with three being observed across three generations (Fig. 4a). The chromosome 6 centromere has the greatest number of nearly perfectly identical (>99.9%) α-satellite HORs (Extended Data Fig. 4c).

We also assessed 18 SV events for their potential effect on the hypomethylation pocket associated with the centromere dip region (CDR)—a marker of the site of kinetochore attachment[38,39] (Methods). We find that 11 SVs mapping outside of the CDR have a marginal effect on changing the centre point of the CDR (<100 kb) from one generation to another (Extended Data Fig. 4d,e), while SVs mapping within the CDR have a more marked effect (average shift of around 260 kb) and/or they completely alter the distribution of the CDR (Fig. 4g and Extended Data Fig. 4f,g). Although follow-up experiments using CENP-A chromatin immunoprecipitation–sequencing are needed to confirm the actual binding site of the kinetochore, these findings suggest that structural mutations may have epigenetic consequences in changing the position of kinetochore.

Finally, using 31 parent–child transmissions of centromeres (150.5 Mb), we identify 16 SNV DNMs in centromeres, including five within the α-satellite HOR arrays, for a DNM rate of $1.01 \times 10^{-7}$ mutations per bp per generation (95% CI = $5.75 \times 10^{-8}$–$1.63 \times 10^{-7}$). This rate is comparable to the rate from our read-based mapping approach, which identified 14 centromeric SNVs, albeit over more than 10 times the amount of sequence, resulting in a DNM rate of $3.27 \times 10^{-8}$ mutations per bp per generation (95% CI = $1.79 \times 10^{-8}$–$5.51 \times 10^{-8}$) (Fig. 2d and Supplementary Table 10). By combining the data, we estimate a significantly higher SNV DNM rate for centromeres of $4.94 \times 10^{-8}$ (two-sided $t$-test, $P = 0.017$). We believe that this is a conservative estimate because we required validation of all events by both the ONT and HiFi sequencing platforms.

### Y chromosome mutations

There are nine male members who carry the R1b1a-Z302 Y haplogroup across the four generations (Fig. 5a and Supplementary Table 12) and we use the great-grandfather (G1-NA12889; Fig. 1) Y-chromosome assembly as a reference for DNM detection across 48.8 Mb of the male-specific Y-chromosomal region (MSY) (Methods and Supplementary Note 6). The de novo assembly-based approach increases by more than twofold the number of accessible base pairs when compared to HiFi read-based calling but increases by more than sevenfold the discovery of de novo SNVs. In total, we identify 48 de novo SNVs in the MSY across the 5 G2–G3 male individuals, ranging from 7 to 11 SNVs per Y transmission (mean, 9.6; median, 10) (Supplementary Table 13). Only 2 SNVs map to the Y euchromatic regions, 1 to the pericentromeric regions and the remaining 45 out of 48 map to the Yq12 heterochromatic satellite regions (Fig. 5b). In total, we estimate a de novo SNV rate of $1.99 \times 10^{-7}$ mutations per bp per generation (95% CI = $1.59 \times 10^{-7}$–$2.39 \times 10^{-7}$) for the entire MSY. This estimate is an order of magnitude higher than that previously reported for Y euchromatic regions[40] due to access to Yq12 satellite DNA (Supplementary Table 13). We note that 13 out of 45 (29%) of the DNMs had 100% identical matches elsewhere in the Yq12 region (but not at orthologous positions) and probably result from interlocus gene conversion events within the *DYZ1*/*DYZ2* repeats[41] (Methods). We also identify a total of nine de novo indels (<50 bp, homopolymers excluded) ranging from 1–3 indels per sample (mean, 1.8 events per Y transmission) and five de novo SVs (≥50 bp) (Fig. 5b and Supplementary Table 13). The latter range from 2,416 to 4,839 bp in size, each affecting an entire *DYZ2* repeat unit(s), with an average of one SV per Y transmission. All applicable DNMs (SNVs, $n = 20$ out of 48; indels, $n = 6$ out of 9; SVs, $n = 4$ out of 5) are concordant with the expected transmission through generations (that is, from G2 to G3–G4 and from G3-NA12866 to his three male descendants in G4) (Fig. 5b). Overall, 82% (51 out of 62) of the DNMs identified on chromosome Y (42 out of 48 SNVs; 4 out of 9 of indels; and 5 out of 5 SVs) are located in regions where short reads cannot be reliably mapped (mapping quality = 0).

### De novo SVs

In total, we validated 41 de novo SVs across eight individuals (G3), including 16 insertions and 25 deletions (Methods) of which 68% (28 out of 41) originate in the paternal germline with a trend towards an increase in SVs with paternal age (Supplementary Fig. 37). Almost all SVs (40 out of 41) correspond to TRs, including mutation in centromeres, Y chromosome satellites and clustered SDs (Supplementary Table 10). We estimate around 5 SVs (95% CI = 3–7) per transmission affecting approximately 4.4 kb of DNA (median, 4,875 bp). If we exclude de novo SVs mapping to the centromere and Y chromosomes ($n = 14$), the median size of the events drops by an order of magnitude (median, 362 bp). Non-allelic homologous recombination (NAHR) has frequently been invoked as a mechanism to underlie TR expansions and contractions[42,43]. However, we find that none of the 27 euchromatic de novo SVs coincide with recombination crossovers (Supplementary Fig. 37e). This argues against NAHR between homologous chromosomes during

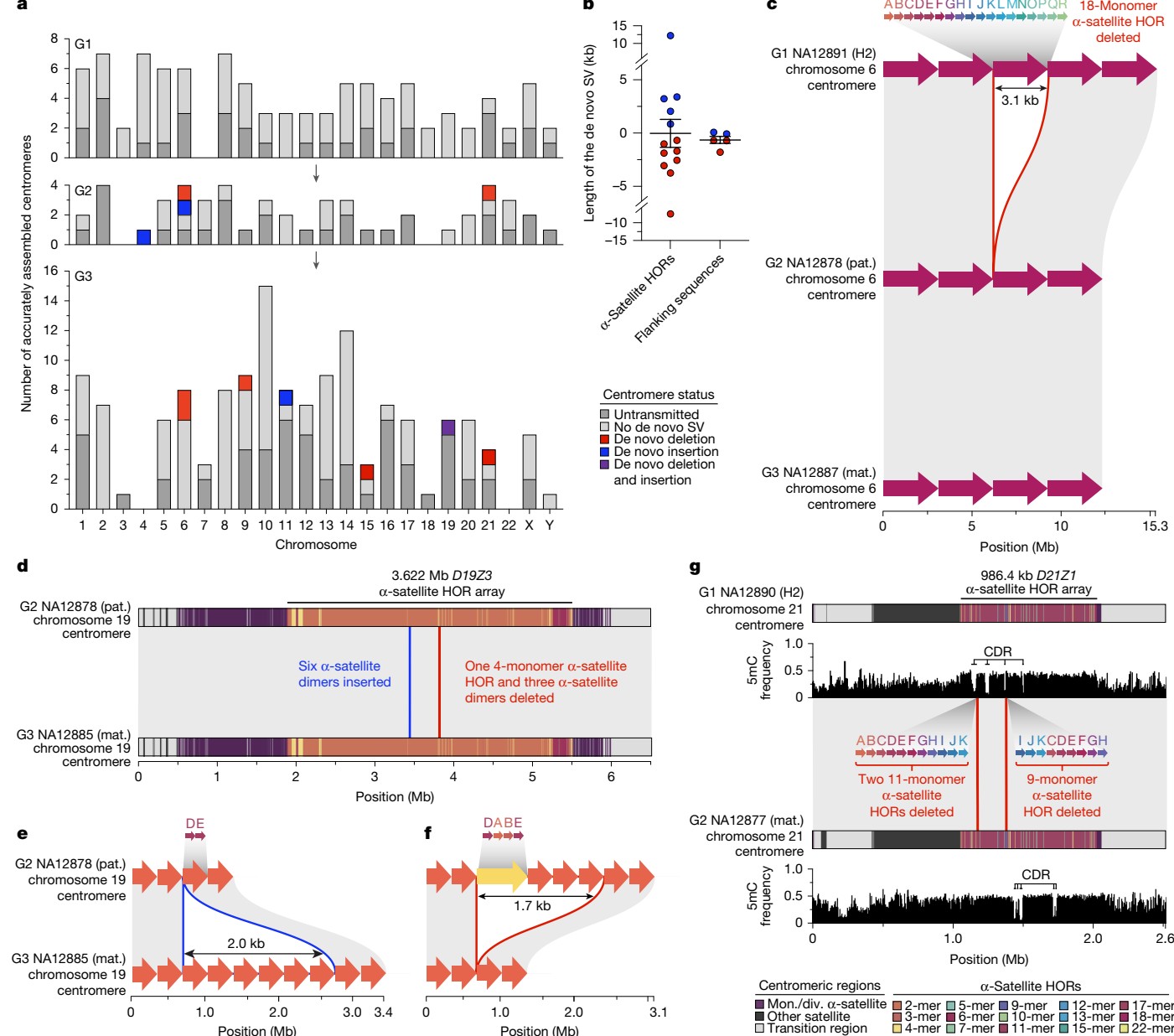

**Fig. 4 | De novo SVs among centromeres transmitted across generations.**
**a**, Summary of the number of correctly assembled centromeres (dark grey) as well as those transmitted to the next generation (light grey). Transmitted centromeres that carry a de novo deletion, insertion or both are coloured.
**b**, The lengths of the de novo SVs within α-satellite HOR arrays and flanking regions. **c**, An example of a de novo deletion in the chromosome 6 α-satellite HOR array in G2-NA12878 that was inherited in G3-NA12887. The red arrows over each haplotype show the α-satellite HOR structure, and the grey blocks between haplotypes show syntenic regions. The deleted region is highlighted by a red outline. Mat., maternal; pat., paternal. **d**, An example of a de novo insertion

and deletion in the chromosome 19 α-satellite HOR array of G3-NA12885.
**e**,**f**, Magnification of the α-satellite HOR structure of the inserted (blue outline; **e**) and deleted (red outline; **f**) α-satellite HORs from **d**. The coloured arrows at the top of each haplotype show the α-satellite HOR structure. **g**, Example of two de novo deletions in the chromosome 21 centromere of G2-NA12877. The deletions reside within a hypomethylated region of the centromeric α-satellite HOR array, known as the CDR, which is thought to be the site of kinetochore assembly. The deletion of three α-satellite HORs within the CDR results in a shift of the CDR by around 260 kb in G2-NA12877.

meiosis I as the primary mechanism for their origin, although we cannot preclude other mechanisms associated with double-stranded breaks not involving recombination. We identify one retrotransposition event: a full-length (3,407 bp long) de novo insertion of an SVA element (SVAF subfamily) (G3-NA12887)[44] with the predicted donor mapping around 23 Mb upstream (Fig. 5c and Supplementary Fig. 38). This insertion is present at a low frequency (around 11% of reads) in the parent (G2-NA12878) but not in the grandparental transmitting haplotype, consistent with a germline mosaic event arising in G2 postzygotically (Fig. 5d and Supplementary Fig. 39).

## Discussion

Most DNM studies[40,45–49] are based on SRS data from large groups of trios and converge on around 60–70 DNMs per generation; however, these studies often exclude highly mutable regions of the genome[7]. Our multiplatform and multigenerational, assembly-based approach provides access to some of the most repetitive regions, such as centromeres and heterochromatic regions on the Y chromosome. The use of parental references in addition to the standard references and the ability to confirm transmissions across subsequent generations

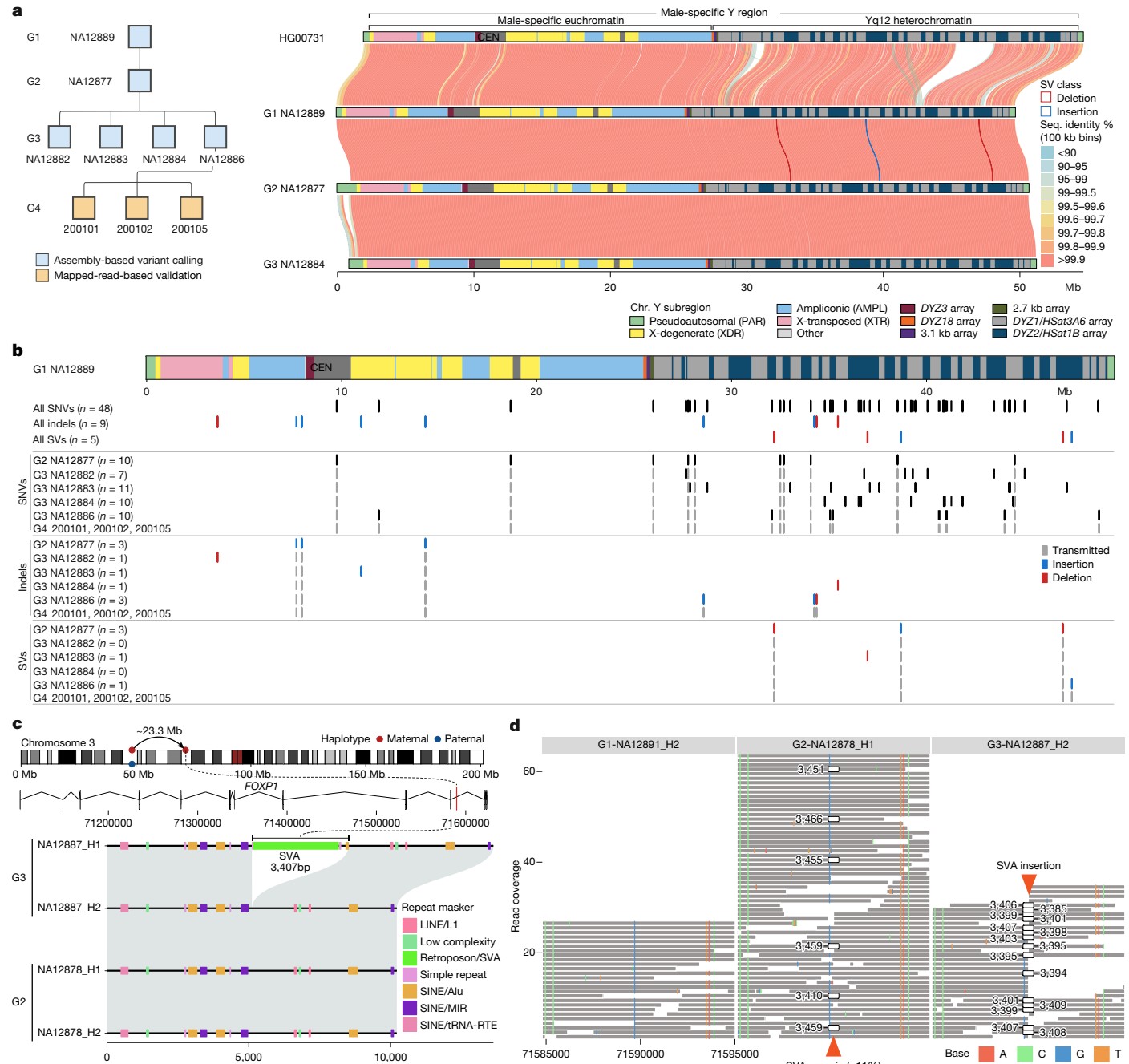

**Fig. 5 | Chromosome Y and an example of a de novo mobile element.**
**a**, Pedigree of the nine male individuals carrying the R1b1a-Z302 Y chromosomes (left) and pairwise comparison of Y assemblies: closely related Y from HG00731 (R1b1a-Z225) and the most contiguous R1b1a-Z302 Y assemblies from three generations. Y-chromosomal sequence classes are shown with the pairwise sequence identity between samples in 100 kb bins, with quality-control-passed SVs identified in the pedigree male individuals shown as blue and red outlines. **b**, Summary of chromosome Y DNMs. Top, the structure of chromosome Y of G1-NA12889. Below the Y structure, all of the identified DNMs across G1–G3 Y assemblies are shown. Bottom, breakout by mutation class and by sample. DNMs that show evidence of transmission from G2 to G3–G4, and from G3-NA12886 to his male descendants in G4 are shown in grey. **c**, De novo SVA insertion in G3-NA12887. **d**, HiFi read support for the de novo SVA insertion in G3-NA12887.

improves both sensitivity and specificity. In this multigenerational pedigree, we estimate a range of 98–206 DNMs per transmission (average of 152 per generation) and observe a strong paternal de novo bias (70–80%) and an increase with advancing paternal age, not only for SNVs but also for indels and SVs, including TRs.

The rate of de novo SNVs varies by more than an order of magnitude depending on the genomic context, consistent with recent human population-based analyses[7,50] and theoretical predictions[51]. SD regions show an 88% increase ($2.2 \times 10^{-8}$ (95% CI = $1.64 \times 10^{-8}$–$2.88 \times 10^{-8}$) versus

$1.17 \times 10^{-8}$). This is driven by SDs with >95% identity. We also observe a significant decrease in the de novo transition/transversion ratio compared with the genome ($\chi^2$ test, $P = 0.0109$) as predicted[7] (Supplementary Note 7). We estimate that satellite DNA in the Yq12 heterochromatic region[41,52] is at least 30 times more mutable than autosomal euchromatin ($3.86 \times 10^{-7}$ mutations per bp per generation). It is composed of thousands of short satellite DNA repeats (*DYZ1/Hsat3A6* and *DYZ2/Hsat1B*) organized into Mb blocks that are >98% identical[41,52]. This, along with the fact that 29% of mutational changes match to non-orthologous

sites in Yq12, is consistent with 'interlocus gene conversion' driving this >20-fold excess, potentially as a result of increased sister chromatid exchange events[41].

Previous studies predicted that 6–10% of DNMs are not germline in origin, but instead arise sometime after fertilization, giving rise to a mosaic variant[14,53]. This distinction has been based on allele balance thresholds[53] or incomplete linkage to nearby SNVs across three generations[14]. LRS increases sensitivity by assigning nearly every de novo SNV to a parental haplotype and define PZM by its incomplete linkage to that haplotype. We classify 16% of de novo SNVs as postzygotic in origin ($n$ = 119 PZMs/745 de novo SNVs). As all sequencing data in this study are derived from blood, we cannot demonstrate that every PZM is present in multiple tissues, but we can use transmission to the next generation as a proxy, as it reveals that the mutation is also present in germ cells. PZMs account for 12% of all SNVs transmitted to the next generation ($n$ = 33 PZMs/275 transmitted SNVs), an increase over previous estimates. Early cell divisions of human embryos are frequently error prone[54,55] with an accelerated rate of cell division potentially contributing to the large fraction of PZMs with high (>25%) allele balance. Such events would previously have been classified as germline but, consistent with PZM expectations, we find no paternal bias associated with these DNMs (Fig. 2c).

TRs are among the most mutable loci of our genome[36,56,57], with the number of such de novo events comparable to germline SNVs[58] but affecting more than an order of magnitude more base pairs per generation. We find a threefold differential in TR DNM rate with increasing repeat number and motif length generally correlating with mutation rate. However, we observe an apparent mutation rate trough between dinucleotides and larger motif lengths (>10 bp) (Fig. 3b), which may reflect different mutational mechanisms based on locus size, motif length and complexity. For example, larger TR motifs may be more likely to mutate through NAHR, synthesis-dependent strand annealing or interlocus gene conversion while mutational events at STRs may be biased toward traditional replication-based slippage mutational mechanisms[42,43]. Consistent with some earlier genome-wide analyses of minisatellites[59], we did not find evidence that TR changes are mediated by unequal crossover between homologues during meiosis as none of our TR de novo SVs ($n$ = 27) coincided with recombination breakpoints. Of particular interest in this regard is the discovery of 32 recurrently mutated TRs—loci rarely discovered out of the context of unstable disease alleles[60]. At five of these recurrent loci, we discovered multiple DNMs within a single generation (G3); these DNMs may be the outcome of germline mosaicism in a G2 parent or the activity of hypermutable TRs. Nearly all of these highly recurrent de novo events produced TR alleles that are significantly longer than the average short-read length and were detectable only using LRS. This includes changes in the length of around 7% of human centromeres in which insertions and deletions all occur as multiples of the predominant HOR unit[56]. The rate of de novo SVs increased from previous estimates of 0.2–0.3 per generation[15,61] to 3–4 de novo SVs per generation reported in this study.

There are several limitations to this study. First, homopolymers still remain challenging even with the use of Element data as longer alleles and motifs embedded in larger repeats are still not reliably assayed with short reads. Second, we were unable to characterize DNMs in the acrocentric regions due to the repetitive nature of the regions and rampant ectopic recombination[4]. Third, we limited DNM discovery to the first three generations of only one multigenerational family and used G4 for validation purposes of transmitted variants. We acknowledge that familial variation depends on the genetic background[14,36,62] and, therefore, many more families will be required to establish a reliable estimate of the mutation rate, especially for complex regions of the genome. In that regard, it is perhaps noteworthy that efforts are underway to characterize additional pedigrees. Notwithstanding, this study highlights that a single sequencing technology and a single human genome reference are insufficient to comprehensively estimate mutation rates. Multigenerational resources such as these will further refine DNM estimates and serve as another useful benchmark[63] for new algorithms and new sequencing technologies.

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

# Methods

## Ethics declarations

**Human participants.** Informed consent was obtained from the CEPH/Utah individuals, and the University of Utah Institutional Review Board approved the study (University of Utah IRB reference IRB_00065564). This includes informed consent for publication of research data for 23 family members; the remaining 5 provided informed consent for biobanking with controlled access (Data availability).

## Cell lines

Cell lines for 14 members of the CEPH 1463 family (G1-GM12889, G1-GM12890, G1-GM12891, G1-GM12892, G2-GM12877, G2-GM12878, G3-GM12879, G3-GM12881, G3-GM12882, G3-GM12883, G3-GM12884, G3-GM12885, G3-GM12886 and G3-GM12887) were obtained from Coriell Institute for Medical Research (CEPH collection). Cell lines for G3 spouses and G4 family members ($n = 13$) were generated in-house as EBV transformed lymphoblastoid cell lines and include: G3-200080-spouse, G4-200081, G4-200082, G4-200084, G4-200085, G4-200086, G4-200087, G3-200100-spouse, G4-200101, G4-200102, G4-200103, G4-200104 and G4-200106.

All cell lines were authenticated by WGS of the DNA and subsequent variant calling to match the expected sex of the individual and sequencing results from blood-derived DNA from the same individual. Furthermore, we explored whether the obtained sequencing data match the expected inheritance patterns of parents and offspring. To our knowledge, none of the cell lines mentioned above were tested for mycoplasma contamination.

## Sample and DNA preparation

Family members from G2 and G3 were re-engaged for the purpose of updating informed consent and health history, and for enrolling their children (G4) and the marry-in parent (G3). Archived DNA from G2 and G3 was extracted from whole blood. Newly enrolled family members underwent informed consent, and blood was obtained for DNA and cell lines. DNA was extracted from whole blood using the Flexigene system (Qiagen 51206). All samples are broadly consented for scientific purposes, which makes this dataset ideal for future tool development and benchmarking studies.

## Sequence data generation

Sequencing data from orthogonal short- and long-read platforms were generated as follows:

**Illumina data generation.** Illumina WGS data for G1–G3 were generated as previously described[14]. Illumina WGS data for G4 and marry-in spouses for G3 were generated by the Northwest Genomics Center using the PCR-free TruSeq library prep kit and sequenced to approximately 30× on the NovaSeq 6000 with paired-end 150 bp reads.

**PacBio HiFi sequencing.** PacBio HiFi data were generated according to the manufacturer's recommendations. In brief, DNA was extracted from blood samples as described or cultured lymphoblasts using the Monarch HMW DNA Extraction Kit for Cells & Blood (New England Biolabs, T3050L). At all steps, quantification was performed with Qubit dsDNA HS (Thermo Fisher Scientific, Q32854) measured on DS-11 FX (Denovix) and the size distribution checked using FEMTO Pulse (Agilent, M5330AA and FP-1002-0275.) HMW DNA was sheared with the Megaruptor 3 (Diagenode, B06010003 & E07010003) system using the settings 28/30, 28/31 or 27/29 based on the initial quality check to target a peak size of ~22 kb. After shearing, the DNA was used to generate PacBio HiFi libraries using the SMRTbell prep kit 3.0 (PacBio, 102-182-700). Size selection was performed either with diluted AMPure PB beads according to the protocol, or with Pippin HT using a high-pass cut-off between 10–17 kb based on shear size (Sage Science,

HTP0001 and HPE7510). Libraries were sequenced either on the Sequel II platform on SMRT Cells 8M (PacBio, 101-389-001) using Sequel II sequencing chemistry 3.2 (PacBio,102-333-300) with 2 h pre-extension and 30 h movies on SMRT Link v.11.0 or 11.1, or on the Revio platform on Revio SMRT Cells (PacBio, 102-202-200) and Revio polymerase kit v1 (PacBio, 102-817-600) with 2 h pre-extension and 24 h movies on SMRT Link v.12.0.

**ONT sequencing.** To generate UL sequencing reads >100 kb, we used ONT sequencing. Ultra-high molecular mass gDNA was extracted from the lymphoblastoid cell lines according to a previously published protocol[64]. In brief, $3–5 × 10^7$ cells were lysed in a buffer containing 10 mM Tris-Cl (pH 8.0), 0.1 M EDTA (pH 8.0), 0.5% (w/v) SDS, and 20 mg ml$^{-1}$ RNase A for 1 h at 37 °C. Then, 200 µg ml$^{-1}$ proteinase K was added, and the solution was incubated at 50 °C for 2 h. DNA was purified through two rounds of 25:24:1 phenol–chloroform–isoamyl alcohol extraction followed by ethanol precipitation. Precipitated DNA was solubilized in 10 mM Tris (pH 8.0) containing 0.02% Triton X-100 at 4 °C for 2 days.

Libraries were constructed using the Ultra-Long DNA Sequencing Kit (ONT, SQK-ULK001) with modifications to the manufacturer's protocol: ~40 µg of DNA was mixed with FRA enzyme and FDB buffer as described in the protocol and incubated for 5 min at room temperature, followed by heat inactivation for 5 min at 75 °C. RAP enzyme was mixed with the DNA solution and incubated at room temperature for 1 h before the clean-up step. Clean-up was performed using the Nanobind UL Library Prep Kit (Circulomics, NB-900-601-01) and eluted in 450 µl EB. Then, 75 µl of library was loaded onto a primed FLO-PRO002 R9.4.1 flow cell for sequencing on the PromethION (using MinKNOW software v.21.02.17–23.04.5), with two nuclease washes and reloads after 24 and 48 h of sequencing. All G1–G3 ONT base calling was done with Guppy (v.6.3.7).

**Element (AVITI) sequencing.** Element WGS data were generated according to the manufacturer's recommendations. In brief, DNA was extracted from whole blood as described above. PCR-free libraries were prepared using mechanical shearing, yielding ~350 bp fragments, and the Element Elevate library preparation kit (Element Biosciences, 830-00008). Linear libraries were quantified by quantitative PCR and sequenced on AVITI 2 × 150 bp flow cells (Element Biosciences, not yet commercially available). Bases2Fastq Software (Element Biosciences) was used to generate demultiplexed FASTQ files.

**Strand-seq library preparation.** Single-cell Strand-seq libraries were prepared using a streamlined version of the established OP-Strand-seq protocol[65] with the following modifications. In brief, EBV cells from G1–3 were cultured for 24 h in the presence of BrdU and nuclei with BrdU in the G1 phase of the cell cycle were sorted using fluorescence-activated cell sorting as described previously[65]. Next, single nuclei were dispensed into individual wells of an open 72 × 72 well nanowell array and treated with heat-labile protease, followed by digestion of DNA with the restriction enzymes AluI and HpyCH4V (NEB) instead of micrococcal nuclease (MNase). Next, fragments were A-tailed, ligated to forked adapters, UV-treated and PCR-amplified with index primers. The use of restriction enzymes results in short, reproducible, blunt-ended DNA fragments (>90% smaller than 1 kb) that do not require end-repair before adapter ligation, in contrast to the ends of DNA generated by MNase. Omitting end-repair enzymes allows dispensing of index primers in advance of dispensing individual nuclei. The pre-spotted, dried primers survive and do not interfere with the library preparation steps before PCR amplification. Pre-spotting of index primers is more reliable than the transfer of index primers between arrays during library preparation as described previously[65]. Strand-seq libraries were pooled and cleaned with AMPure XP beads, and library fragments between 300 and 700 bp were gel purified before PE75 sequencing on either the NextSeq 550 or the AVITI (Element Biosciences) system.

Supplementary Fig. 40 shows examples of Strand-seq libraries made with restriction enzymes.

## Strand-seq data post-processing

The demultiplexed FASTQ files were aligned to both GRCh38 and T2T-CHM13 reference assemblies (Supplementary Table 14) using BWA[66] (v.0.7.17-r1188) for standard library selection. Aligned reads were sorted by genomic position using SAMtools[67] (v.1.10) and duplicate reads were marked using sambamba[68] (v.1.0). Libraries passing quality filters were pre-selected using ASHLEYS[69] (v.0.2.0). We also evaluated such selected Strand-seq libraries manually and further excluded libraries with an uneven coverage, or an excess of 'background reads' (reads mapped in opposing orientation for chromosomes expected to inherit only Crick or Watson strands) as previously described[70]. This is done to ensure accurate inversion detection and phasing.

## Strand-seq inversion detection

Polymorphic inversions for G1–G3 were detected by mapping Strand-seq read orientation with respect to the reference genome as previously described[71,72]. For each sample, we selected 60+ Strand-seq libraries (range, 62–90) with a median of around 274,000 reads with mapping quality ≥10 per library, translating to about 0.67% genome (T2T-CHM13) being covered per library (Supplementary Fig. 41). Then we ran breakpointR[73] (v.1.15.1) across selected Strand-seq libraries to detect points of strand-state changes[73]. We used these results to generate sample-specific composite files using breakpointR function 'synchronizeReadDir' as described previously[71]. Again, we ran breakpointR on such composite files to detect regions where Strand-seq reads map in reverse orientation and are indicative of an inversion. Lastly, we manually evaluated each reported inverted region by inspection of Strand-seq read mapping in UCSC Genome Browser[74] and removed any low-confidence calls. We phased all inversions using Strand-seq data as well and then synchronized the phase with phased genome assemblies based on haplotype concordance. Lastly, we evaluated the Mendelian concordance of detected and fully phased inversions. We mark sites where at least half of the G3 samples were fully phased by Strand-seq and concordant with possible inherited G2 parental alleles as being Mendelian concordant (Supplementary Table 7).

## Generation of phased genome assemblies

Phased genome assemblies were generated using two different algorithms, namely Verkko (v.1.3.1 and v.1.4.1)[16] and hifiasm (UL) with ONT support (v.0.19.5)[17]. Owing to active development of the Verkko and hifiasm algorithms, assemblies were generated with two different versions. Phased assemblies for G2–G3 were generated using a combination of HiFi and ONT reads using parental Illumina *k*-mers for phasing. To generate phased genome assemblies of G1, we still used a combination of HiFi and ONT reads with the Verkko pipeline and used Strand-seq to phase assembly graphs[75]. Lastly, G4 samples were assembled using HiFi reads only with hifiasm (v.0.19.5).

Note that trio-based phasing with Verkko assigns maternal to haplotype 1 and paternal to haplotype 2. By contrast, for hifiasm assemblies, we report switched haplotype labelling such that haplotype 1 is paternal and haplotype 2 is maternal to match HPRC standard for hifiasm assemblies.

## Evaluation of phased genome assemblies

To evaluate the base pair and structural accuracy of each phased assembly, we used a multitude of assembly evaluation tools as well as orthogonal datasets such as PacBio HiFi, ONT, Strand-seq, Illumina and Element data. Known assembly issues are listed in Supplementary Table 4. We note that we fixed four haplotype switch errors in our assembly-based variant callsets to avoid biases in subsequent analysis. The assembly-quality terminology used in this Article is described in Supplementary Note 8.

**Strand-seq validation.** We used Strand-seq data to evaluate directional and structural accuracy of each phased assembly. First, we aligned selected Strand-seq libraries for each sample to the phased de novo assembly using BWA[66] (v.0.7.17-r1188). We next ran breakpointR[73] (v.1.15.1) using aligned BAM files as the input. We then created directional composite files using the breakpointR function createCompositeFiles followed by running breakpointR on such composite files using the runBreakpointR function. This provided us, for any given sample, with regions where strand-state changes across all single-cell Strand-seq libraries. Many such regions point to real heterozygous inversions. However, regions where Strand-seq reads mapped in opposite orientation with respect to surrounding regions are probably caused by misorientation. Moreover, positions where the strand state of Strand-seq reads changes repeatedly in multiple libraries might be a sign of an assembly misjoin and such regions were investigated more closely to rule out any such large structural assembly inconsistencies.

**Read to assembly alignment.** To evaluate de novo assembly accuracy, we aligned sample-specific PacBio HiFi reads to their corresponding phased genome assemblies using Winnowmap[76] (v.2.03) with the following parameters:

```
-I 10G -Y -ax map-pb --MD --cs -L --eqx
```

**Flagger validation.** Flagger[9] was used to detect misassemblies using HiFi read alignments to the assemblies and the assemblies aligned to the reference genome. Regions were flagged on the basis of read alignment divergence and specific reference-biased regions. A reference-specific BED file (chm13v2.0.sd.bed) was used, setting a maximum read divergence of 2% and specifying reference-biased blocks. These flagged regions were analysed to identify collapses, false duplications, erroneous regions and correctly assembled haploid blocks with the expected read coverage.

We used Flagger v.0.3.3 (https://github.com/mobinasri/flagger) to run the flagger_end_to_end WDL.

Required inputs include following:
(1) Read-to-contig alignments—Winnowmap alignments of all HiFi reads to the assembly (hap1, hap2 and unassigned.fasta)
(2) A combined assembly fasta file with hap1, hap2 and unassigned contigs
(3) BAM alignments of assembly to the CHM13v2.0 reference

hap1, hap2 and unassigned fasta files of the assembly were aligned to CHM13v2.0 using a pipeline available at GitHub (https://github.com/mrvollger/asm-to-reference-alignment).

**NucFreq validation.** NucFreq[77] (v.0.1) was used to calculate nucleotide frequencies for HiFi reads aligned using Winnowmap[76] (v.2.03). This was used to identify regions of collapses, where the second-highest nucleotide count exceeded 5; and misassembly, where all nucleotide counts were zero.

The NucFreq analysis pipeline is available at GitHub (https://github.com/mrvollger/NucFreq).

**Assembly base-pair quality.** To evaluate the accuracy of the genome assembly, we used a pipeline that uses Meryl[78] (v.1.0) to count the *k*-mers of length 21 from Illumina reads using the following command:

```
meryl k=21 count {input.fastq} output {output.meryl}
```

We then used Merqury[78] (v.1.1), which compares the *k*-mers from the sequencing reads against those in the assembled genome and flags discrepancies where *k*-mers are uniquely found only in the assembly. These unique *k*-mers indicate potential base-pair errors. Merqury then calculates the quality value based on the *k*-mer survival rate, estimated

from Meryl's $k$-mer counts, providing a quantitative measure to assess the completeness and correctness of the genome assembly.

**Gene completeness validation.** To evaluate the completeness of single-copy genes in our assemblies, we used compleasm[79] (v.0.2.4). Further details are available at GitHub (https://github.com/huang-nengCSU/compleasm).

We ran compleasm with the following parameters:

```
compleasm.py run -a {assembly.fasta} -o results/{sample.id} -t {threads} -l {params.lineage} -m {params.mode} -L {params.mb_downloads}
-l primates
-m busco
-L {params.mb_downloads}
```

and downloaded using: compleasm_kit/compleasm.py download primates.

**Assembly to reference alignment.** All de novo assemblies were aligned to both GRCh38 as well as to the complete version of the human reference genome T2T-CHM13 (v2) using minimap2 (ref. 80) (v.2.24) with the following command:

```
minimap2 -K 8G -t {threads} -ax asm20 \
--secondary=no --eqx -s 25000 \
{input.ref} {input.query} \
| samtools view -F 4 -b - > {output.bam}
```

A complete pipeline for this reference alignment is available at GitHub: (https://github.com/mrvollger/asm-to-reference-alignment).

We also generated a trimmed version of these alignments using the rustybam (v.0.1.33) (https://github.com/mrvollger/rustybam) function trim-paf to trim redundant alignments that mostly appear at highly identical SDs. With this, we aim to reduce the effect of multiple alignments of a single contig over these duplicated regions.

**Definition of stable diploid regions.** For this analysis, we use assembly to reference alignments (see the 'Assembly to reference alignment' section), reported as PAF files. We used trimmed PAF files reported by the rustybam trim-paf function. Stable diploid regions were defined as regions where phased genome assemblies report exactly one contig alignment for haplotype 1 as well as haplotype 2 and are assigned as '2n' regions. Any region with two or more alignments per haplotype is assigned as 'multi' alignment. Lastly, regions with only single-contig alignment in a single haplotype are assigned as '1n' regions. These reports were generated using the 'getPloidy' R function (Code availability).

**Detection and analysis of meiotic recombination breakpoints**
We constructed a high-resolution recombination map of G2 and G3 individuals using three orthogonal approaches that differ either on the basis of the underlying sequencing technology or on the detection algorithm applied to the data. The first approach is based on chromosome-length haplotypes extracted from Strand-seq data using R package StrandPhaseR[81] (v.0.99). The second approach uses inheritance vectors derived from Mendelian consistency of small variants across the family pedigree[13]. Our final approach uses trio-based phased genome assemblies followed by small variant calling using PAV and Dipcall to more precisely define the meiotic breakpoints.

By contrast, a recombination map of G4 individuals was constructed using a combination of Strand-seq data for G3 spouses and an assembly-based variant callset (Dipcall) of G4 samples. Owing to the use of a low-variant density of Strand-seq-based haplotypes of G4 spouses, reported recombination breakpoints are of lower resolution in comparison to G2 and G3 samples (Supplementary Table 8).

**Detection of recombination breakpoints using circular binary segmentation.** To map meiotic recombination breakpoints using circular binary segmentation, we used two different datasets. The first dataset represents phased small variants (SNVs and indels) as reported by Strand-seq-based (SSQ) phasing[22,81]. The other is based on small variants reported in trio-based phased assemblies either by PAV[8] (v.2.3.4) or Dipcall[82] (v.0.3). With this approach, we set to detect recombination breakpoints as positions where a child's haplotype switches from matching H1 to H2 of a given parent or vice versa. To detect these positions, we first established which homologue in a child was inherited from either parent by calculating the level of agreement between child's alleles and homozygous variants in each parent. Next, we compared each child's homologue to both homologues of the corresponding parent and encoded them as 0 or 1 if they match H1 or H2, respectively. We applied a circular binary segmentation algorithm on such binary vectors by using the R function fastseg implemented in the R package fastseg[83] (v.1.46.0) with the following parameters: fastseg (binary.vector, minSeg={}, segMedianT=c (0.8, 0.2)). In the case of sparse Strand-seq haplotypes, we set the fastseg parameter minSeg to 20 and, in the case of dense assembly-based haplotypes, we used a larger window of 400 and 500 for Dipcall- and PAV-based variant calls to achieve comparable sensitivity in detecting recombination breakpoints. The regions with a segmentation mean of ≤0.25 are then marked as H1 while regions with a segmentation mean of ≥0.75 are assigned as H2. Regions with a segmentation mean in between these values were deemed to be ambiguous and were excluded. Moreover, we filtered out regions shorter than 500 kb and merged consecutive regions assigned the same haplotype (Code availability).

**Detection of meiotic recombination breakpoints using inheritance vectors.** DeepVariant calls (see the 'Read-based variant calling' section) from HiFi sequencing data from G1, G2 and G3 pedigree members allow us to identify the haplotype of origin for heterozygous loci in G3 and infer the occurrence of a recombination along the chromosome when the haplotype of origin changes between loci. An initial outline of the inheritance vectors was identified by first applying a depth filter to remove variants outside the expected coverage distribution per sample; inheritance was then sketched out using a custom script, requiring a minimum of 10 SNVs supporting a particular haplotype, and manually refined to remove biologically unlikely haplotype blocks, or add additional haplotype blocks, where support existed, and refine haplotype coordinates. Missing recombinations were identified from the occurrence of blocks of pedigree-violating variants, matching the location of assembly-based recombination calls. We developed a hidden Markov model framework to identify the most probable sequence of inheritance vectors from SNV sites using the Viterbi algorithm. For details including the transition/emission probabilities see ref. 18 and the associated GitHub repository (https://github.com/Platinum-Pedigree-Consortium/Platinum-Pedigree-Inheritance).

The transition matrix defines the probability of a given inheritance state transition (recombination). The emission matrix defines the probability that a variant call at a particular locus accurately describes the inheritance state. The values contained within transition and emission matrices were refined to recapitulate the previously identified inheritance vectors, while correctly identifying missing vectors. The Viterbi algorithm identified 539 recombinations, a maternal recombination rate of 1.29 cM per Mb, and a paternal recombination rate of 0.99 cM per Mb. Maternal bias was observed in the pedigree, with 57% of recombinations identified in G3 of maternal origin.

**Merging of meiotic recombination maps.** Meiotic recombination breakpoints reported by different orthogonal technologies and algorithms (see the sections 'Detection of meiotic recombination breakpoints using circular binary segmentation' and 'Detection of meiotic recombination breakpoints using inheritance vectors') were merged

separately for G2 and G3 samples. We started with the G3 recombination map where we used an inheritance-based map as a reference and then looked for support of each reference breakpoint in recombination maps reported based on PAV, Dipcall and Strand-seq (SSQ) phased variants. A recombination breakpoint was supported if for a given sample and homologue an orthogonal technology reported a breakpoint no further than 1 Mb from the reference breakpoint. Any recombination breakpoint that is further apart is reported as unique. We repeated this for the G2 recombination map as well. However, in the case of the G2 recombination map, we used a PAV-based map as a reference. This is because inheritance-based approaches need three generations to map recombination breakpoints in G3. We also report a column called 'best.range', which is the narrowest breakpoint across all orthogonal recombination maps that directly overlaps with a given reference breakpoint. Lastly, we report a 'min.range' column that represents for any given breakpoint a range with the highest coverage across all orthogonal datasets. Merged recombination breakpoints are reported in Supplementary Table 8.

**Meiotic recombination breakpoint enrichment.** We tested enrichment of all ($n = 1,503$) recombination breakpoints detected in G2–G4 with respect to T2T-CHM13 if they cluster towards the ends of the chromosomes depending on parental homologue origin. For this, we counted the number of recombination breakpoints in the last 5% of each chromosome end specifically for maternal and paternal breakpoints. We then shuffle detected recombination breakpoints along each chromosome 1,000 times and redo the counts. For the permutation analysis, we used the R package regioneR[84] (v.1.32.0) and its function permTest with the following parameters:

```
permTest(
A=breakpoints, B=chrEnds.regions,
randomize.function=circularRandomizeRegions,
evaluate.function=numOverlaps,
genome=genome, ntimes=1000,
allow.overlaps=FALSE, per.chromosome=TRUE,
mask=region.mask, count.once=FALSE)
```

**Refinement of meiotic recombination breakpoints using MSA.** Up to this point, all meiotic recombination breakpoints were called using variation detected with respect to a single linear reference (GRCh38 or T2T-CHM13). To alleviate any possible biases introduced by comparison to a single reference genome, we set out to refine detected recombination breakpoints for each inherited homologue (in child) directly in comparison to parental haplotypes from whom the homologue was inherited from. We start with a set of merged T2T-CHM13 reference breakpoints for G3 only by selecting the 'best.range' column (Supplementary Table 8). Then, for each breakpoint, we set a 'lookup' region to 750 kb on each side from the breakpoint boundaries and used the SVbyEye[85] (v.0.99.0) function subsetPafAlignments to subset PAF alignments of a phased assembly to the reference (T2T-CHM13) to a given region. We next extract the FASTA sequence for a given region from the phased assembly. We did this separately for inherited child homologues (recombined) and the corresponding parental haplotypes that belong to a parent from whom the child homologue was inherited from.

Next, we created a multiple sequence alignment (MSA) for three sequences (child-inherited homologue, parental homologue 1 and parental homologue 2) using the R package DECIPHER[86] (v.2.28.0; with the function AlignSeqs). Fasta sequences of which the size differ by more than 100 kb or their nucleotide frequencies differ by more than 10,000 bases are skipped due to increased computational time needed to align such different sequences optimally using DECIPHER. After MSA construction, we selected positions with at least one mismatch and also removed sites where both parental haplotypes carry the same allele. A recombination breakpoint is a region where the

inherited child homologue is partly matching alleles coming from parental homologues 1 and 2. We therefore skipped analysis of MSAs in which a child's alleles are more than 99% identical to a single parental homologue. If this filter is passed, we use the custom R function getAlleleChangepoints (Code availability) to detect changepoints where the child's inherited haplotype switches from matching alleles coming from parental haplotype 1 to alleles coming from parental haplotype 2. Such MSA-specific changepoints are then reported as a new range where a recombination breakpoint probably occurred. Lastly, we attempt to report reference coordinates of such MSA-specific breakpoints by extracting 1 kb long $k$-mers from the breakpoint boundaries and matching such $k$-mers against reference sequence (per chromosome) using R package Biostrings (v.2.70.2) with its function 'matchPattern' and allowing for up to 10 mismatches. A list of refined recombination breakpoints is reported in Supplementary Table 8.

**Detection of allelic gene conversion using phased genome assemblies.** We set out to detect smaller localized changes in parental allele inheritance using a previously defined recombination map of this family. We did this analysis for all G3 samples ($n = 8$) in comparison to G2 parents. For this, we iterated over each child's homologue (in each sample) and compared it to both parental homologues from which the child's homologue was inherited from. We did this by comparing SNV and indel calls obtained from phased genome assemblies between the child and corresponding parent. To consider only reliable variants, we retained only those supported by at least two read-based callers (either DeepVariant-HiFi, Clair3-ONT or dragen-Illumina callset). We further retained only variable sites that are heterozygous in the parent and were also called in the child. After such strict variant filtering, we slide by two consecutive child's variants at a time and compare them to both haplotype 1 and haplotype 2 of the respective parent of origin. For this similarity calculation, we use the custom R function getHaplotypeSimilarity (Code Availability). Then, for each haplotype segment, defined by recombination breakpoints, we report regions where at least two consecutive variants match the opposing parental haplotype in contrast to the expected parental homologue defined by recombination map. We further merge consecutive regions that are ≤5 kb apart. For the list of putative gene conversion events, we retained only regions that have not been reported as problematic by Flagger. We also removed regions that are ≤100 kb from previously defined recombination events and events that overlap centromeric satellite regions and highly identical SDs (≥99% identical). Lastly, we evaluated the list of putative allelic gene conversion events by visual inspection of phased HiFi reads.

## Read-based variant calling

PacBio HiFi data were processed with the human-WGS-WDL available at GitHub (https://github.com/PacificBiosciences/HiFi-human-WGS-WDL/releases/tag/v1.0.3). The pipeline aligns, phases and calls small variants (using DeepVariant[87] v.1.6.0) and SVs (using PBSV v.2.9.0; https://github.com/PacificBiosciences/pbsv). We used the aligned haplotype-tagged HiFi BAMs for all downstream PacBio analysis.

## Clair3

Clair3 (ref. 88) (v.1.0.7) variant calls were made based on the alignments with default models for PacBio HiFi and ONT (ont_guppy5) data, respectively, with phasing and gVCF generation enabled. Variant calling was conducted on each chromosome individually and concatenated into one VCF. gVCFs were then fed into GLNexus[89] with a custom configuration file.

PacBio HiFi

```
run_clair3.sh --bam_fn={input.bam} --sample_name={sample} --ref_
fn={input.ref} --threads=8 --platform=hifi --model_path=/path/to/
models/hifi --output={output.dir} --ctg_name={contig} --enable_
phasing --gvcf
```

ONT

```
run_clair3.sh --bam_fn={input.bam} --sample_name={sample}
--ref_fn={input.ref} --threads=8 --platform=ont --model_path=/path/
to/models/ont_guppy5 --output={output.dir} --ctg_name={contig}
--enable_phasing –gvcf
```

ONT reads for Clair3 calling were aligned with minimap2 (v.2.21) with the following parameters: -L --MD --secondary=no --eqx -x map-ont.

### Generation of truth set of genetic variation using inheritance vectors

We used a previously established framework to define ground truth genetic variation[13]. Our analysis, in contrast to trio-based filtering, uses all four alleles to detect genotyping errors, whereas, in a trio, only two alleles are transmitted and observed. By testing the genotype patterns in the third generation against the phased haplotypes of the first generation (A,B,C,D), we can test for the correct transmission of alleles from the second to third generations. We establish a map of the haplotypes across the third generation (inheritance vector) from which we can adjudicate variant calls against. To test for pedigree consistency, we implemented code that uses the inheritance vector as the expected haplotypes and test the possible genotype configurations within the query VCF file. Using the haplotype structure, we phase the pedigree consistent variants. These functions are implemented as a single binary tool that requires the inheritance vectors and a standard formatted VCF file, for example:

```
concordance -i ceph.grch38.hifi.g3.csv –father NA12877 –mother
NA12878 –vcf input.vcf –prefix pedigree_filtered > info.stdout
```

The pedigree filtering and additional steps to build a small variant truth set are available at GitHub (https://github.com/Platinum-Pedigree-Consortium/Platinum-Pedigree-Inheritance).

### Detection of small de novo variants

Following the parameters outlined previously[10], we called variants in HiFi data aligned to T2T-CHM13 using GATK HaplotypeCaller[90] (v.4.3.0.0) and DeepVariant[87] (v.1.4.0) and naively identified variants unique to each G2 and G3 sample. We separated out SNV and indel calls and applied basic quality filters, such as removing clusters of three or more SNVs in a 1 kb window. We combined this set of variant calls generated by a secondary calling method (https://github.com/Platinum-Pedigree-Consortium/Platinum-Pedigree-Inheritance/blob/main/analyses/Denovo.md) and subjected all calls to the following validation process.

We validated both SNVs and indels by examining them in HiFi, ONT and Illumina read data, excluding reads that failed to reach the mapping quality (59 for long reads, 0 for short reads) thresholds. Reads with high base quality (>20) and low base quality (<20) at the variant site were counted separately. We retained variants that were present in at least two types of sequencing data for the child, and absent from high-base-quality parental reads. For SNV calls, we next examined HiFi data for every sample in the pedigree. We determined an SNV was truly de novo if it was absent from every family member that was not a direct descendant of the de novo sample. Finally, we examined the allele balance of every variant, determined which variants were in TRs and re-evaluated parental read data across all sequencing platforms, removing variants with noisy sequencing data or more than two low-quality parental reads supporting the alternative allele (Supplementary Note 9).

### DNM phasing and postzygotic assignment

To determine the parent of origin for the de novo SNVs, we re-examined the long reads containing the de novo allele. First, we used our initial GATK variant calls to identify informative sites in an 80 kb window around the DNM, selecting any single-nucleotide polymorphisms (SNPs) where one allele could be uniquely assigned to one parent (for example, a site that is homozygous reference in a father and heterozygous in a mother). For every DNM, we evaluated every ONT and HiFi read that aligned to the site of the de novo allele and assigned it to either a paternal or maternal haplotype (if informative SNPs were available) by calculating an inheritance score as outlined previously[10]. DNMs that were exclusively assigned to maternal or paternal haplotypes were successfully phased, whereas DNMs on conflicting haplotypes were excluded from our final callset. Unphased variants were determined to be postzygotic in origin ($n$ = 7) if their allele balance was not significantly different across platforms (by a $\chi^2$ test) and if their combined allele balance was significantly different from 0.5.

Once we assigned every read to a parental haplotype, we counted the number of maternal and paternal reads that had either the reference or alternative allele. We determined that a DNM was germline in origin if it was present on every read from a given parent's haplotype. Conversely, if a DNM was present on only a fraction of reads from a parental haplotype, we determined that it was postzygotic in origin.

### Sex chromosome DNM calling and validation

To identify DNMs on the X chromosome, we applied the same strategy as autosomal variants, with one exception: we used only variant calls generated by GATK. For male individuals, we reran GATK in haploid mode, such that it would only identify one genotype on the X chromosome.

To identify DNMs on the Y chromosome, we aligned male HiFi, ONT and Illumina data to the G1-NA12889 chromosome Y assembly and then called variants using GATK in haploid mode on the aligned HiFi data. We directly compared each male to his father, selecting variants unique to the son. We validated SNVs and indels by examining the father's HiFi, ONT and Illumina data and excluded any variants that were present in the parental reads, applying the same logic that we used for autosomal variants.

### Callable genome and mutation rate calculations

To determine where we were able to identify de novo variation in the genome, we assessed HiFi data for every trio. We first used GATK HaplotypeCaller[90] (v.4.3.0.0) with the option 'ERC BP_RESOLUTION' to generate a genotype call at every site in the genome. Only sites where both parents were genotyped as homozygous reference (0/0) were considered callable, as sites with a parental alternative allele were excluded from our de novo discovery pipeline. We then examined the HiFi reads from a sample and its parents, restricting to only primary alignments with mapping quality of at least 59. For children, we only considered HiFi reads derived from blood, but we considered blood and cell line data for parents. We counted the number of reads with a minimum base quality score of 20 at every site in the genome and then combined this information with our variant calls. A site was deemed to be callable if both parents and the child each had at least one high-quality read with a high-quality base call. We observed an average of 2.67 Gb of accessible sequence across the autosomes (out of 2.90 Gb total, s.d. = 24.9 Mb). For female children, callable X chromosome was determined in the same way, whereas, for the male children, we only considered the mother's HiFi data when examining the X chromosome and the father's HiFi data when examining the Y chromosome. Moreover, male sex chromosomes were not restricted to sites where both parents were genotyped as reference—each parent was allowed to carry an alternative allele.

We calculated the germline autosomal mutation rate for every sample by dividing the number of germline autosomal DNMs by twice the number of base pairs we determined to be callable. For PZMs, we used the same denominator. In female individuals, the amount of callable sex chromosomes was defined as twice the number of callable bases on the X chromosome, and in males it was defined as the sum of the callable bases on the X and Y chromosomes. For each feature-specific mutation rate (such as SDs), we intersected both a sample's de novo SNVs and the sample's callable regions with coordinates of the relevant

feature. We then calculated the mutation rate by dividing the number of SNVs in the region by the amount of callable genomic sequence where alignments could be reliably made.

## Analysis of STRs and VNTRs

Given the challenges associated with assaying mutations in STRs (1–6 bp motifs) and VNTRs (≥7 bp motifs), we applied a targeted HiFi genotyping strategy coupled with validation by transmission and orthogonal sequencing.

**Defining the TR catalogues.** The command trf-mod -s 20 -l 160 {reference.fasta} was used, resulting in a minimum reference locus size of 10 bp and motif sizes of 1 to 2,000 bp (https://github.com/lh3/TRF-mod)[91]. Loci within 50 bp were merged, and then any loci >10,000 bp were discarded. The remaining loci were annotated with tr-solve (https://github.com/trgt-paper/tr-solve) to resolve locus structure in compound loci. Only TRs annotated on Chromosomes 1–22, X and Y were considered (Data availability).

**TR genotyping with TRGT.** TRGT[32] is a software tool for genotyping TR alleles using PacBio HiFi sequencing reads (https://github.com/PacificBiosciences/trgt). Provided with aligned HiFi sequencing reads (in BAM format) and a file that enumerates the genomic locations and motif structures of a collection of TR loci, TRGT will return a VCF file with inferred genotypes at each TR locus. In this analysis, we ran TRGT (v.0.7.0-493ef25) on each member of the CEPH 1463 pedigree using the TR catalogue defined above. TRGT was run using the default parameters:

```
trgt --threads 32 --genome {in_reference} --repeats {in_bed} --reads
{in_bam} --output-prefix {out_prefix} --karyotype {karyotype}`
bcftools sort -m 3072M -Ob -o {out_prefix}.sorted.vcf.gz {out_prefix}.
vcf.gz
bcftools index --threads 4 {out_prefix}.sorted.vcf.gz
samtools sort -@ 8 -o {out_prefix}.spanning.sorted.bam {out_prefix}.
spanning.bam
samtools index -@ 8 {out_prefix}.spanning.sorted.bam
```

**Measuring concordant inheritance of TRs.** To determine the concordant inheritance of TRs, we calculated the possible Manhattan distances derived from all possible combinations of a proband's allele length (AL) from TRGT with both the maternal and paternal AL values. We considered a locus to be concordant if the minimum Manhattan distance from all computed distances was found to be 0, suggesting that a combination of the proband's AL values matched the parental AL values perfectly. By contrast, if the minimum Manhattan distance was greater than 0, suggesting that all combinations of the proband's AL values exhibited some deviation from the parental AL values, we regarded the locus as discordant and recorded it as a potential Mendelian inheritance error. For each TR locus, we calculated the number of concordant trios, the number of MIE trios and the number of trios that had missing values and could not be fully genotyped. Loci with any missing genotypes were excluded when calculating the percent concordance; however, individual complete trios were considered for de novo variant calling below.

**Calling de novo TRs.** We focused de novo TR calling on G3 for several reasons. First, their G2 parents (NA12877 and NA12878) were sequenced to 99 and 109 HiFi sequencing depths, resulting in a far lower chance of parental allelic dropout than samples with more modest sequencing depths. Second, G1 DNA was derived from cell lines, increasing the risk of artefacts when calling DNMs in G2. And finally, DNMs in the two individuals in G3 with sequenced children in our study can be further assessed by transmission.

We used TRGT-denovo[33] (v.0.1.3), a companion tool to TRGT, to enable in-depth analysis of TR DNMs in family trios using HiFi sequencing data (https://github.com/PacificBiosciences/trgt-denovo). TRGT-denovo uses consensus allele sequences and genotyping data generated by TRGT and also incorporates additional evidence from spanning HiFi reads used to predict these allele sequences. In brief, TRGT-denovo extracts and partitions spanning reads from each family member (mother, father and child) to their most likely alleles. Parental spanning reads are realigned to each of the two consensus allele sequences in the child, and alignment scores (which summarize the difference between a parental read and a consensus allele sequence) are computed for each read. At every TR locus, each of the two child alleles is independently considered as a putative de novo candidate. For each child allele, TRGT-denovo reports the presence or absence of evidence for a de novo event, which includes the following: denovo_coverage (the number of reads supporting a unique AL in the child that is absent from the parent's reads); overlap_coverage (the number of reads in the parents supporting an AL that is highly similar to the putative de novo allele); and magnitude of the putative de novo event (expressed as the absolute mean difference of the read alignment scores with de novo coverage relative to the closest parental allele).

**Calculating the size of a de novo TR expansion or contraction.** We measured the sizes of de novo TR alleles with respect to the parental TR allele that most likely experienced a contraction or expansion event. If TRGT-denovo reported a de novo expansion or contraction at a particular locus, we did the following to calculate the size of the event.

Given the ALs reported by TRGT for each member of the trio, we computed the difference in size (which we call a 'diff') between the de novo TR allele in the child and all four TR alleles in the child's parents. For example, if TRGT reported ALs of 100,100 in the father, 50,150 in the mother, and 200,100 in the child, and the allele of length 200 was reported to be de novo in the child, the diffs would be 100,100 in the father and 150,50 in the mother. If we were able to phase the de novo TR allele to a parent of origin, we simply identify the minimum diff among that parent's ALs and treat it as the likely expansion/contraction size. Otherwise, we assume that the smallest diff across all parental ALs represents the likely de novo size.

**De novo filtering.** We applied a series of filters to the candidate TR DNMs (identified by TRGT-denovo) to remove likely false positives. For each de novo allele observed in a child, we required the following (Supplementary Notes 9 and 10):
- HiFi sequencing depth in the child, mother, and father ≥10 reads.
- The candidate de novo AL in the child must be unique: as in ref. 37, we removed candidate de novo TR alleles if (1) the child's de novo AL matched one of the father's ALs and the child's non-de novo AL matched one of the mother's ALs or (2) the child's de novo AL matched one of the mother's ALs and the child's non-de novo AL matched one of the father's ALs.
- The candidate de novo allele must represent an expansion or contraction with respect to the parental allele.
- At least two HiFi reads supporting the candidate de novo allele (denovo_coverage ≥ 2) in the child, and at least 20% of total reads supporting the candidate de novo allele (child_ratio ≥ 0.2).
- Fewer than 5% of parental reads likely supporting the candidate de novo AL in the child.

To calculate TR DNM rates in a given individual, we first calculated the total number of TR loci (among the ~7.8 million loci genotyped using TRGT) that were covered by at least 10 HiFi sequencing reads in each member of the focal individual's trio (that is, the focal individual and both of their parents). We then divided the total count of de novo TR alleles by the total number of callable loci to obtain an overall DNM rate, expressed per locus per generation. Finally, we divided that rate by 2 to produce a mutation rate expressed per locus, per haplotype, per generation. As shown in Fig. 3a, we also estimated DNM rates as a function

of the minimum motif size observed within a locus. For example, a locus with motif structure AT($n$)AGA($n$)T($n$) would have a minimum motif size of 1. We counted the number of TR DNMs that occurred at loci with a minimum motif size of $N$ and divided that count by the total number of TR loci with a minimum motif size of $N$ that passed filtering thresholds. We then divided that rate by 2 to produce a mutation rate per locus, per haplotype, per generation. When calculating STR, VNTR and complex mutation rates, we defined STR loci as loci at which all constituent motifs were between 1 and 6 bp; we defined VNTR loci as loci at which all motifs were larger than 6 bp; and we defined complex loci as loci at which there were both STR (1–6 bp) and VNTR (≥7 bp) motifs. For example, both an A($n$) locus and an AT($n$)AGA($n$)T($n$) locus would be classified as STRs, as they both purely contain STR motifs.

Previous studies usually measured STR mutation rates at loci that are polymorphic within the cohort of interest. To generate mutation rate estimates that are more consistent with these previous studies, we also calculated the number of STR loci that were polymorphic within the CEPH 1463 pedigree. Loci were defined as polymorphic if at least two unique ALs were observed among the CEPH 1463 individuals at a given TR locus. We note that this definition of polymorphic STRs is sensitive to both the size of the cohort and the sequencing technology used to genotype STRs. As discussed in previous studies[37], the number of polymorphic loci is proportional to the size of the cohort. Moreover, by defining loci as polymorphic if we observed more than one unique AL across the cohort, we may erroneously classify loci as polymorphic if HiFi sequencing reads exhibited a substantial amount of stutter at those loci, producing variable estimates of STR ALs across individuals. In total, 1,096,430 STRs were polymorphic within the cohort. To calculate mutation rates in each G3 individual, we applied the same coverage quality thresholds as described above.

**Phasing of TRs.** The STRs genotyped by TRGT were phased using HiPhase[92] (v.1.0.0-f1bc7a8). We followed HiPhase's guidelines for jointly phasing small variants, SVs and TRs by inputting the relevant VCF files from DeepVariant, PBSV and TRGT into HiPhase, resulting in three phased VCF files for each analysed sample. We also activated global realignment through the --global-realignment-cputime parameter to improve allele assignment accuracy. Note that HiPhase specifically excludes variants that fall entirely within genotyped STRs from the phasing process. This is motivated because STRs often encompass numerous smaller variants.

```
hiphase --threads 32 --io-threads 4 --sample-name {sample_id}
--vcf {in_vcf_deepvariant} --vcf {in_vcf_pbsv} --vcf {in_vcf_trgt}
--output-vcf {out_vcf_deepvariant} --output-vcf {out_vcf_pbsv}
--output-vcf{out_vcf_trgt} --bam {in_bam} --reference {in_refer-
ence} --summary-file {out_summary} --blocks-file {out_blocks}
--global-realignment-cputime 300
```

**Parent-of-origin determination.** We used the phased genotypes inferred by HiPhase to determine the likely parent of origin for de novo TR expansions and contractions. For each phased de novo allele that we observed in a child, we examined all informative SNVs in that child's parents ±500 kb from the de novo allele. We defined informative sites using the following criteria: sites must be biallelic SNVs; total read depth in the mother, father and child must be at least 10 reads; Phred-scaled genotype quality in the mother, father and child must be at least 20; the child's genotype must be heterozygous; and the parents' genotypes must not be identical-by-state. Using the child's phased SNV VCF, we then determine whether the child's REF or ALT allele at the informative site was inherited from either the mother or father. For example, if the mother's genotype is 0/0, the father's genotype is 0/1 (note that the parental genotypes need not be phased), and the child's genotype is 1|0, we know that the child's first haplotype was inherited from the

father and the second haplotype was inherited from the mother. We repeat this process for all informative sites within the ±500 kb interval. We then find the $N$ informative sites that are (1) closest to the de novo TR allele (either upstream or downstream) while (2) supporting a consistent inheritance pattern in the child (that is, all support the same parent of origin for the child's two haplotypes) and (3) all reside within the same HiPhase phase block (defined using the PS tag in the HiPhase output VCF). Finally, we use the phased TR VCF produced by HiPhase to check whether the de novo allele was phased to either the first or second haplotype in the child. We then confirm that the de novo allele shares the same PS tag as the informative sites identified above and use the $N$ informative sites to determine whether the haplotype to which the de novo allele was phased was probably inherited from either the mother or the father.

**Measuring concordance with orthogonal sequencing technology.** At each candidate de novo TR allele, we calculated concordance between the de novo ALs estimated by TRGT and the ALs supported by Element, ONT or HiFi reads. We restricted our concordance analyses to autosomal TR loci with a single expansion or contraction (that is, we did not analyse 'complex' TR loci containing multiple unique expansions and/or contractions).

TRGT reports two AL estimates for every member of a trio at an autosomal TR locus, and TRGT-denovo assigns one of these two ALs to be the de novo AL in the child. At each TR locus, we calculated the difference between the length of the locus in the reference genome (in base pairs) and each of the two ALs in a given individual. We refer to the difference between the TRGT AL and the reference locus size as the relative AL. We then queried BAM files containing Element, Illumina, ONT or PacBio HiFi reads at each TR locus. Using the pysam library (https://github.com/pysam-developers/pysam), we iterated over all reads that completely spanned the TR locus and had a mapping quality of 60. To estimate the AL of a TR expansion/contraction in a read with respect to the reference genome, we counted the number of nucleotides associated with every CIGAR operation that overlapped the TR locus. For example, an Element read might have the following CIGAR string: 100M2D10M6I32M. For each of the CIGAR operations that overlap the TR locus, we increment a counter by OP * BP, where OP equals 0 for 'match' CIGAR operations, 1 for 'insertion' operations, and -1 for 'deletion' operations, and BP equals the number of base pairs associated with the given CIGAR operation. Thus, at each TR locus, we generated a distribution of net CIGAR operations in each member of the trio.

We used these net CIGAR operations to validate candidate de novo TR alleles in each child. For each de novo TR allele, we calculated the number of Element reads in the child that supported the de novo AL estimated by TRGT (allowing the Element reads to support the de novo AL ± 1 bp). We then calculated the number of Element reads in that child's parents supporting the de novo AL (also allowing for off-by-one errors). If at least one Element read supported the de novo TR AL in the child, and zero Element reads supported the de novo TR AL in both parents, we considered the de novo TR to be validated.

**Validating recurrent TR DNMs.** To assemble a confident list of candidate recurrent de novo TR alleles, we first assembled a list of TR loci where two or more CEPH 1463 individuals (in either G2, G3 or G4) harboured evidence for a de novo TR allele. For each candidate locus, we then required that all members of the CEPH 1463 pedigree were genotyped for a TR allele at the locus and had at least 10 aligned HiFi reads at the locus. These filters produced a list of 49 candidate loci where we observed evidence of either intragenerational or intergenerational recurrence. We visually inspected HiFi read evidence using the Integrated Genomics Viewer (IGV)[93], as well as bespoke plots of HiFi CIGAR operations, at each locus to determine whether the candidate de novo TR alleles seemed plausible.

## Detection and filtering of de novo SVs
We attempted to obtain putative de novo SVs from three different sources. The first one is based on reporting de novo SVs from read-based callsets (PBSV (v.2.9.0), Sniffles[94] (v.0.12.0), Sawfish[95] (v.2.2)). The second reports putative de novo SVs from variants called in phased genome assemblies. The last used pangenome graphs constructed from phased genome assemblies to report de novo SVs.

**Assembly-based detection of de novo SVs.**
(1) SVPOP[8] (v.3.4.0) (https://github.com/EichlerLab/svpop) was used to produce a merged PAV callset across all samples. It merges a single source (single SV caller) across multiple samples. The merge definition used was: nr::ro:szro:exact:match. The samples were provided in this order (G1–G2–G3): NA12889, NA12890, NA12891, NA12892, NA12877, NA12878, NA12879, NA12881, NA12882, NA12883, NA12884, NA12885, NA12886, NA12887.
(2) For each sample in G3, we selected variants unique to that sample alone.
(3) To compare variant calls against the previous generation, SVPOP was used again to do a PBSV/PAV intersection. This involved intersecting the PAV calls for G3 with the PBSV calls for G2, comparing each sample in G3 against each sample in G2.
(4) The callable BED files from PAV, intersections with G2's PBSV calls, and the list of putative de novo calls went into our validation pipeline.
(5) The pipeline (1) checks if the putative de novo variant was called by PBSV in either parent. (2) Checks if the putative de novo variant is seen in HiFi reads in either parent by running subseq (https://github.com/EichlerLab/subseq). (3) Checks if the variant was in a callable region in either parent. (4) Performs an MSA using DECIPHER of the two haplotypes of the sample, and both parents, in the location of the SV with 1,000 bp flank on either side.

**Pangenome graph detection of de novo SVs.** Verkko assemblies were partitioned by chromosome by mapping them against the GRCh38, T2T-CHM13 and HG002 (v.1.0.1) human reference genomes using WFMASH (v.0.13.1-251f4e1) pangenome aligner. On each set of contigs, we applied PGGB (v0.6.0-87510bc) to build chromosome-level unbiased pangenome variation graphs[96] with the following parameters: -s 20k -p 95 -k 47 -V chm13:100000, grch38:100000. We used the Variation graph toolkit[97] (v.1.40.0) to call variants from the graphs with respect to both the T2T-CHM13 and GRCh38 reference genomes. Variants were then decomposed by applying VCFBUB (v.0.1.0-26a1f0c) to retain those found in top-level bubbles that are anchored on the genome used as reference, and VCFWAVE (v.1.0.3) to homogenize SV representation across samples. Subsequently, raw VCF files were used as an input for pedigree-based filtering of putative de novo SVs.

**De novo SV filtering in SV callsets (PGGB, PAV, PBSV, Sniffles, Sawfish).** Filtering of de novo SVs was done using BCFtools (v.1.17) +fill-tags followed by filtering the joint-called VCF for singleton-derived alleles at sites where all samples had a genotype call. By considering all G2/G3 family members (not just trios), we increased de novo SV specificity. We used the command line:

```
bcftools view -i 'INFO/AC = 1' {VCF FILE} | bcftools +fill-tags -- -t 'all,F_MISSING' | bcftools view -i 'F_MISSING = 0.0' --max-alleles 2 | bcftools view --samples {SAMPLE} | bcftools +fill-tags | bcftools view -i 'INFO/AC = 1' | bcftools view -i '(ILEN < -49 || ILEN > 49)' | bcftools view -i 'QUAL > 49' | vcf2tsv
```

All candidate de novo SVs collected across all regions of the genomes were further evaluated using phased genome assemblies and long-read alignments. Further details are provided in Supplementary Note 10.

**Extracting donor site of de novo SVA insertion.** We first extracted an inserted SVA element in the de novo Verkko assembly of NA12887 (maternal haplotype, haplotype 1). Next, we used minimap2 (ref. 80) (v.2.24) to align this ~3.4-kb-long piece of DNA to both maternal and paternal Verkko assemblies using the parameters reported below:

```
minimap2 -x asm20 -c --eqx --secondary=yes {assembly.fasta} {sva.fasta} > {output.paf}
```

With these parameters we reported all locations of this DNA segment. We defined a putative donor site as an alignment position in maternal haplotype that has nearly perfect match with SVA de novo insertion.

## Analysis of centromeric regions
To identify completely and accurately assembled centromeres from each genome assembly, we first aligned the genome assemblies generated via Verkko[16] or hifiasm (UL)[17] to the T2T-CHM13 reference genome[1] using minimap2 (ref. 80) and the following parameters: -a --eqx -x asm20 -s 5000 -l 10G -t {threads}. We then filtered the whole-genome alignments to only those contigs that aligned to the centromeres in the T2T-CHM13 reference genome. We checked whether these centromeric contigs spanned the centromeres by checking to see whether they contained sequence from the p- and the q-arms in the regions directly adjacent to the centromere. We then validated the assembly of the centromeric regions by aligning native PacBio HiFi data from the same source genome to each whole-genome assembly using pbmm2 (v.1.1.0; https://github.com/PacificBiosciences/pbmm2) and the following command: align --log-level DEBUG --preset SUBREAD --min-length 5000 -j {threads}, and next assessed the assemblies for uniform read depth across the centromeric regions via NucFreq[77] (v.0.1). We also aligned native ONT data >30 kb in length from the same source genome to each whole-genome assembly using minimap2 (v.2.28) and assessed the assemblies for uniform read depth across the centromeric regions using IGV browser[93].

To identify de novo SVs and SNVs within each centromeric region, we first aligned each child's genome assembly to the relevant parent's genome assembly using minimap2 and the following parameters: -a --eqx -x asm20 -s 5000 -l 10G -t {threads}. We then used the resulting PAF file to identify de novo SVs and SNVs using SVbyEye[85] (v.0.99.0), filtering our results to only those centromeres that were completely and accurately assembled. We checked each SV and SNV call with Nuc-Freq, Flagger[9] and native ONT data to ensure that the underlying data supported each call. Further details are provided in Supplementary Notes 9 and 10.

## Analysis of telomeric regions
We processed all G1, G2 and G3 assemblies with Tandem Repeats Finder (TRF)[91] to determine the existence of the canonical telomeric repeat (p-arm, CCCTAA; q-arm, TTAGGG) within the distal regions of each assembled contig; TRF (v.4.09.1) was run with parameters: '2 7 7 80 10 50 10 -d -h-ngs', recommended for young (in this context, non-deteriorated) repeats as implemented in RepeatMasker (v.4.1.6). The assembled contigs, in turn, were aligned to the T2T-CHM13 reference with minimap2 (ref. 80) (v.2.24) using the asm20 preset to establish the identities of each sequence (that is, whether a given contig represented the whole reference chromosome or a part of it, and whether it should be reverse-complemented to represent it canonically). With identities established, TRF annotations were crawled from the outside in (from the 5′ end on p-arms and from the 3′ end on q-arms, with respect to reverse complementarity as reported by minimap2) until the canonical repeat was encountered; incidences of non-canonical interspersed repeats were also retained.

Moreover, PacBio HiFi reads were mapped to the contigs to assess by how many HiFi reads each region of each assembly was supported (coverage depth); distal regions supported by fewer than five HiFi reads were masked. Of the non-acrocentric chromosome ends across all

G1, G2 and G3 samples, 74.2% of the Verkko assemblies (893 out of the possible 1,204 across all participants and haplotypes) were found to terminate in a canonical telomeric repeat (either spanning from the very start or end of the contig, or immediately adjacent to the region masked due to low coverage) with the median length of such repeats being 5,608 bp (Supplementary Table 3). Moreover, out of the T2T-CHM13 chromosomes for which both p and q telomeric ends were recovered, 64.6% (221 out of 342) were represented each by a single assembled contig spanning from the p telomere to the q telomere.

The G4 hifiasm assemblies were processed in the same fashion; however, only 56.8% of the telomeric regions (342 out of the possible 602) were recovered (Supplementary Fig. 3) with a median length of the canonical repeat being 4,674 bp (Supplementary Table 3; same as for G1–G3), and the contiguity was markedly worse: only one chromosome (chromosome 9 in haplotype 1 of individual G4-200101) was verifiably spanned by a single contig (h1tg000017l).

### CpG methylation analysis

To determine the CpG methylation status of each centromere, we first base called raw ONT data with Guppy (https://community. nanoporetech.com; v.6.5.7) using the sup-prom model and the dna_r9.4.1_450bps_modbases_5hmc_5mc_cg_sup_prom.cfg config file. Next, we aligned the ONT data from each sample to the respective genome assembly using minimap2 (ref. 80) (v.2.28) with the following parameters: -ax lr:hq -y -t 4 -I 8 g. We converted the resulting BAM file to a bedMethyl file using modbam2bed (https://github.com/epi2me-labs/modbam2bed) and the following parameters: -e -m 5mC --cpg -t {threads} {input.bam} > {output.bed}. Next, we converted the bed-Methyl file into a bedGraph using the following command: awk 'BEGIN {OFS="\t"}; {print $1, $2, $3, $11}' {input.bed} | grep -v "nan" | sort -k1,1 -k2,2n >{output.bedgraph} and subsequently converted the bedGraph into a bigwig using bedGraphToBigWig (https://www.encodeproject. org/software/bedgraphtobigwig/) and then visualized the bigwig file using Integrative Genomics Viewer[93,98] (v.2.16.0). To determine the size of a hypomethylated region (termed the CDR[2,39]) in each centromere, we used CDR-Finder (https://github.com/arozanski97/CDR-Finder), which first bins the bedGraph into 5 kb windows, computes the median CpG methylation frequency within windows containing α-satellite (as determined by RepeatMasker[99] (v.4.1.0)), selects bins that have a lower CpG methylation frequency than the median frequency in the region, merges consecutive bins into a larger bin, filters for merged bins >50 kb and reports the location of these bins.

### Y-chromosomal analysis

**Construction and dating of Y phylogeny.** The construction and dating of Y-chromosomal phylogeny for 58 total samples, combining the 14 pedigree males from the current study with 44 individuals, for which long-read-based Y assemblies have previously been published, was done as described previously in detail[52]. In brief, all sites were called from the Illumina high-coverage data[14] of the 14 pedigree males using the approximately 10.4 Mb of Y-chromosomal sequence previously defined as accessible to SRS[100]. BCFtools[101,102] (v.1.16) was used with a minimum base quality 20, mapping quality 20 and ploidy 1. SNVs within 5 bp of an indel call (SnpGap) and all indels were removed, followed by filtering all calls for a minimum read depth of 3 and a requirement of ≥85% of reads covering the position to support the called genotype. The VCF was merged with a similarly filtered VCF from ref. 52 for the 44 individuals using BCFtools, and then sites with ≥5% of missing calls, that is, missing in more than 3 out of 58 samples, were removed using VCFtools[103] (v.0.1.16). After filtering, a total of 10,404,104 sites remained, including 13,443 variant sites.

The Y haplogroups of each sample were predicted as previously described[104] and correspond to the International Society of Genetic Genealogy nomenclature (ISOGG; https://isogg.org; v.15.73). A coalescence-based method implemented in BEAST[105] (v.1.10.4) was used to estimate the ages of internal nodes. RAxML[106] (v.8.2.10) with the GTRGAMMA substitution model was used to construct a starting maximum-likelihood phylogenetic tree for BEAST. Markov-chain Monte Carlo samples were based on 200 million iterations, logging every 1,000 iterations, with the first 10% of iterations discarded as a burn-in. A constant-sized coalescent tree prior, the GTR substitution model, accounting for site heterogeneity (gamma), and a strict clock with a normal distribution based on the 95% CI of the substitution rate ($0.76 \times 10^{-9}$ (95% CI = $0.67 \times 10^{-9}$–$0.86 \times 10^{-9}$) single-nucleotide mutations per base pair per year) was used[107]. A summary tree was produced using Tree-Annotator (v.1.10.4) and visualized using the FigTree software (v.1.4.4).

**Identification of sex-chromosome contigs.** Detailed analysis of Y-chromosomal DNMs focused on seven male individuals (R1b1a-Z302 Y haplogroup, G1-NA12889, G2-NA12877, G3-NA12882, G3-NA12883, G3-NA12884 and G3-NA12886) for whom phased Verkko assemblies were generated. Contigs containing X- and Y-chromosomal sequences were identified and extracted from the whole-genome assemblies as previously described[52]. Moreover, the pseudoautosomal regions from the G1 grandmother NA12890 and G2 mother NA12878 genome assemblies were identified by aligning the respective sequences from the T2T-CHM13 reference genome to these assemblies using minimap2 (ref. 80) (v.2.26).

**Annotation of Y-chromosomal subregions.** The annotation of Y-chromosomal subregions of the Verkko assemblies was performed using both the GRCh38 and T2T-CHM13 Y reference sequences as previously described[52]. The centromeric α-satellite repeats for the purpose of Y subregion annotation were identified using Repeat-Masker[99] (v.4.1.2-p1) with the default parameters. The Yq12 repeat annotations were generated using HMMER[108] (v.3.3.2dev) with published *DYZ1*, *DYZ2*, *DYZ18*, 2k7bp and 3k1bp sequences[52], followed by manual checking of repeat unit orientation and distance from each other. Dot plots to compare Y-chromosomal sequences were generated using Gepard[109] (v.2.0).

**Detection and validation of DNMs.** Human Y chromosomes vary extensively in the size and composition of repetitive regions[52], including the T2T-CHM13 Y (haplogroup J1a-L816) and the R1b1a-Z302 haplogroup Y chromosomes carried by the seven pedigree males (Supplementary Note 6). For this reason, the Y assembly of the G1 grandfather NA12889 was used as a reference for DNM detection. The DNMs were called from the Y assemblies of five G2 (NA12877) and G3 (NA12882, NA12883, NA12884 and NA12886) males using Dipcall[82] (v.0.3) with the default parameters recommended for male samples. Variants were identified from the MSY only, that is, the pseudoauto-somal regions were excluded from this analysis. All identified variants were filtered as follows: any variant calls overlapping with regions flagged by Flagger or NucFreq in either reference or query assembly were filtered out.

For SNVs, the final filtered calls were supported by 100% of HiFi reads (that is, no reads supported the reference allele in offspring or alternative allele in the father) and ONT reads mapped to both the reference and each individual assembly were checked for support.

For indels (≤50 bp), homopolymer tracts were excluded from the analysis, while the rest of the calls were validated using the read data (HiFi, ONT, Illumina) as follows. Individual reads mapped to the reference (G1 NA12889 Y assembly) and covering the indel call plus 150 bp of flanking sequence were extracted from all samples using subseq (https://github.com/EichlerLab/subseq), followed by alignment using MAFFT[110,111] (v.7.508) with the default parameters. All alignments were manually checked and any calls where the HiFi data had two or more reads supporting a reference allele and one or more reads support-ing an alternative allele were removed. All final SNV and indel calls

were additionally supported (if unique mapping to the region was possible) by both Illumina and Element read data mapped to the reference.

For all SV calls, HiFi read depth for reference and alternative alleles were visualized and SVs in regions showing high levels of read depth variation coinciding with clusters of SNVs with >10% of reads supporting an alternative allele removed. HiFi and ONT reads mapped to both the reference and individual assemblies were checked for support.

For all variants, concordance with the expected transmission through generations was confirmed. Moreover, the HiFi data available for three G4 male individuals (200101, 200102 and 200105) were checked for support of the identified variants.

**Y-chromosomal DNM rate calculation.** The assembly-based DNM rates were calculated for each of the five male individuals based on the accessible regions of each individual Y assembly (that is, any regions flagged by Flagger and/or NucFreq were removed).

### Mobile element analysis

Mobile element analysis was performed on PacBio HiFi reads using xTea[112] (v.0.1.9). Potential non-reference mobile element insertions (MEI) identified with xTea were visualized using IGV to ensure that the insertions were identifiable in the sequencing reads and to determine whether any of these events were de novo. Using BEDTools[113], we intersected the non-reference insertions with introns, exons, 5′-UTRs and 3′-UTRs from T2T-CHM13. To identify potential source elements of the non-reference LINE-1 insertions, we used BLAT[114] to find the best matching insertion in the T2T-CHM13 reference genome. If there were multiple matches in the reference genome that had the same score, a source element was not called. MEI sequences representing known Alu, L1 and SVA subclasses were obtained from previous work[115], Dfam[116] and UCSC Genome Browser[74]. Reference and novel sequences for each MEI class were combined into class-specific files. Sequences were oriented to plus-strand. Highly truncated sequences were removed. MEI sequences were aligned using the MUSCLE[117] (v.3.8.31) aligner. Pairwise distances among MEI sequences were calculated using a Kimera two-parameter method and then converted to correlations. Principal components were obtained by eigenvalue decomposition of the pairwise correlation matrix. The first three principal components were plotted to visualize the relationships among the non-reference MEIs and the known MEI subfamily sequences.

### Reporting summary

Further information on research design is available in the Nature Portfolio Reporting Summary linked to this article.

### Data availability

All underlying data from 28 members of the family are available as part of the AWS Open Data program, European Nucleotide Archive (ENA) or dbGaP. Variant calls, mapped sequencing data and assemblies for 23 family members (G1-GM12889, G1-GM12890, G1-GM12891, G1-GM12892, G2-GM12877, G2-GM12878, G3-GM12879, G3-GM12881, G3-GM12882, G3-GM12885, G3-GM12886, G3-200080-spouse, G4-200081, G4-200082, G4-200084, G4-200085, G4-200086, G4-200087, G3-200100-spouse, G4-200101, G4-200102, G4-200104 and G4-200106) who provided consent for their data to be publicly accessible similar to the 1000 Genomes Project samples to allow for development of new technologies, study of human variation, research on the biology of DNA and study of health and disease are available via the AWS Open Data program (s3://platinum-pedigree-data/) as well as the European Nucleotide Archive (BioProject: PRJEB86317). Specific details on how to access the data are provided at GitHub (https://github.com/Platinum-Pedigree-Consortium/Platinum-Pedigree-Datasets). Mapped sequencing data and assemblies for five family members

(G3-NA12883, G3-NA12884, G3-NA12887, G4-200103 and G4-200105) who did not consent for open access are available at dbGaP (phs003793.v1.p1; Platinum Pedigree Consortium LRS). These also include variant calls for the whole family (28 members). The TR catalogues are available at Zenodo (https://doi.org/10.5281/zenodo.13178746). The Y-chromosomal assembly for a closely related R1b haplogroup sample HG00731 was downloaded from the Human Genome Structural Variation Consortium IGSR site (https://ftp.1000genomes.ebi.ac.uk/vol1/ftp/data_collections/HGSVC3/working/20230927_verkko_batch2/assemblies/HG00731/). Reference genomes and their annotations used in this study are listed in Supplementary Table 14.

### Code availability

Custom code and pipelines used in this study are publicly available at GitHub (https://github.com/orgs/Platinum-Pedigree-Consortium/repositories).

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

**Acknowledgements** We thank S. Jankauskiene, M. Lee and T. Nguyen for technical assistance with preparation and sequencing of Strand-seq libraries; C. Steidl for use of the NextSeq550 sequence platform; and T. Brown for edits in the preparation of this manuscript. Library pools were also sequenced on the Element AVITI at the University of California Davis DNA Technologies Core. The following cell lines were obtained from the NIGMS Human Genetic Cell Repository at the Coriell Institute for Medical Research: GM12889, GM12890, GM12891, GM12892, GM12877, GM12878, GM12879, GM12881, GM12882, GM12883, GM12884, GM12885, GM12886 and GM12887. This research was supported in part by funding from the National Institutes of Health (NIH) grants R01HG002385, R01HG010169 and R01MH101221 (to E.E.E.) and U24HG007497 (to E.E.E. and C. Lee) as well as support by the Simons Foundation (SFARI, 810018EE to E.E.E.). E.E.E. is an investigator of the Howard Hughes Medical Institute (HHMI). P.M.L. was funded in part by a program project grant (1074) from the Terry Fox Research Foundation and a research grant (159787) from the Canadian Institutes of Health Research. This research was further supported by funding to H.D. by 5K99HG012796-02; to C.J.S. by R00HG011657; and to L.B.J. and W.S.W. by NIH R35GM118335. G.A.L. was supported by NIH GM147352. This Article is subject to HHMI's Open Access to Publications policy. HHMI laboratory heads have previously granted a non-exclusive CC BY 4.0 license to the public and a sublicensable license to HHMI in their research articles. Pursuant to those licences, the author-accepted manuscript of this article can be made freely available under a CC BY 4.0 license immediately on publication.

**Author contributions** Conceptualization: D.P., L.B.J. and E.E.E. Chromosome Y analysis: P.H., P.E. and C. Lee. DNM analysis: M.D.N., D.P., H.D., T.A.S., P.H., G.A.L. and Z.N.K. Generation of de novo assemblies and validation: N.K., W.T.H. and D.P. Data analysis support: N.K., W.T.H., W.J.R., J.L., T.Y.L., V.C.T.H. and H.J. Centromere analysis: D.P., K.K.O. and G.A.L. Telomere analysis: K.G. and C.E.M. Meiotic recombination analysis: D.P., C.N., Z.N.K., M.E.G. and A.G. TR analysis: H.D., T.A.S., T.M., E.D., T.J.N., M.E.G. and A.R.Q. MEI analysis: C.J.S. and W.S.W. Generated sequencing data: K.M.M., K.H., D.D.C., Y.W., J.K., G.H.G., C.F. and C. Lambert. Short-read callset generation: B.S.P., H.C.H., S.M. and J.D.S. Developed main figures: D.P., H.D., T.A.S., T.M., G.A.L., P.H., M.D.N. and E.E.E. Manuscript writing: D.P., H.D., T.A.S., G.A.L., P.H., M.D.N., Z.N.K. and E.E.E. Supervised experiments and analyses: S.L., C.E.M., E.G., P.M.L., D.W.N., L.B.J., A.R.Q., M.A.E. and E.E.E.

**Competing interests** E.E.E. is a scientific advisory board member of Variant Bio. C. Lee is a scientific advisory board member of Nabsys and Genome Insight. D.P. has previously disclosed a patent application (no. EP19169090) relevant to Strand-seq. Z.N.K., C.N., E.D., C.F., C. Lambert, T.M., W.J.R. and M.A.E. are employees and shareholders of PacBio. Z.N.K. is a private shareholder in Phase Genomics. The other authors declare no competing interests.

**Additional information**
**Correspondence and requests for materials** should be addressed to Evan E. Eichler.

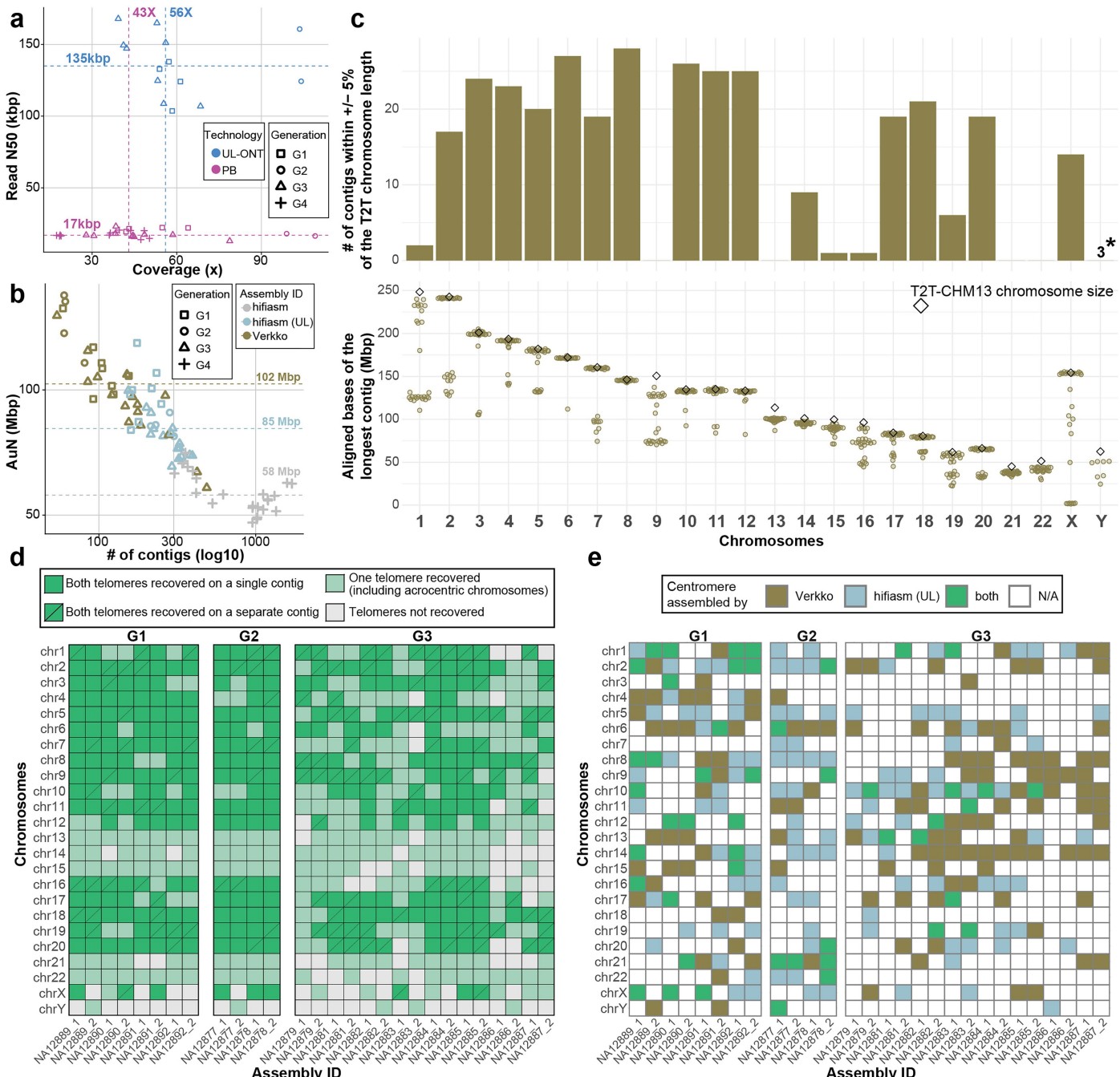

**Extended Data Fig. 1 | Long-read sequencing and assembly contiguity.**
**a**) Scatterplot of sequence read depth and read length N50 for ONT (blue) and PacBio (PB; magenta) with median coverage (dashed line) and different generations indicated (point shape). **b**) Scatterplot of the assembly contiguity measured in AuN values for Verkko (brown), hifiasm (UL) (light blue), and hifiasm (light grey) assemblies of G1-G4. Note: G4 samples were assembled using PacBio HiFi data (hifiasm) only; hifiasm (UL) refers to hifiasm assemblies integrating both PacBio HiFi and ONT data. **c**) Top: Total number of Verkko contigs whose maximum aligned bases are within +/−5% of the total T2T-CHM13 chromosome length. *Due to substantial size differences between the T2T-CHM13 Y (haplogroup J1a-L816) and the Y chromosome of this pedigree

(haplogroup R1b1a-Z302), three contigs are shown that span the entire male-specific Y region without breaks (i.e., excluding the pseudoautosomal regions). Bottom: Each dot represents a single Verkko contig with the highest number of aligned bases in a given chromosome. **d**) Chromosomes containing complete telomeres and being spanned by a single contig are annotated as solid squares. In instances where the p- and q-arms are not continuously assembled and for acrocentric chromosomes, we plot diagonally divided and colour-coded triangles. **e**) Evaluation of centromere completeness across G1-G3 assemblies and across all chromosomes. We mark centromeres assembled by Verkko (brown), hifiasm (UL) (light blue), or both (green).

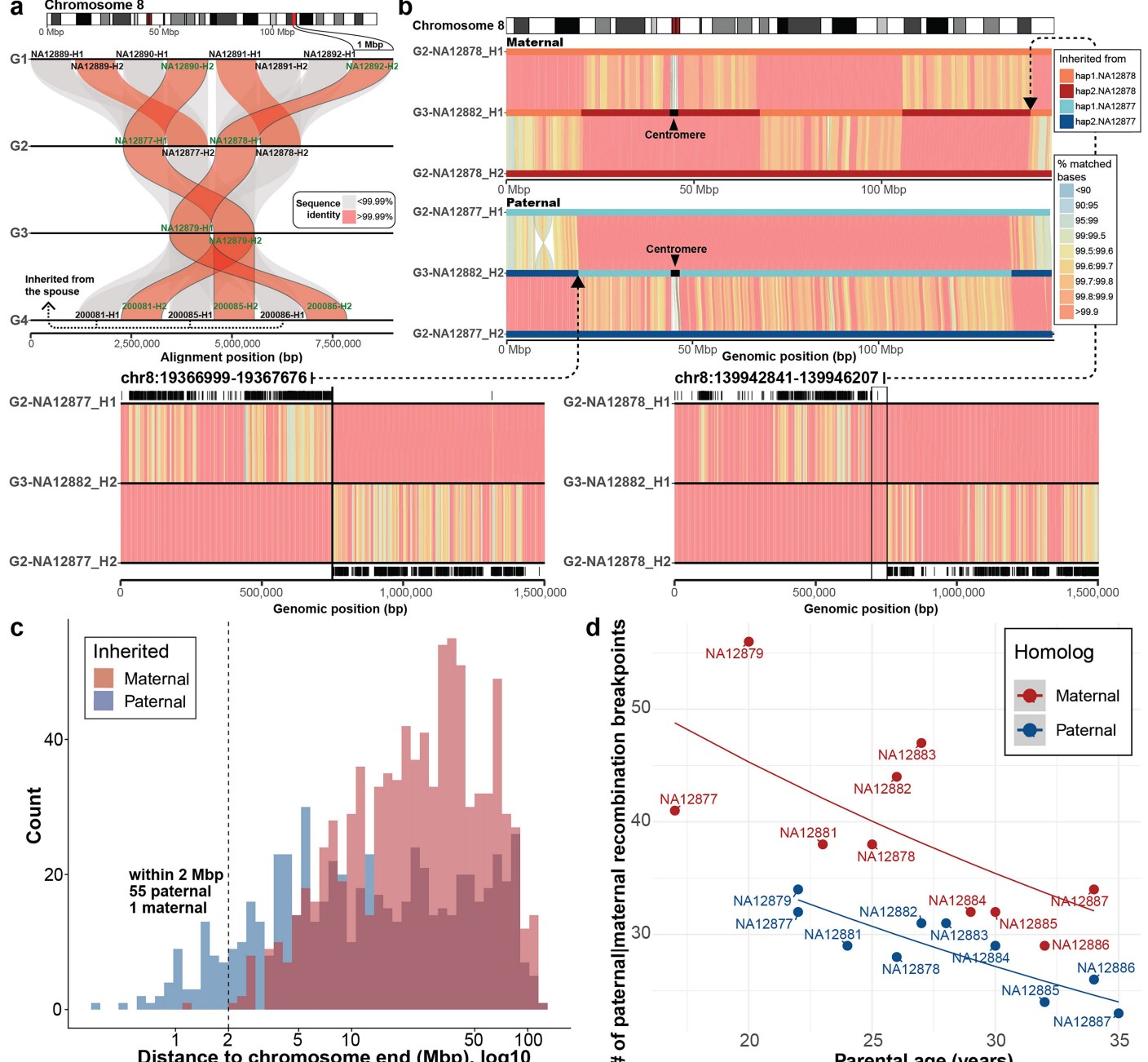

**Extended Data Fig. 2 | Recombination breakpoint map of CEPH1463.**
**a)** Depiction of intergenerational (G1- > G4) inheritance of a 1 Mbp assembled contig. Alignments transmitted between generations that are >99.99% identical (red) are contrasted with non-transmitted with lower sequence identity (grey). **b)** T2T recombination between child and parental haplotypes for Chromosome 8. Alignments between the parental and child haplotypes are binned into 500 kbp long bins and coloured based on the percentage of matched bases. Inherited maternal (shades of red) and paternal (shades of blue) segments are marked on top. Dashed arrows show zoom-in of the two recombination breakpoints that differ in size of the region of homology at the

recombination breakpoint. Black tick marks show positions of mismatches between parental and child haplotypes. **c)** Distribution of distances of maternal (red) and paternal (blue) recombination breakpoints (G2-G4) to chromosome ends with respect to T2T-CHM13 (histogram bin size: 50). **d)** Significant association between the number of recombination breaks (y-axis) and parental age (x-axis) shown separately for maternal (red) and paternal (blue) recombination breakpoints (G2-G3) detected with respect to T2T-CHM13. Regression lines were fitted using Poisson GLM with a log link (p = 2.02 × 10⁻³, 7.88 × 10⁻⁴ for parental age and sex effects, respectively).

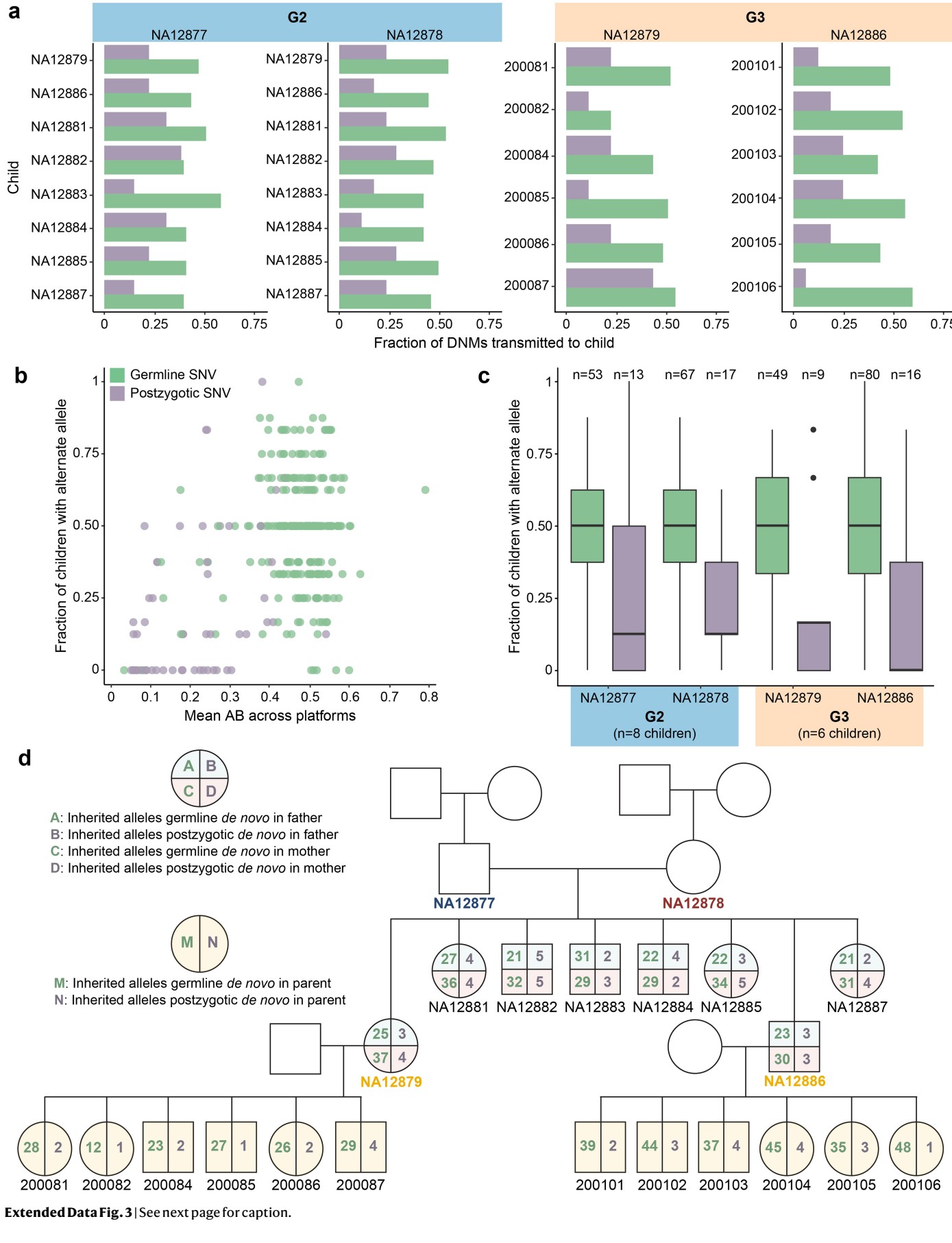

**Extended Data Fig. 3** | See next page for caption.

**Extended Data Fig. 3 | Number of germline and postzygotic SNVs transmitted to children. a**) The fraction of a parent's germline SNVs (green, DNMs) and postzygotic SNVs (purple, PZMs) transferred to each child. **b**) The mean allele balance (AB) of DNMs (n = 249) and PZMs (n = 55) across HiFi, Illumina, and ONT data plotted against the fraction of children who inherited a variant are significantly correlated for DNMs (two-sided t-test, p = 0.0084) and PZMs (p = 0.00021). Half of PZMs with AB < 0.25 are transmitted to at least one child (n = 18/36). **c**) On average, DNMs are transmitted to 50% of children, while PZMs are transmitted to less than 25% of children. Boxes represent IQR including median line; whiskers extend to 25% − 1.5 × IQR and 75% + 1.5 × IQR, outliers are shown as dots. **d**) Number of DNMs and PZMs transmitted to each child in the pedigree.

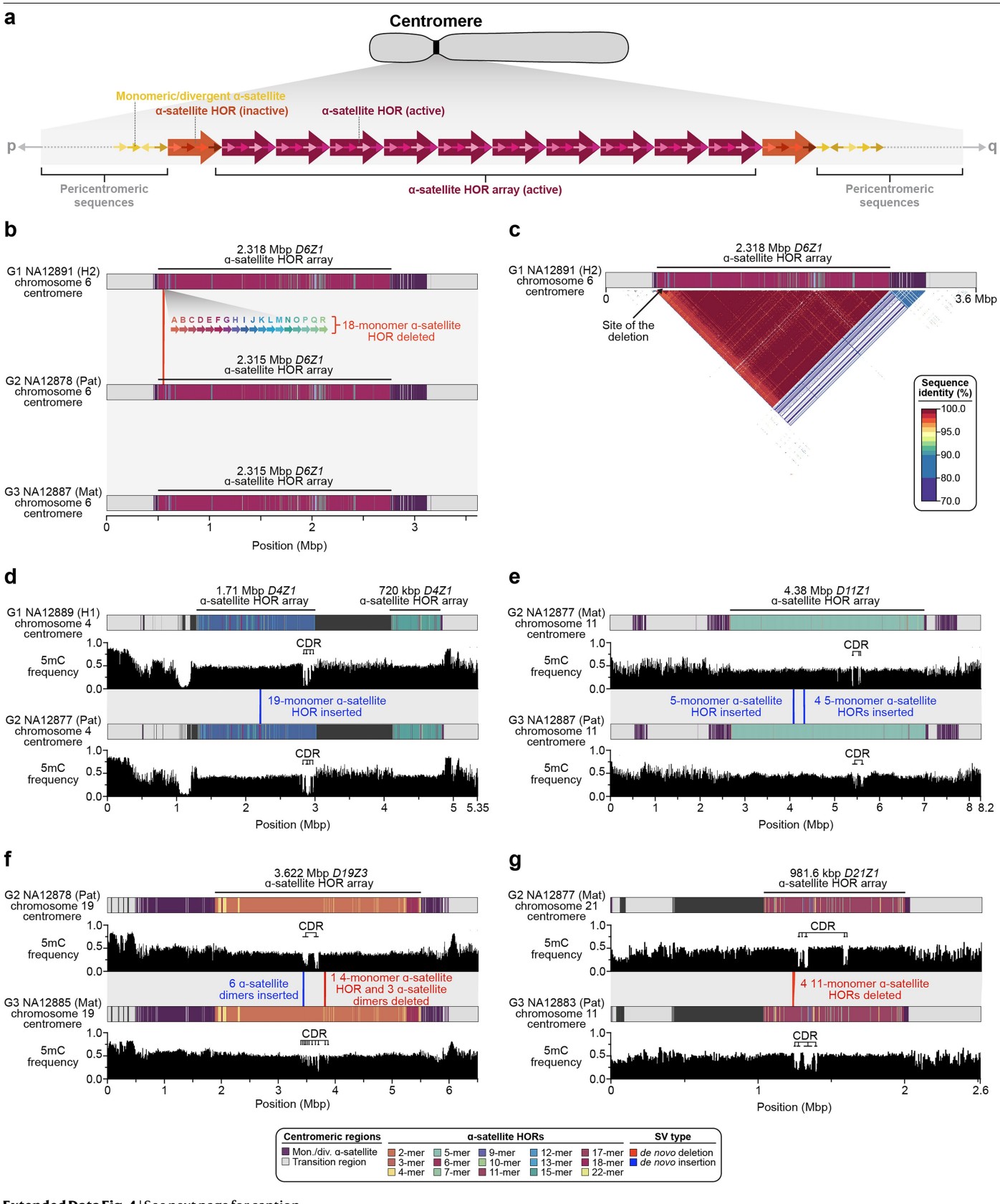

**Extended Data Fig. 4** | See next page for caption.

**Extended Data Fig. 4 | Changes in centromere sequence, structure, and DNA methylation patterns across generations. a**) Schematic of the generalized organization of human centromeres and their flanking sequence. Major components and their structures are shown. HOR, higher-order repeat. Not drawn to scale. **b**) Deletion of an 18-monomer α-satellite HOR within the Chromosome 6 centromere of G2-NA12878 is inherited in G3-NA12887, shortening the length of the α-satellite HOR array by ~3 kbp. **c**) Sequence identity heatmap of the Chromosome 6 centromere in G1-NA128991 shows the high (~100%) sequence identity of α-satellite HORs along the entire centromeric array and at the site of the de novo deletion. **d,e**) Deletions of α-satellite HORs in regions outside of the centromere dip region (CDR) in the **d**) Chromosome 4 and **e**) Chromosome 11 centromeres does not affect the position of the CDR. **f,g**) Deletions and insertions of α-satellite HORs within the CDR in the **f**) Chromosome 19 and **g**) Chromosome 21 centromeres alter the distribution of the CDR.

**Extended Data Table 1 | Recurrently mutated tandem repeat loci**

| Position CHM13 | Motif structure | Unique de novo events | Range of de novo allele lengths (bp) | Range of de novo allele size changes (bp) |
|---|---|---|---|---|
| chr1:54393726-54394070 | (GTGAGA)n(AAACC)n(AAACA)n | 12 | 379 – 529 | -37 – 60 |
| chr8:2376919-2377075 | (GAGGCGCCAGGAGAGAGCGCT)n(ACGGG)n | 10 | 229 – 628 | -57 – 133 |
| chr7:2500010-2500042 | (AAAG)n | 8 | 206 – 436 | -89 – 59 |
| chr4:79949242-79949442 | (TTGA)n(GCATA)n(AGCAC)n | 8 | 745 – 845 | -60 – 31 |
| chr4:21696993-21697153 | (TTATT)n | 8 | 231 – 291 | -15 – 9 |
| chr12:119907035-119907158 | (GGAGAC)n(GAGGCG)n(AGAGGC)n | 8 | 300 – 625 | -66 – 37 |
| chr12:114852499-114852706 | (GAGGG)n(GGAGA)n | 7 | 303 – 520 | -30 – 34 |
| chr7:42892201-42892385 | (AAG)n | 6 | 170 – 251 | -35 – 28 |
| chr21:33731357-33731465 | (GCCACTT)n(ATTCT)n | 5 | 158 – 203 | -10 – 5 |
| chr9:36529968-36530006 | (T)n | 4 | 273 – 303 | -39 – -19 |
| chr7:6540708-6540973 | (CAGGCAGCGCGGGAGGCG)n | 4 | 373 – 549 | 18 – 54 |
| chr7:152489617-152489683 | (AAAAT)n | 4 | 401 – 411 | -15 – -5 |
| chr12:95884953-95885246 | (GGAGAG)n | 4 | 251 – 311 | -12 – 9 |
| chr7:13334154-13334671 | (TTTC)n(TTCT)n(TTTC)n | 3 | 68 – 482 | -82 – 55 |
| chr15:32243116-32243499 | (CGCCGCCGTCCTCGCCG)n | 3 | 400 – 451 | -17 – -17 |
| chr14:95031468-95031513 | (TTTC)n(T)n | 3 | 218 – 222 | -16 – -12 |

Loci with at least three DNMs shown here. See **Supplementary Table 11** for the full list of recurrent TR DNMs.

# Reporting Summary

## Statistics

For all statistical analyses, confirm that the following items are present in the figure legend, table legend, main text, or Methods section.

| n/a | Confirmed | |
|---|---|---|
| ☐ | ☒ | The exact sample size (*n*) for each experimental group/condition, given as a discrete number and unit of measurement |
| ☐ | ☒ | A statement on whether measurements were taken from distinct samples or whether the same sample was measured repeatedly |
| ☐ | ☒ | The statistical test(s) used AND whether they are one- or two-sided <br> *Only common tests should be described solely by name; describe more complex techniques in the Methods section.* |
| ☒ | ☐ | A description of all covariates tested |
| ☒ | ☐ | A description of any assumptions or corrections, such as tests of normality and adjustment for multiple comparisons |
| ☐ | ☒ | A full description of the statistical parameters including central tendency (e.g. means) or other basic estimates (e.g. regression coefficient) AND variation (e.g. standard deviation) or associated estimates of uncertainty (e.g. confidence intervals) |
| ☐ | ☒ | For null hypothesis testing, the test statistic (e.g. *F*, *t*, *r*) with confidence intervals, effect sizes, degrees of freedom and *P* value noted <br> *Give P values as exact values whenever suitable.* |
| ☒ | ☐ | For Bayesian analysis, information on the choice of priors and Markov chain Monte Carlo settings |
| ☒ | ☐ | For hierarchical and complex designs, identification of the appropriate level for tests and full reporting of outcomes |
| ☒ | ☐ | Estimates of effect sizes (e.g. Cohen's *d*, Pearson's *r*), indicating how they were calculated |

*Our web collection on statistics for biologists contains articles on many of the points above.*

## Software and code

Policy information about availability of computer code

| Data collection | The software used to collect sequencing data are Pacific Biosciences SMRT Link (v11.0.0, 11.0.1, and 12.0) and Oxford Nanopore Technologies MinKNOW software (v21.02.17 - 23.04.5). |
|---|---|
| Data analysis | Custom code and pipelines used in this study are publicly available via the following GitHub repositories: <br> https://github.com/orgs/Platinum-Pedigree-Consortium/repositories <br><br> Publicly available software used in this study include: <br> BWA-MEM (v0.7.17-r1188), SAMtools (v1.10), sambamba (v1.0), Verkko (v1.3.1 and v1.4.1), hifiasm (v0.19.5), minimap2 (v2.21, >=v2.24) winnowmap (v2.03), Meryl (v1.0), Merqury (v1.1), TRGT (v0.7.0-493ef25), rustybam (v0.1.33), HiPhase (v1.0.0-f1bc7a8), Clair3 (v1.0.7), GATK (v4.3.0.0), DeepVariant (v1.4.0 and v1.6.0), WFMASH (v0.13.1), PGGB (v0.6.0), VCFBUB (v0.1.0), VCFWAVE (v1.0.3), pbmm2 (v1.1.0), TRF (v4.09.1), RepeatMasker (v4.1.0, v4.1.2-p1, and v4.1.6), BCFtools (v1.16 and v1.17), VCFtools (v0.1.16), ISOGG (v15.73), BEAST (v1.10.4), RAxML (v8.2.10), Tree-Annotator (v.1.10.4), FigTree (v.1.4.4), HMMER (v.3.3.2dev), Gepard (v2.0), PAV (v2.3.4), DipCall (v0.3), MAFFT (v7.508), compleasm (v0.2.4), TRGT-denovo (v0.1.3), Variation graph toolkit (vg, v.1.40.0), muscle (v3.8.31), ASHLEYS (v0.2.0), Flagger (v0.3.3), NucFreq (v0.1), compleasm (v0.2.4), SVbyEye (v0.99.0), SVPOP (v3.4.0), PBSV (v2.9.0), Sniffles (v0.12.0), Sawfish (v2.2), Integrative Genomics Viewer (IGV, v2.16.0), Guppy (v6.3.7 and v6.5.7) <br> We also used following R packages: fastseg (v1.46.0), breakpointR (v1.15.1), regioneR (v1.32.0), DECIPHER (v2.28.0), Biostrings (v2.70.2), StrandPhaseR (v0.99) |

For manuscripts utilizing custom algorithms or software that are central to the research but not yet described in published literature, software must be made available to editors and reviewers. We strongly encourage code deposition in a community repository (e.g. GitHub). See the Nature Portfolio guidelines for submitting code & software for further information.

## Data

Policy information about availability of data

All manuscripts must include a data availability statement. This statement should provide the following information, where applicable:
- Accession codes, unique identifiers, or web links for publicly available datasets
- A description of any restrictions on data availability
- For clinical datasets or third party data, please ensure that the statement adheres to our policy

> All underlying data from 28 members of the family are available as part of AWS Open Data program or dbGaP.
> Variant calls, mapped sequencing data, and assemblies for 23 family members (G1-GM12889, G1-GM12890, G1-GM12891, G1-GM12892, G2-GM12877, G2-GM12878, G3-GM12879, G3-GM12881, G3-GM12882, G3-GM12885, G3-GM12886, G3-200080-spouse, G4-200081, G4-200082, G4-200084, G4-200085, G4-200086, G4-200087, G3-200100-spouse, G4-200101, G4-200102, G4-200104, G4-200106) consented for their data to be publicly accessible similar to the 1000 Genomes Project samples to allow for development of new technologies, study of human variation, research on the biology of DNA, and study of health and disease are available via the AWS Open Data program: s3://platinum-pedigree-data/.
>
> See https://github.com/Platinum-Pedigree-Consortium/Platinum-Pedigree-Datasets for specific details on how to access.
> In addition, mapped sequencing data and assemblies for five family members (G3-NA12883, G3-NA12884, G3-NA12887, G4-200103, G4-200105) that are not consented for open access are available via dbGaP under Accession ID: phs003793.v1.p1 (Platinum Pedigree Consortium long-read sequencing). This includes also variant calls for the whole family (28 members).
>
> The tandem repeat catalogs are available on Zenodo DOI: 10.5281/zenodo.13178746.
>
> The Y-chromosomal assembly for a closely related R1b haplogroup sample HG00731 was downloaded from the Human Genome Structural Variation Consortium IGSR site (https://ftp.1000genomes.ebi.ac.uk/vol1/ftp/data_collections/HGSVC3/working/20230927_verkko_batch2/assemblies/HG00731/).

## Research involving human participants, their data, or biological material

Policy information about studies with human participants or human data. See also policy information about sex, gender (identity/presentation), and sexual orientation and race, ethnicity and racism.

| Reporting on sex and gender | Research participants self-report gender as male, female or other. In the manuscript, sex is reported based on genetic analysis. |
|---|---|
| Reporting on race, ethnicity, or other socially relevant groupings | Research participants self-report race as American Indian/Alaska Native, Asian, Black or African American, Native Hawaiian or other Pacific Islander, or White. Research participants self-report ethnicity as Hispanic/Latino or Not Hispanic/Latino. They also can select that they do not wish to provide some or all of the information. |
| Population characteristics | The contributing study population was selected not for disease but for families of large sibship size, living parents, and living grandparents as described by Dausset et al., 1990, Genomics. Four individuals from the first generation were enrolled at ages 75 to 83 years; two individuals from the second generation were enrolled at ages 57 and 58; seven individuals in the third generation were enrolled at ages 23 to 36, and most recently, the spouses of the third generation were enrolled at ages 58 and 71, and the fourth generation were enrolled at ages 24 to 49. Fourteen individuals are male. Fourteen individuals are female. All family members are White and Non-Hispanic/Latino. |
| Recruitment | Identification and recruitment of large families was through community engagement and word-of-mouth. When eligible families (4 grandparents, 2 parents, 6 or more children) were identified, family members advocated and recruited their immediate family members into the study. There was no selection based on sex or gender. Although there was no selection based on race or ethnicity, all families were ultimately Caucasian/White due to the demographics of the communities involved. |
| Ethics oversight | The study is approved and overseen by the Institutional Review Board of the University of Utah under IRB_00065564. |

Note that full information on the approval of the study protocol must also be provided in the manuscript.

# Field-specific reporting

Please select the one below that is the best fit for your research. If you are not sure, read the appropriate sections before making your selection.

☒ Life sciences          ☐ Behavioural & social sciences          ☐ Ecological, evolutionary & environmental sciences

For a reference copy of the document with all sections, see nature.com/documents/nr-reporting-summary-flat.pdf

# Life sciences study design

All studies must disclose on these points even when the disclosure is negative.

| Sample size | Sample size was defined based on the availability of consented family members of the CEPH (1463) family. |
|---|---|

| Data exclusions | We excluded from the analysis three individuals (NA12880, NA12888, and NA12893) who did not choose consent for biobanking and broad data access. |
|---|---|
| Replication | Whole-genome sequencing was conducted using five complimentary short- and long-read sequencing platforms on the same DNA samples to create a high level of confidence in the genomic data. Two hybrid genome assembly pipelines, hifiasm and Verkko, were applied to reinforce the confidence in the highly contiguous phased genome assemblies. With this rigor, further attempts at replication were not done. |
| Randomization | N/A: this was not an interventional trial. |
| Blinding | N/A: this was not an interventional trial. |

# Reporting for specific materials, systems and methods

We require information from authors about some types of materials, experimental systems and methods used in many studies. Here, indicate whether each material, system or method listed is relevant to your study. If you are not sure if a list item applies to your research, read the appropriate section before selecting a response.

## Materials & experimental systems

| n/a | Involved in the study |
|---|---|
| ☒ | ☐ Antibodies |
| ☐ | ☒ Eukaryotic cell lines |
| ☒ | ☐ Palaeontology and archaeology |
| ☒ | ☐ Animals and other organisms |
| ☒ | ☐ Clinical data |
| ☒ | ☐ Dual use research of concern |
| ☒ | ☐ Plants |

## Methods

| n/a | Involved in the study |
|---|---|
| ☒ | ☐ ChIP-seq |
| ☒ | ☐ Flow cytometry |
| ☒ | ☐ MRI-based neuroimaging |

## Eukaryotic cell lines

Policy information about cell lines and Sex and Gender in Research

| Cell line source(s) | Coriell Institute for Medical Research, NIGMS Human Genetic Cell Repository, CEPH collection. Cell line IDs for 14 members of the CEPH 1463 family: G1-GM12889, G1-GM12890, G1-GM12891, G1-GM12892, G2-GM12877, G2-GM12878, G3-GM12879, G3-GM12881, G3-GM12882, G3-GM12883, G3-GM12884, G3-GM12885, G3-GM12886, G3-GM12887. EBV transformed lymphoblastoid cell lines were generated for G3 spouses and G4 family members (n=13): G3-200080-spouse, G4-200081, G4-200082, G4-200084, G4-200085, G4-200086, G4-200087, G3-200100-spouse, G4-200101, G4-200102, G4-200103, G4-200104, G4-200106. |
|---|---|
| Authentication | Cell lines were authenticated by whole-genome sequencing of the DNA and subsequent variant calling. Sequence results must match a) the sex of the individual, b) sequencing results from blood-derived DNA from the same individual, and c) inheritance pattern of parents and offspring. |
| Mycoplasma contamination | Cell lines were not tested for mycoplasma contamination. |
| Commonly misidentified lines (See ICLAC register) | No commonly misidentified cell lines were used in the study. |

## Plants

| Seed stocks | N/A |
|---|---|
| Novel plant genotypes | N/A |
| Authentication | N/A |

