## [Peer Review File · Nature]

Human *de novo* mutation rates from a four-generation pedigree reference

Corresponding Author: Professor Evan Eichler

Version 0:

Reviewer comments:

Referee #1

(Remarks to the Author)

Porubsky et al present an analysis of *de novo* mutation and recombination in a four generation pedigree with 28 family members. This dataset represents an impressive amount of work and a tremendous resource - 5 sequencing technologies (Pacbio Hifi, UL ONT, strand-seq, Illumina, Element), allowing detailed assessment and validation of a range of variant types including SNVs, TRs, and SVs. It is also great that the data is mostly blood-derived rather than cell lines and many have been consented to be made publicly available. The dataset reveals some novel insights, confirms many previously observed mutation patterns (e.g. paternal bias, similar mutation rates for different variant types to previous estimates), and importantly will serve as a ground truth dataset for future such studies.

Below I have included some suggestions for areas that could be improved or clarified. I also note my experience is more in variant calling and less on assembly, which I hope other reviewers can comment more on.

1. The bimodal parental distribution of inherited segment lengths is interesting. Was this observed previously? If so it would be good to include a citation. If not it would be good to highlight that the finding is novel.
2. Methylation is first mentioned on line 499 and comes out of the blue. I could not find any description of methylation data generation or analysis in the methods. Also some of this part is speculative or about future directions and could be moved to discussion instead of results.
3. Most of the sequencing used in this study is blood derived, whereas a lot of previous data on some of these samples (from 1000G) are based on LCLs. Although I understand this may be beyond the scope of the current study, I was left wondering if there are any insights into biases in sequencing from cell lines previously used? This dataset could offer a great opportunity to compare *de novo*/mosaic mutations obtained from blood vs. artifacts in cell line data which could be mentioned in the discussion.
4. Another opportunity of this dataset is to detect gene conversion events (line 263, Supplementary Fig. 25). (In Supp Fig. 25, did the legend mean to say the child's haplotype 2 is a mixture of alleles from NA12878, not NA12877? That's what the arrows seem to imply). Is it possible though that some of these events could be due to dropout of sequencing data from one of the two alleles in the child in that region? That would be unlikely if it was supported by multiple technologies but it seems like maybe only pacbio data was used for manual inspection of these cases. Is there any prior expectation on the distribution of the number or size of these events that could be used to evaluate if they are reliable or some other way to validate any of those?
5. Although generally the paper is very detailed there were some methods sections that could use more detail: e.g. (1) For compleasm - "see more details at". Can methods summarize important points? (2) Flagger validation - flagged regions were analyzed to identify collapses, etc. -> were specific metrics used for that? (3) On page 41 a novel HMM is described but not very many details are given. E.g. how were the transition/emission probabilities of the HMM set? What was the set of possible states? (4) I could not find where details of the SNV/indel calling with DeepVariant are described. The calls are used for the recombination analysis but not described prior to that section.
6. Line 441: A higher mutation rate is observed for dinucleotide vs. homopolymer TRs - could this be due to challenges in

assaying homopolymers which results in many of those getting filtered?

7. It is exciting that the authors were able to provide a genome-wide analysis of de novo VNTR mutations, which have been hard to include in previous short read based datasets. I was surprised by Figure 3a, which shows mutation rates increase, pretty substantially, with the length of the VNTR repeat unit. As noted this is the opposite of STRs, in which generally longer repeat units have slower mutation rates. The mutation rates for the VNTRs get quite high (approaching 10^{-2}). Has this increasing trend been observed before? Although some potential biological explanations are given in the discussion, I am left wondering if there is any technical artifact that could cause this trend. E.g. are VNTRs with longer units harder to assemble accurately? Or are VNTRs with longer units closer to centromeres or other hard to call regions?

8. Is there any bias for expansion vs. contraction mutations at the STRs, VNTRs, or other indels? Besides mutation rate vs. motif size, do the mutations recover the known relationship between total repeat length and mutation rate? Looking at that relationship for VNTRs may give insight into the previous point regarding the increasing trend between repeat unit length and mutation rate.

9. Although the variant set included is already quite broad, could the authors comment on whether this data can be used to assess copy number variants (CNVs) (and de novo mutations affecting those?).

10. It is indicated that the sequence data and phased genome assemblies will be made available. It would be great if the variant calls will also be released.

Other minor points

1. Figure 1 legend "The pedigree dataset has been expanded..to include...G3 spouses (NA12879 and NA12886) - aren't the other datasets (200080 and 200100) the new ones? This made it sound like NA12879/NA12886 are new.

2. It would be helpful to briefly define the "auN" metric upon first use. Similarly for other metrics used to describe the assemblies - e.g. "quality value" and other jargon terms used throughout.

3. Line 234: "Some of the largest gains occur among SDs" -> what are SDs? Should that be SVs?

4. Line 245: what is a "morbid copy number variant"?

5. Figure 2c mentions a strong paternal age effect but it also seems like there is a correlation with maternal age? It would be nice to annotate with Pearson r/p-value. If the maternal age effect is also significant it is worth pointing out.

6. For Extended Data Fig. 3b: is there a significant correlation between the allele balance vs. fraction of children with the alternate allele for the PZMs?

7. Line 277: "95% [of SNVs/indels] and 70% [of SVs], respectively, show evidence of transmission from G2-G3". What is the expectation? It seems like this implies unsurprisingly that the SV calls are slightly less reliable?

8. Are the PRDM9 genotypes used for anything? It is mentioned briefly (line 295+) but there is no rationale given for why those were computed (to look for correlation with recombination patterns) and that section ends abruptly.

9. Figure 2b: Can the difference in allelic balance for DNMs vs. PZMs be explained by thresholds for detection? E.g. a very common PZM might be hard to distinguish between germline DNM and by definition have to be lower.

10. For the intragenerational recurrent mutations - although it is mentioned in discussion it might be worth mentioning in the main text that this could be due to parental mosaicism.

11. It would be helpful to mention in the methods if the Illumina data was PCR-free? Does the G4 data differ in either pcr+/- or read length compared to the G1-G3 data?

12. Were DNM Y chromosome events similarly assessed for transmission to the next generation as the autosomal events were? I wasn't sure after reading the chrY section.

Referee #2

(Remarks to the Author)

The characterization of mutations and recombinations in families via high quality assembled genomes is novel. These various sequencing technologies used by the authors enabled them to search for mutations in centromeres which are notoriously hard to analyze due to their repeat structure. The accumulation of mutations in the germline in the unique parts of genome are better characterized with much larger cohorts of trios and short reads, this study adds little to that characterization.

The authors are detecting mutations in repeated regions which is a formidable task, but I was hoping for a more nuanced analysis of the accumulation of mutations in repeated regions of the genome. The de novo assembly of the genomes seems to be of high quality and state of the art, however, many of the de novo mutation analysis lack appropriate quality control and

scientific context. For example, is there any correction done for the difference in false positives/negative rates of DNM calls between unique and repeated regions? Further, in the calculation of mutation rates, unique parts of the genome are generally used to derive a denominator but in the case of repeats this assumption breaks down, could the authors expand on this?

The authors are limited by the number of independent trios in this study, as all their analysis are based on single extended family resulting in anti-conservative statistical tests. The authors should use all the available individuals in analysis of DNMs and recombinations, I value the effort made by the authors to differentiate between the generations.

The decrease in the number of crossovers with parental age is extremely strong and unexpected. The family members in the previous generation determine the detection of the crossovers, therefore the p-values in this recombination analysis are anti-conservative as the analysis is based on a single extended family. These results are going against the grain and are incompatible with current recombination results based on sequence level information (PMID: 30679340), where there is a weak positive age effect for crossovers in the maternal germline and non-significant age-effect in the paternal one. The authors see a strong decrease in number of crossovers in both germlines despite the recombination processes in males and females are wildly different. This result raises questions like, are the assembly metrics of the phased genome assemblies correlated with number of detected crossovers? Further, what would be biological mechanisms to decrease the number of crossovers with age in both germlines?

The manuscript reads like that sections of the manuscript were written by independent authors and loosely coupled together; it would be beneficial to harmonize the reporting of false positives/negatives in the manuscript across the different variant types. Further, the manuscript is too long in its current form as the authors are repeating themselves. This leads to the fact that the interesting aspects of the manuscript (DNMs in the repeated regions) are overshadowed by lengthy technical text not approachable by the general scientist.

Specific points:

L153: What does AuN stand for?

L174: There is no e panel in figure 2.

L177: Need to explain the scale of the quality value and its interpretation to the general reader. This metric is okay for evaluating an assembly but of limited value for QC of DNMs as it suggests that roughly 3,000 false positives would be detected assuming QV of 60.

L179: Is the misorientation rate on the contig level or base level?

L180: Is this for a pair of heterozygotic genotypes?

Extended figure1d: This presentation of the assemblies' metrics could be improved.

L227-228: The parent pair in G2 has 8 offspring and there are two individuals in G3 that have 6 offspring. Are the authors referring to a transmission to at least one offspring of a parent pair or per trio? Given that the authors report a transmission rate of 95% it appears to be the former. In an ideal world you expect the probability of transmission to at least one child, to be $1 - 0.56 \approx 98.4\%$. This indicates that the SNV and SV truth sets are of low quality. It would be beneficial to describe to the general reader what to expect and the observed deficit in transmission. The qc metrics employed by the authors could be improved.

L238: Missing transmission rate per trio.

L243: Missing transmission rate per trio.

L254: What is the correlation between the number of contigs and the number of crossovers? And other assembly metrics such as AuN?

L258: Why did the authors exclude the G4 individuals from this analysis? Especially in the light of limited sample sizes.

L269: Depending on the genetic length of the chromosome?

L275-L284: The tests are probably anti-conservative as these tests are mainly based on offspring (G3) from single parent pair (G2).

L289: Is the difference significant? Especially when considering the trio as the sample unit?

L320: What is the correlation between the AB of DNM using Illumina and Element? How many DNMs/PZM showed a

significant AB difference between methods?

L322: Why did the authors exclude G4 individuals from this analysis? I realize some analyses are only possible for the different generations, but most of the analyses portrayed in Figure 2 could be done with the entire set.

L327: It is not clear to me how the authors derive that a sequence variant that is inconsistent with a haplotype is necessarily a PZ mutation that occurred early in development. There are other plausible explanations for that, somatic mutations that occurred postnatally and rise in frequency with advancing age (e.g. clonal hematopoiesis) and/or genotype artifacts. Postnatal somatic mutations and artifacts would be expected to be in equal frequency on the paternal/maternal chromosomes of the proband.

L329: How does the allelic balance of the transmitted PZ allele look like in the offspring? The allele balance should be around 50%? Do the offspring carrying the transmitted PZ allele share a haplotype? The authors could try to construct the phylogenetic trees of the early cell lineages in the G2 parents (similar to Figure 3 from PMID: 31492841 and Figure 2 in PMID: 30397338). 64.5% transmission to at least one child suggests 7% ($7\% \approx 1 - \exp(\log(0.645)/6)$) transmission rate which is much lower than the roughly 25% allelic balance in Figure 2b. This suggests that these post-zygotic mutations calls are generally somatic mutations or false positives.

L357: Please state the p-value and reword the statement that they differ, and references to previous studies of mutational spectrum of PZ mutations would be appropriate here.

L362-L371: Given that the PZ mutations are generally not in the germline based on the transmission patterns, how are the authors correcting for this fact? Do the authors quantify difference in false positive/negative rate across the different regions? The fact that the PZ mutations generally do not transmit, and the authors see a greater fraction of PZ mutations in the SDs, this indicates that there is a higher false positive rate in the SDs. Furthermore, given that the SD and centromeres are composed of variable number of repeats this introduce a variability in the denominator, could the authors expand on this?

L388: How about the transmission rate to a single offspring?

L404: Did the authors try to use the HiFi data to validate these STRs/VNTRs DNMs via adjacent het SNPs?

L423: Once again 76.2% transmission to at least one child of 6 is a severe under transmission from the expected 50% transmission to a single child. How does this look for homopolymers?

Figure 3d: It is hard to see the lines in this plot and this visualization could be improved. Are the TR allele sizes consistent with flanking phase informative SNVs? For example, reads in NA12877 supporting the 248 allele are they linked with SNVs on the H1 haplotype of the NA12889?

L479: Most of the centromeres are not completely sequenced and assembled, it would be useful to the reader to state the fraction of centromeres that were assembled.

L482: What is the transmission rate here? This is especially important here as the event indicated in Figure 4d could be a somatic cell-line duplication in G1 NA12891 which would appear as a germline deletion in G2 NA12788. It would be beneficial to have the genotypes for all the individuals as a supplementary table or figure for these 18 events.

L485-487: This sentence is cryptic, it would also be nice to have a schematic view of the terminology used here for the general reader (HOR arrays, HOR cassettes and pericentromeric flanking sequence).

L509-L512: What is the false positive/negative rate of the SNVs detected within these alpha-satellite HORs? As before the denominator is probably tricky to derive within these repeats.

L548-L555: How many DNMs here are transmitted to the next male? They seem to generally being transmitted to G2 based on Figure 5 but it would be nice to get a quantification of that. There is no comparison to previous estimates of the Y de novo mutation rate, it would be advisable to restrict to the regions of the Y that are accessible to short reads to see if the estimate of the mutation rate is comparable to previous ones.

L586: Missing quantification for this statement, and this statement should be reworded according to the level of evidence from the quantitative analysis.

L595: Most of the double strand breaks (DSB) formed during meiosis will not be resolved as crossovers, the lack of correlation between crossover location and de novo SVs in this limited number of meiosis is inconclusive. The interpretation should reflect that.

L603-606: Is this consistent with the PZ SNV mutations transmitted to the offspring? The early mutations transmitted to the offspring determine the early cell lineages of parent.

L631-634: All the claims should be revisited after quantification of the false positive/negative rates.

L646-648: Missing CIs for the estimate presented here, and it would be cleaner to have this comparison in the results.

L650-655: I am confused as most of the PZ mutations identified by the authors did not transmit (100%-64.5%) despite the numerous offspring and here they are attributing them all to the germline. In a previous analysis by the authors [PMID: 31549960] and using monozygotic twins [PMID 33414551] a much lower fraction of pre-PCGS mutations is detected.

L657: Is this reference appropriate? The reference is describing mitotic errors, rather than mutations. There are numerous studies looking at this developmental epoch through lens of colonies/laser dissection which the authors could refer to. Further, is the rate of mitotic divisions abnormally high in the cleavage state compared to somatic cell divisions?

Referee #3

(Remarks to the Author)

This paper describes telomere-to-telomere assembly of a four-generations pedigree, with the goal of studying de novo mutations and recombination rates. Although this pedigree has been analyzed before, the previous studies were limited by short-read sequencing; thus only small de novo mutations were profiled in previous studies. This work uniquely enables the analysis of highly repetitive regions of the genome, such as segmental duplications and centromeres, which more than doubled the number of discovered de novo mutations. High quality phased assemblies also enabled generation of high-resolution recombination maps.

Overall, this paper highlighted important new biological insights on de novo mutations and recombination, and generated a highly valuable resource for many future studies, as the authors are making sequencing data, assemblies and variant calls publicly available. This work employs state of the art assembly, variant calling and QC techniques, and convincingly demonstrates the high quality of the produced resources. I only have a few small requests below.

1. It seems that manual curation was required for generating the final call set for de novo SVs and centromeric variants. It is understandable that the fully automated generation of this benchmark may not be practical. I am curious how many variants were selected for manual screening and what was the rate of rejected variants? Did the analysis reveal any patterns of systematic errors?
2. The authors used multiple tools and techniques to generate initial de novo variant candidates. For example, to generate an initial set of SVs, multiple reference-based tools, as well as linear assemblies and pangenome graphs analysis were used. It would be interesting to see if the different variant calling methods were in general in good agreement. Was there any evidence that some real variants may be uniquely captured by just one or a few methods only?

Version 1:

Reviewer comments:

Referee #1

(Remarks to the Author)

The authors have done an impressive job responding to the comments. I appreciate the new detailed analysis of VNTR vs. STR mutation rates. The revised Figure 3a makes the point nicely that a small number of VNTRs with large units have very high mutation rates, and the observed "trough" in mutation rates is a really cool finding. The cell line artifact is also a nice addition and highlights the benefits of using the blood-derived data. Some minor comments on the cell line part:

1. It wasn't clear how the mutations from Conrad et al. were classified as cell line artifact vs. false positives. Did you use data from the current study to do that classification?
2. "Note, because we are using DNA derived from primary material (peripheral blood), the number of SNVs arising from cell line artifacts is dramatically reduced" -> aren't cell line artifacts eliminated altogether, not just reduced? Of course there are other sources of error but I wouldn't classify them as cell line artifacts.

Below are some minor comments on statistics and figure legends:

1. L245-246: indicate if p-values are one or two-sided. I assume two-sided
2. Extended data figure 2d, Figure 2c, Supp Figure 31: I could not find how the gray shaded regions, which I assume represent confidence intervals, are defined.
3. Figure 2b, Extended data figure 2c, supp figure 22, supp figure 33c: boxplot elements are not defined
4. Figure 2d, supp fig 33c: I could not find the significance levels denoted by different numbers of *'s.
5. For some of the tests comparing maternal vs. paternal statistics a t-test is used (comparing maternal vs. paternal recombination events) whereas elsewhere a Wilcoxon signed rank test is used (number of DNMs on maternal vs. paternal

haplotypes). Wilcoxon signed rank test seems more appropriate since it compares matched samples and has fewer assumptions.

6. There are many places where rates are given for mutations/bp/generation with a 95% CI. In the legend of Figure 3a it is described how the CI is constructed (Poisson confidence intervals (computed using a chi-square distribution)). Is this how the other ones were computed? This info should be stated in the text, perhaps just the first time it is used if it is the same for all of the subsequent CIs.

7. Supp Figure 1: define the red dots

8. Supp Figure 42: the legend doesn't indicate what red vs. black mean in the bar plots

Referee #2

(Remarks to the Author)

The authors have addressed my concerns about the unexpected decrease in number of crossovers with advancing parental age, although the presentation and statistical rigor could be improved. There is a clear parent pair effect in this set which is probably of technical nature. This strongly suggests the p-values reported in the main text (L245-246) and in Extended Data Figure 2d are anti-conservative and a mixed effect model would be more appropriate with random effect for the parent pair.

The authors have improved the analysis of the false positive rate across the different DNM types and genomic regions. More specifically by estimating and reporting the transmission rate for the different genomic regions and types of variants, it has enabled the reader to gauge the accuracy of their DNM calls. This work by the authors has clearly resulted in better estimates of the mutation rates and improved presentation of their results.

However, the quantification and presentation of the post-zygotic mutation contribution should be improved. The authors quote that 17% of DNMs are post-zygotic per transmission in origin in the abstract, there are numerous issues with this statement. To my understanding, this statement is based on a simple aggregate across the G2 and G3 generations, tallying the number of mutations that are fully linked with (germline/pre-zygotic) or not (post-zygotic) with one of the parental haplotypes. However, most of the individuals (6/8) in the G3 generation do not have sequenced children, therefore the authors have generally no information whether these PZM variants in G3 will be transmitted to the next generation, i.e. G4. Further, a PZM transmitted to multiple offspring should be weighted differently than one transmitted to a single offspring. This is especially important in the case of a PZM with no evidence of transmission which could just be confined to a somatic cell type and hence irrelevant for estimation of germline mutation rates.

Based on Supplementary Figure 45 there seems to be transmission of 24 PZM variants from G2 to G3 resulting in 61 transmitted PZM-DNMs to the G3 offspring or on average 7.6 PZM mutations are transmitted from G2 to G3. In contrast the G3 have received 506 germline mutations from G2 (based on Supplementary Table 10) resulting in 63 DNMs on average, then the fraction of PZM in G3 received from G2 is roughly 11% ($7.6/(63+7.6)$) based on these data. This roughly halves the statistic reported by the authors and is more in line with other reported estimates. However, it is hard to generalize this estimate from the authors as this is based on developmental mutations identified in two individuals and there is considerable variability in the contribution of these early developmental mutations to the different cell types (PMID:34433963; 33414551; 38605218).

Further there are clear QC issues with the PZMs (Supplementary Figure 45), false positive calls seem to be more frequent in the G3 generation and are present in nearly all of the individuals in high frequency (~25%). This indicates that the false positive variants are biased towards more frequent variants, which hinders accurate quantification of PZMs contribution to the germline for individuals that do not have children.

In other words, this indicates that it does not suffice to restrict to mutations that are partially linked to long reads to claim that they will be accurately transmitted to the next generation based on their AB. These artifacts would probably have been evident by reporting the grand parental haplotypes of the G3 individuals at the PZM positions in Supplementary Table 10.

Specific Comments:

- L321-328: This PZM rate should be revisited in the context of transmitted mutations.
- L327: Further given the high frequency false positives in G3 the enrichment in SDs should be revisited how many of the FPs (3)
- L635-648: This should be revisited and here the authors should acknowledge the limitation of their setup to assess the quality of the PZM calls especially in the SD regions as they can only assess transmission of 40 PZMs.
- Supplementary Table 10: It would be useful to get the AB of the parents from all sequencing technologies for the SNVs/Indels in Supplementary Table 10. Further, the grand parental origin of the children's haplotypes for the carriers and non-carriers of the DNM allele would help in the QC of the PZM transmissions.

Referee #3

(Remarks to the Author)

I thank the authors for answering my requests in detail. I have no other critical points. I have one more suggestion that I think may be helpful for the readers. The new supplementary figure 13 now shows an upset plot with all SVs called by different

methods. It could be helpful to see a similar plot, but only for the curated SV variant set. This way a reader can see which tools were better at detecting the true variants. I leave this to the discretion of the authors and do not need to re-review the manuscript.

Version 2:

Reviewer comments:

Referee #2

(Remarks to the Author)

The authors model the number of recombinations per individual against parental conceptions, with an exponential effect. It is not clear to me why a log-link would be natural for this compared to a linear-link (you can use `family=poisson(link='identity')` in R). I appreciate that the authors tried to incorporate a random effect into the regression, and it seems to be the case that the authors are limited by the effective sample size, which is a caveat of their family design. I also appreciate that the authors admit further sampling is needed, however, proper scientific context of the result is necessary. I have pointed out this in a previous round of review, this result is going against the grain and there is no obvious biological mechanism that would lead to an equal decrease in both germ lines. This context should be described in their manuscript and that this is in opposite direction with previous estimates of the maternal age effect (6.4 cM/year) based on 70,086 maternal meioses (PMID: 30679340).

The description of the postzygotic mutations in the main manuscript is much clearer, and I value haplotype confirmation of the postzygotic mutations. However, it is clear that we disagree on the quantification of relative contribution of postzygotic mutations in the abstract. This quoted "16% of de novo SNVs are postzygotic in origin with no paternal bias" in the abstract is misleading as the authors are counting constitutional mutations equally with mosaic mutations in the next generation as mentioned before. Perhaps the authors can approach this from a different perspective, report the absolute number of postzygotic mutations per individual rather than mixing un-transmitted and transmitted mutations. Further, this sentence is incomplete as all de novo SNVs are postzygotic in some generation, the more relevant question is which generation they occurred in and what is their contribution to the next generation. In other words, the sentence is missing a qualifier, "early" or something similar.

I have no further comments if these two issues are resolved, and I congratulate the authors on their important scientific work of detecting mutation in the repeated parts of the human genome.

REFEREE #1:

Porubsky et al present an analysis of de novo mutation and recombination in a four generation pedigree with 28 family members. This dataset represents an impressive amount of work and a tremendous resource - 5 sequencing technologies (Pacbio Hifi, UL ONT, strand-seq, Illumina, Element), allowing detailed assessment and validation of a range of variant types including SNVs, TRs, and SVs. It is also great that the data is mostly blood-derived rather than cell lines and many have been consented to be made publicly available. The dataset reveals some novel insights, confirms many previously observed mutation patterns (e.g. paternal bias, similar mutation rates for different variant types to previous estimates), and importantly will serve as a ground truth dataset for future such studies.

Below I have included some suggestions for areas that could be improved or clarified. I also note my experience is more in variant calling and less on assembly, which I hope other reviewers can comment more on.

We thank the reviewer for the thoughtful feedback and the acknowledgement of the amount of work we put into this manuscript by generating a multigenerational genome resource with multiple sequencing technologies. The data should aid future discovery of inherited and *de novo* variation and a benchmark for the development of new computational tools to do so.

Major points:

1. The bimodal parental distribution of inherited segment lengths is interesting. Was this observed previously? If so it would be good to include a citation. If not it would be good to highlight that the finding is novel.

This observation has been reported previously (Broman et al. 1998; Kong et al. 2002; Bhérer, Campbell, and Auton 2017). It is known that paternal meiotic breakpoints tend to cluster at chromosome ends, which results in bimodal distribution of inherited parental segments. We modified the text to include relevant citations:

“Paternal recombination is significantly biased towards the ends of human chromosomes with 55 paternal recombination events mapping within 2 Mbp of the telomere compared to one in females creating a bimodal paternal distribution of inherited segment lengths (Broman et al. 1998; Kong et al. 2002; Bhérer, Campbell, and Auton 2017) (**Extended Data Fig. 2c, Supplementary Fig.30, Methods**).”

2. Methylation is first mentioned on line 499 and comes out of the blue. I could not find any description of methylation data generation or analysis in the methods. Also some of this part is speculative or about future directions and could be moved to discussion instead of results.

We thank the reviewer for pointing this out as we indeed omitted this description in the Methods section. We now include the description of this analysis in the Methods and revised the text:

“Although the data are still preliminary, we also assess 18 SV events for their potential effect on the hypomethylation pocket associated with the centromere dip region (CDR)—a marker of the site of kinetochore attachment (**Methods**).”

Methods:

“CpG methylation analysis

To determine the CpG methylation status of each centromere, we first base called raw ONT data with Guppy (<https://community.nanoporetech.com>; v6.5.7) using the “sup-prom” model and the “dna_r9.4.1_450bps_modbases_5hmc_5mc_cg_sup_prom.cfg” config file. Next, we aligned the ONT data from each sample to the respective genome assembly using minimap2 (v2.28) with the following parameters: -ax lr:hq -y -t 4 -l 8g. We converted the resulting BAM file to a bedMethyl file using modbam2bed (<https://github.com/epi2me-labs/modbam2bed>) and the following parameters: -e -m 5mC --cpg -t {threads} {input.bam} > {output.bed}. Then, we converted the bedMethyl file into a bedGraph using the following command: `awk 'BEGIN {OFS="\t"}; {print $1, $2, $3, $11}' {input.bed} | grep -v "nan" | sort -k1,1 -k2,2n > {output.bedgraph}`. Then, we converted the bedGraph into a bigwig using bedGraphToBigWig (<https://www.encodeproject.org/software/bedgraphtobigwig/>) and then visualized the bigwig with Integrative Genomics Viewer (Robinson et al. 2011) (IGV; v2.16.0). To determine the size of a hypomethylated region (termed “centromere dip region” or CDR (Gershman et al. 2022; Altomose et al. 2022) in each centromere, we used CDR-Finder (<https://github.com/arozanski97/CDR-Finder>), which first bins the bedGraph into 5 kbp windows, computes the median CpG methylation frequency within windows containing α -satellite [as determined by RepeatMasker (Smit, Hubley, and Green 2013) (v4.1.0)], selects bins that have a lower CpG methylation frequency than the median frequency in the region, merges consecutive bins into a larger bin, filters for merged bins >50 kbp, and reports the location of these bins.”

3. Most of the sequencing used in this study is blood derived, whereas a lot of previous data on some of these samples (from 1000G) are based on LCLs. Although I understand this may be beyond the scope of the current study, I was left wondering if there are any insights into biases in sequencing from cell lines previously used? This dataset could offer a great opportunity to compare *de novo*/mosaic mutations obtained from blood vs. artifacts in cell line data which could be mentioned in the discussion.

We agree that this is a very interesting subject and the cell lines in this family have been previously studied for *de novo* mutations (DNMs) (Conrad et al. 2011; Eberle et al. 2017). We adjusted Supplementary Note 1 as follows:

“To evaluate the extent of single-nucleotide variants (SNVs) arising from cell line artifacts, we compared our callset with previous studies (Conrad et al. 2011; Eberle et al. 2017). For example, Eberle and colleagues cataloged ~1,869 *de novo* SNVs per individual (Eberle et al. 2017) indicating that cell line *de novo* events were ~23x more common than true *de novo* events. Although the data from Eberle et al. is not easily accessible, we were able to reevaluate those from Conrad and colleagues. In that study, there are a total of 3,236 mutations reported with respect to NCBI36 (hg18). Of those, we were able to liftOver 3,038 sites to GRCh38

coordinates. Among these, there are 48 germline *de novo* mutations (DNMs) and 888 predicted cell-line-specific DNMs. The rest are inherited variants and false positives (**Supplementary Fig. 42a**). Of the 48 predicted germline DNMs, 45 are also reported in the platinum DNM callset (see Results section ‘*De novo* SNVs and small indels’, **Supplementary Table 10**). Only three germline DNMs are not seen in our callset (**Supplementary Fig. 42b**). There are 38 DNMs reported only in our manuscript and, of those, 10 were reported by Conrad and colleagues as ‘False positive call’ or ‘no call’. So 28 (no overlap with Conrad et al. 2011) are truly unique sites in our callset (**Supplementary Fig. 42c**) with 10 likely missed by Conrad and colleagues. Importantly, none of our reported DNMs were marked as cell line artifacts by Conrad et al. (2011). Further, we evaluated sequence properties of reported cell-line-specific DNMs (n=888, “cell line DNMs”). Specifically, we estimated the frequency of transition (Ti) and transversion (Tv) for various categories and found that the Ti/Tv ratio for true germline events was more than double (2.43 vs. 0.95) that of those originating from cell line artifacts.”

Supplementary Figure 42: Evaluation of cell-line-specific artifacts.

a) Counts of various mutation classes reported by Conrad and colleagues (Conrad et al. 2011) stratified by mutation ID. A total of 3,038 sites were lifted over to GRCh38 coordinates. **b**) A Venn diagram showing an overlap between the current set of single-nucleotide DNMs (Platinum pedigree; n=83) and Conrad and colleagues (n=48). **c**) Overlap of platinum pedigree DNMs with Conrad and colleagues DNMs colored by mutation ID. **d**) A barplot showing Ti/Tv ratios per defined mutation ID in Conrad and colleagues DNMs.

Although not a main finding of the paper, we added this analysis to the Supplementary Notes and refer to it in the main text as follows:

“We establish a variant truth set and identify a total of 5.95 million SNVs/indels and 35,662 SVs—all of which are pedigree consistent across the second and third generations (**Supplementary Table 5, Supplementary Fig. 13, Methods**) (Kronenberg et al. 2024). Of the 5.95 million, 77% of small variants are supported by all three technologies and callers. Note, because we are using DNA derived from primary material (peripheral blood), the number of SNVs arising from cell line artifacts is dramatically reduced when compared to previous studies (Conrad et al. 2011; Eberle et al. 2017) (**Supplementary Note 1**).”

4. Another opportunity of this dataset is to detect gene conversion events (line 263, Supplementary Fig. 25). (In Supp Fig. 25, did the legend mean to say the child’s haplotype 2 is a mixture of alleles from NA12878, not NA12877? That’s what the arrows seem to imply). Is it possible though that some of these events could be due to dropout of sequencing data from one of the two alleles in the child in that region? That would be unlikely if it was supported by multiple technologies but it seems like maybe only pacbio data was used for manual inspection of these cases. Is there any prior expectation on the distribution of the number or size of these events that could be used to evaluate if they are reliable or some other way to validate any of those?

Indeed, there is a typo in the figure legend, which has been corrected as stated below.

We spent several months trying to discover allelic gene conversion events but many of our sites mapped to low-complexity DNA and failed to validate across platforms. When detecting allelic gene conversion (AGC), we implemented a conservative approach to avoid detecting false positives caused by assembly errors or misphased variants. Therefore, we only analyzed variable sites detected in phased assemblies supported by at least two read-based callers (either DeepVariant-HiFi, Clair3-ONT, or dragen-Illumina). This resulted in an initial list of 78 putative AGC events. These were further examined for support in phased HiFi reads. However, our manual inspection reports only eight AGC events with such support. For instance, **Supplementary Fig. 25** showcases such a validated event.

The possibility of sequence dropout for our reported 78 events is unlikely since each evaluated variant (SNV or indel) had to be observed in read-based callsets as well as assembled in our assemblies. However, we appreciate that our visual validation rate might be affected by errors in PacBio read phasing. The gene conversion events are about 500 bp to a few kbp in size, but in general shorter than 100 kbp (Williams et al. 2015; Halldorsson et al. 2016). Previous studies reported approximately 103 sites discovered among 98 meioses (Williams et al. 2015), while others reported very similar gene conversion rate of ~5.9 conversions/Mbp per generation (Halldorsson et al. 2016). Given our strict filtering criteria and requirement of at least two consecutive variants to be involved in AGC, we are likely underestimating the number of AGC events in our current study.

In addition to correcting the typographical error, we added the citations to the main text. We also made a note that our reported AGC events are most likely an underestimate due to our strict filtering criteria:

“The child’s haplotype two is composed of a mixture of alleles from NA12878, which is in line with allelic gene conversion or eventually two double-strand breaks and resolved as cross-over.”

“We also characterize 78 smaller haplotype segment “switches” in G3 (median size of ~1 kbp) (Ahn and Livingston 1986; Williams et al. 2015; Halldorsson et al. 2016) that would be consistent with either a double crossover or an allelic gene conversion event, although many (n=17) overlap with low-complexity DNA. This is likely an underestimate due to our strict filtering criteria (**Supplementary Table 9, Supplementary Fig. 25, Methods**).”

5. Although generally the paper is very detailed there were some methods sections that could use more detail: e.g. (1) For compleasm - “see more details at”. Can methods summarize important points? (2) Flagger validation - flagged regions were analyzed to identify collapses, etc. -> were specific metrics used for that? (3) On page 41 a novel HMM is described but not very many details are given. E.g. how were the transition/emission probabilities of the HMM set? What was the set of possible states? (4) I could not find where details of the SNV/indel calling with DeepVariant are described. The calls are used for the recombination analysis but not described prior to that section.

1) We appreciate the reviewer pointing this out. We expanded these sections in the Methods including more details as follows:

“Gene completeness validation

To evaluate the completeness of single-copy genes in our assemblies, we used compleasm (v0.2.4). See more details at <https://github.com/huangnengCSU/compleasm>. We ran compleasm with the following parameters:

```
compleasm.py run -a {assembly.fasta} -o results/{sample.id} -t {threads} -l {params.lineage} -m {params.mode} -L {params.mb_downloads}
```

```
-l primates
```

```
-m busco
```

```
-L {params.mb_downloads}
```

```
downloaded using: compleasm_kit/compleasm.py download primates
```

Flagger was used to detect misassemblies using HiFi read alignments to the assemblies and the assemblies aligned to the reference genome (github.com/mrvollger/asm-to-reference-alignment.git). Regions were flagged based on read alignment divergence and specific reference-biased regions. A reference-specific BED file (chm13v2.0.sd.bed) was used setting a maximum read divergence of 2% and

specifying reference-biased blocks. These flagged regions were analyzed to identify collapses, false duplications, erroneous regions, and correctly assembled haploid blocks with the expected read coverage.

We used Flagger v0.3.3 (<https://github.com/mobinasri/flagger>) to run the "flagger_end_to_end" WDL.

Required inputs include following:

1. Read-to-contig alignments - Winnowmap alignments of all HiFi reads to the assembly (hap1, hap2 and unassigned.fasta)
 2. A combined assembly fasta file with hap1, hap2 and unassigned contigs
 3. BAM alignments of assembly to the CHM13v2.0 reference
- hap1, hap2 and unassigned fasta files of the assembly were aligned to CHM13v2.0 using this pipeline: <https://github.com/mrvollger/asm-to-reference-alignment>."

2) Thank you for pointing out that we were missing a few key details of the HMM. We revised the text to point readers to a follow-up paper that is now available on bioRxiv (<https://www.biorxiv.org/content/10.1101/2024.10.02.616333v1>). We provide a full description of the model in the methods of this manuscript. Additionally, as per the reviewer's request, we uploaded the emission/transmission probabilities to GitHub (<https://github.com/Platinum-Pedigree-Consortium/Platinum-Pedigree-Inheritance/tree/main/pipelines/inheritance-vectors/data>).

Revised text:

"DeepVariant calls from HiFi sequencing data from G1, G2, and G3 pedigree members allow us to identify the haplotype of origin for heterozygous loci in G3 and infer the occurrence of a recombination along the chromosome when the haplotype of origin changes between loci. An initial outline of the inheritance vectors was identified by first applying a depth filter to remove variants outside the expected coverage distribution per sample; inheritance was then sketched out via a custom script, requiring a minimum of 10 single-nucleotide variants (SNVs) supporting a particular haplotype, and manually refined to remove biologically unlikely haplotype blocks, or add additional haplotype blocks, where support existed, and refine haplotype coordinates. Missing recombinations were identified from the occurrence of blocks of pedigree-violating variants, matching the location of assembly-based recombination calls. We developed a hidden Markov model framework to identify the most probable sequence of inheritance vectors from SNV sites using the Viterbi algorithm. For details including the transition/emission probabilities see Kronenberg et. al. 2024 and the associated GitHub repository (<https://github.com/Platinum-Pedigree-Consortium/Platinum-Pedigree-Inheritance>). The transition matrix defines the probability of a given inheritance state transition (recombination). The emission matrix defines the probability that a variant call at a particular locus accurately describes the inheritance state. The values contained within transition and emission matrices were refined to recapitulate the previously identified inheritance vectors, while correctly identifying missing vectors. The Viterbi algorithm identified 539 recombinations, a maternal recombination rate of 1.29 cM/Mbp, and a paternal recombination rate of 0.99 cM/Mbp. Maternal bias was observed in the pedigree, with 57% of recombinations identified in G3 of maternal origin.

The DeepVariant details are already included in the Methods. The human WGS WDL contains detailed information on versioning and small variant calling:

“Read-based variant calling

PacBio HiFi data were processed with the human-WGS-WDL (<https://github.com/PacificBiosciences/HiFi-human-WGS-WDL/releases/tag/v1.0.3>). The pipeline aligns, phases, and calls small variants (using DeepVariant) and SVs (using PBSV). We used the aligned haplotype-tagged HiFi BAMs for all downstream PacBio analysis.”

6. Line 441: A higher mutation rate is observed for dinucleotide vs. homopolymer TRs - could this be due to challenges in assaying homopolymers which results in many of those getting filtered?

Assaying homopolymers is indeed challenging. As mentioned in the main text, many *de novo* alleles at homopolymer loci were less likely to be supported by an orthogonal sequencing technology (Element AVITI) than DNMs at non-homopolymers. This indicates that technical artifacts in the HiFi sequencing data likely impact our ability to detect homopolymer DNMs, and we include a caveat to this effect (original lines 412-415). However, we also note that almost every prior study of STR mutation rates has observed the same bias toward lower mutation rates at homopolymers, regardless of the sequencing technology used (Mitra et al. 2021; Steely et al. 2022; Kristmundsdottir et al. 2023).

To more rigorously address the reviewer’s concern, we tested whether any of our DNM filtering criteria (enumerated in the Methods) might be removing homopolymer DNMs more often than non-homopolymer DNMs. We created a "lenient" DNM callset using the most basic filtering criteria possible: we required ≥ 1 read to support a *de novo* allele in the child, with no additional filters on minimum sequencing depth, parental evidence for the *de novo*, etc., generating a total of 33,477 TR DNMs, for an average of 4,184.6 TR DNMs per sample. We then performed our Element consistency analysis on the “lenient” STR DNMs to determine whether the more stringent filtering criteria described in the manuscript could be removing a substantial fraction of real TR DNMs. As reported in the manuscript (original lines 412-413), 3 of 20 homopolymer DNMs passed our Element consistency analysis, while 53 of 60 non-homopolymer STR DNMs passed. In the lenient STR DNM callset, we found that 7 of 614 homopolymer DNMs and 55 of 554 non-homopolymer STR DNMs passed Element consistency analysis. Thus, we likely “missed” four true positive homopolymer DNMs using our more stringent filtering criteria, along with two true positive non-homopolymer STR DNMs. This result suggests that our filtering criteria are unlikely to remove a substantial number of homopolymer DNMs.

Additionally, we more closely examined the inferred sizes of *de novo* expansions and contractions at TR DNMs and found that 172 of 863 (19.9%) of DNMs were expansions or contractions of a single base pair. Most 1 bp DNM events were at homopolymers (76 of 172; 44.2%). Previously, we did not require TR expansions/contractions to be exact multiples of motif

sizes, because HiFi sequencing reads often exhibit “stutter” that make it challenging to precisely determine the size of a *de novo* allele. However, we found that removing 1 bp DNM events at non-homopolymer loci improved our orthogonal validation rates using the Element technology and, therefore, removed them as part of our updated filtering criteria. This additional filter does not impact our total count of homopolymer DNMs, but it slightly reduces our estimated mutation rates at other STR motif sizes. Nonetheless, we still observe a higher mutation rate at dinucleotides than at homopolymers.

We adjusted the text (L52-3):

Abstract:

“...65.3 *de novo* indels or structural variants (SVs) originating from tandem repeats...”

We revised the text downward throughout this section to reflect this new assessment:

Results:

“After Element and transmission validation, we found an average of 65.3 TR DNMs (including STRs, VNTRs, and complex loci) per sample and estimated a TR DNM rate across all passing TR loci genome-wide of 4.74×10^{-6} per locus per haplotype per generation (95% CI = $4.06 - 5.43 \times 10^{-6}$), with substantial variation across repeat motif sizes (**Fig. 3a**). Collectively, TR DNMs inserted or deleted a mean of 978 bp per sample or 15.0 bp per event (Supplementary Table 11). An average of 54.9 mutations were expansions or contractions of STR motifs, 2.6 affected VNTR motifs, and 7.8 affected “complex” loci comprising both STR and VNTR motifs. The STR (1-6 bp motif) mutation rate was 5.50×10^{-6} *de novo* events per locus per haplotype per generation (95% CI = $5.0 - 6.04 \times 10^{-6}$). The VNTR (7+ bp motif) mutation rate was 0.83×10^{-6} (95% CI = $0.51 - 1.27 \times 10^{-6}$), predominantly comprising loci that could not be assessed in short-read studies. Several prior estimates of the genome-wide STR mutation rate only considered polymorphic STR loci; when we limited our analysis to STR loci that were polymorphic in the 1463 pedigree, we found 3.20×10^{-5} *de novo* events per locus per generation (95% CI = $2.74 - 3.66 \times 10^{-5}$), which is broadly consistent with prior estimates of $4.95 - 5.6 \times 10^{-5}$. TR DNMs were more common in the paternal germline; 75.0% of phased *de novo* TR alleles were paternal in origin (**Fig. 3b**).”

“...we were able to assess 80/613 (13.1%) of *de novo* alleles purely comprising STRs (average of 10 STRs per sample).”

“Of the 80 *de novo* STRs we could assess using Element sequencing reads, 56 (70.0%) passed our strict consistency criteria. The validation rate was lower at homopolymers (3/20; 15%) than at non-homopolymers (53/60; 88.3%), indicating that homopolymers still pose a challenge for long-read genotyping, and that our estimates of mutation rates at these loci may be less precise.” ~~TR DNMs that failed consistency analysis are significantly shorter than those that passed (Mann Whitney U-test for TR allele length change: $p = 1.84 \times 10^{-10}$) and are enriched for *de novo* expansions and contractions of 1 bp; TRGT is known to exhibit higher off-by-one genotyping error rates.~~

“Of the 128 *de novo* TR alleles observed in the two G3 individuals, 96 (75%) were transmitted to the next generation.”

Figure 3: Tandem repeat *de novo* mutations (TR DNMs) show motif size dependent mutation rates, paternal bias, and are highly recurrent at specific loci. a) TR DNM rates (mutations per haplotype, per locus, per generation) are displayed for each TR class (STR, VNTR, or complex) and as a function of the minimum motif size observed at each TR locus motif size in the T2T-CHM13 reference genome (blue; left y-axis). The average number of loci of each motif size that passed filtering criteria in each individual are displayed in gray (right y-axis). ~~Complex TR loci that comprise more than one unique motif were excluded.~~ Error bars denote 95% Poisson confidence intervals (computed using a chi-square distribution) around the mean mutation rate estimate. Mutation rates include all nonrecurrent calls that pass TRGT-denovo filtering criteria and Element consistency analysis ~~but are not adjusted for Element validation.~~

Discussion:

“...indels or SVs originating from TRs (65.3 *de novo*) ...”

7. It is exciting that the authors were able to provide a genome-wide analysis of *de novo* VNTR mutations, which have been hard to include in previous short read based datasets. I was surprised by Figure 3a, which shows mutation rates increase, pretty substantially, with the length of the VNTR repeat unit. As noted this is the opposite of STRs, in which generally longer repeat units have slower mutation rates. The mutation rates for the VNTRs get quite high (approaching 10^{-2}). Has this increasing trend been observed before? Although some potential biological explanations are given in the discussion, I am left wondering if there is any technical artifact that could cause this trend. E.g. are VNTRs with longer units harder to assemble accurately? Or are VNTRs with longer units closer to centromeres or other hard to call regions?

The reviewer raises a reasonable concern regarding the high VNTR mutation rates at large motif sizes. To our knowledge, genome-wide germline mutation rates at VNTRs, and the correlation between VNTR motif size and mutation rate, have not been well characterized. High mutation rates (between 3×10^{-3} and 5×10^{-2}) have been reported at a small number of human VNTR loci (Jeffreys et al. 1988), and in microbes, VNTR mutation rates have been estimated to range from 10^{-7} to 10^{-3} (Eslami Rasekh et al. 2021). We note that our STR and VNTR discovery was done using software tools (TRGT and TRGT-denovo) that operate directly on HiFi sequencing reads rather than assemblies, so the difficulties associated with assembling VNTRs should not impact these findings. However, to more explicitly adjudicate the VNTR mutations observed in G3, we performed two additional analyses and validation steps.

We inspected *de novo* VNTRs using both IGV and our own plotting tools and discovered 28 likely false positive calls out of 62 total VNTRs. We removed these 28 DNMs from our final callset and updated Figure 3a to reflect our filtered counts of VNTR DNMs. As noted in the manuscript, the current version Figure 3a omits “complex” TR loci that comprise more than one unique motif size (i.e., a locus with motif structure $ATT(n)G(n)$ would be excluded, but a locus

with motif structure AT(n)GT(n) would be included. As a result, many loci (~64% of DNMs) were omitted from Figure 3a. To address this, we instead plotted mutation rates as a function of the smallest motif size within a locus. This updated plot includes many more VNTR loci, and more clearly demonstrates a trend toward elevated mutation rates at loci comprising larger VNTR motifs. Our updated version of Figure 3a also shows the average number of reference loci (corresponding to each motif size on the x-axis) that passed our filtering criteria among the G3 individuals. We believe that this additional plot element (now shown as a gray line in Figure 3a) will help readers appreciate the fact that most TR loci are STRs and smaller VNTRs; and while high mutation rates are observed for certain very large VNTRs, these large VNTR loci are relatively infrequent in the human genome.

We also calculated the distances between VNTRs and annotated centromeres in the T2T-CHM13 reference assembly. DNMs at VNTRs were no closer than 10 Mbp to an annotated centromeric satellite sequence, as defined in the “hub_3671779_censat” table in the UCSC Genome Browser.

We thank the reviewer for their comment, which prompted us to do an even more detailed examination of both the VNTR and STR DNM characteristics. In Figure 3a, we still observe a trend of increasing mutation rate as a function of motif size in the VNTRs, as well as a mutation rate “trough” between STRs and VNTRs.

Revised Figure 3a and b.

Revised text:

“As shown in Figure 3a, we also estimated DNM rates as a function of the minimum motif size observed within a locus. For example, a locus with motif structure AT(n)AGA(n)T(n) would have a minimum motif size of 1. We counted the number of TR DNMs that occurred at loci with a minimum motif size of N and divided that count by the total number of TR loci with a minimum motif size of N that passed filtering thresholds. We then divided that rate by 2 to produce a mutation rate per locus, per haplotype, per generation. When calculating overall STR, VNTR, and complex mutation rates, we defined STR loci as loci in which all constituent motifs were between 1 and 6 bp; we defined VNTR loci as loci in which all motifs were larger than 6 bp; and we defined complex loci as loci in which there were both STR (1-6 bp) and VNTR (7+ bp)

motifs. For example, both an A(n) locus and an AT(n)AGA(n)T(n) locus would be classified as STRs, since they both purely contain STR motifs.”

8. Is there any bias for expansion vs. contraction mutations at the STRs, VNTRs, or other indels? Besides mutation rate vs. motif size, do the mutations recover the known relationship between total repeat length and mutation rate? Looking at that relationship for VNTRs may give insight into the previous point regarding the increasing trend between repeat unit length and mutation rate.

We plotted the relationship between mutation rate and total reference repeat length (measured in base pairs) for TR DNMs observed in G3 (note that the x-axis is truncated at a reference allele span of 100 bp, which includes ~95% of all reference loci). As expected, we find that the mutation rate is higher at loci with larger total repeat lengths in the reference.

We also plotted the distribution of likely expansion and contraction sizes at TR DNMs. We observe that both expansions and contractions follow similar distributions of event sizes (measured in base pairs) and note that the large number of 4 bp TR DNM events is almost exclusively driven by expansions and contractions at di- and tetra-nucleotide motifs.

We revised the text as follows:

“The mutation rate for dinucleotide motifs was higher than for homopolymers, and we observed increasing mutation rate with motif size for motifs greater than 6 bp in length (Fig. 3a). As reported in prior studies (Mitra et al. 2021), larger TR loci (defined as the total length of the TR locus in the reference genome sequence) exhibited higher mutation rates (**Supplementary Figure 36a**). We did not observe a bias toward expansions over contractions (**Supplementary Figure 36b**; binomial test $p = 0.97$).”

Supplementary Figure 36: Patterns of TR DNMs.

a) Larger reference loci exhibit higher TR mutation rates. We binned all ~7.8 million TR loci by each locus' size (in bins of 10 bp) in the reference genome. We then binned all TR DNMs by reference allele size in the same fashion and computed TR DNM rates (expressed per bin, per haplotype, per generation) within each 10 bp bin by dividing the observed number of TR DNMs in a given bin by the total number of TR loci of the same bin size. **b**) Expansions and contractions are observed at similar frequencies. We plotted the average fraction of TR DNMs ($\pm 95\%$ CI) of the specified size (in bp) across all G3 individuals. The plot is truncated at a maximum absolute DNM size of 25 bp.

9. Although the variant set included is already quite broad, could the authors comment on whether this data can be used to assess copy number variants (CNVs) (and de novo mutations affecting those?).

This is an interesting idea and one which we had not specifically considered outside of VNTRs. To address, we specifically focused on copy number polymorphic clusters previously defined at the population level based on read-depth analysis across the 2,504 1KG samples with respect to GRCh37 (Sudmant et al. 2015). First, we lifted over CNV coordinates to T2T-CHM13. We successfully lifted 3,073/3,263 CNVs, using the liftOver tool. We then attempted to extract the sequence of each CNV region from phased genome assemblies along with 50 kbp flanking sequence on each side. For regions ($n=3,047$) where both G2 parents were completely assembled in a single contig, we evaluated inheritance of each given region in G3 samples ($n=8$). We used the size of each region across haplotypes as a proxy of the copy number. We

then compared each child's haplotype to both parents and assigned it to the most likely parental haplotype based on size. In total, we assessed 3,047 CNV clusters out of which 2,772 (~91%) could be unambiguously assessed and show clear haplotype inheritance from G2 to at least four and more G3 samples. A small proportion of CNV regions (~9%) have more than half of the child's genotypes missing due to small differences between inherited alleles (**Supplementary Fig. 43a-b**). A large proportion of these can be explained by the marked assembly errors by Flagger (~75%), while the rest of the inconsistencies are likely caused by the sensitivity of our algorithm prototype. We note that our algorithm assigns child alleles to the most similar parental alleles based on the calculation of percentage difference between observed alleles in a child and all possible inherited parental alleles. As a consequence, smaller SVs are penalized more since even a small size change has a large effect on reported percentage difference. We show an example of a fully genotyped region in **Supplementary Fig. 43c** where the size of inherited alleles in children match the size of inherited parental alleles, as expected.

We report this analysis in the results section with all the details provided as Supplementary Note (see below):

“Some of the largest gains occur among SDs and the genes associated with them. In this analysis, for example, we classified 83.7% of the SDs (coverage >95%) as high-confidence regions compared to a previous GIAB analysis, which called variants only in 25.6% of these regions. These regions are frequently highly copy number variable in the human population¹⁹. We find that the majority (>91%) of known CNV regions were stably transmitted in this pedigree while the remaining 9% were often flagged as potentially misassembled (**Supplementary Note 2**).”

Supplementary Figure 43: Assembly-based genotyping of known CNV regions.

a) A barplot showing the number of known CNV loci (y-axis) being genotyped in a given number of G3 samples (n=8; x-axis). **b)** A barplot showing the percentage of known CNV locus (y-axis) being genotyped in a given number of G3 samples (n=8; x-axis). **c)** An example of a CNV loci on Chromosome 8 (GRCh38 coordinates) genotyped in G3 samples (n=8) by measuring the size of a given region in phased genome assemblies. The size of each parental allele in G2 is marked by letters A-B and assigned a unique color.

10. It is indicated that the sequence data and phased genome assemblies will be made available. It would be great if the variant calls will also be released.

All data, including assemblies, sequence data and variant calls, are being publicly released. The 23 family members have agreed to broad, open access and these are publicly available on cloud storage (<https://github.com/Platinum-Pedigree-Consortium/Platinum-Pedigree-Datasets>). Five individuals (three from G3 and two from G4) have agreed to have their data available but through restricted access and will be limited to dbGaP. Variant calls for all individuals will be available in dbGaP.

More specifically, the data release will include:

- Sequencing data
 - PacBio HiFi (aligned BAMs)
 - ONT (aligned BAMs)
 - Element (aligned BAMs)
 - Strand-seq (raw and aligned BAMs)
 - Illumina (for G3 spouses and G4, aligned BAMs)
- Assemblies (Verkko and hifiasm)
- Variant calls
 - SNVs and short indels
 - Tandem repeats (STRs and VNTRs)
 - Structural variants
 - Centromeric variants
 - Y chromosome variants

We added an updated “Data availability” statement to the manuscript as follows:

“Data availability

All underlying variant calls from 28 members of the family will be available in dbGaP under Accession ID: phs003793.v1.p1 (Platinum Pedigree Consortium long-read sequencing). In addition, raw sequencing data for five family members (G3-NA12883, G3-NA12884, G3-NA12887, G4-200103, G4-200105) that are not consented for open access will be also available via dbGaP.

Variant calls, raw sequencing data, and assemblies for 23 family members (G1-GM12889, G1-GM12890, G1-GM12891, G1-GM12892, G2-GM12877, G2-GM12878, G3-GM12879, G3-GM12881, G3-GM12882, G3-GM12885, G3-GM12886, G3-200080-spouse, G4-200081,, G4-200082, G4-200084, G4-200085, G4-200086, G4-200087, G3-200100-spouse, G4-200101, G4-200102, G4-200104, G4-200106) consented for their data to be publicly accessible similar to the 1000 Genomes Project samples to allow for development of new technologies, study of human variation, research on the biology of DNA, and study of health and disease.

Corresponding data and phased genome assemblies will be made available via the AWS Open Data program:

s3://platinum-pedigree-data/data.

See <https://github.com/Platinum-Pedigree-Consortium/Platinum-Pedigree-Datasets> for more details.

The Y-chromosomal assembly for a closely related R1b haplogroup sample HG00731 was downloaded from the Human Genome Structural Variation Consortium IGSR site (https://ftp.1000genomes.ebi.ac.uk/vol1/ftp/data_collections/HGSVC3/working/20230927_verkko_batch2/assemblies/HG00731/).

Reference genomes and their annotations used in this study are listed in **Supplementary Table 14**.

Code availability

Custom code and pipelines used in this study are publicly available via the following GitHub repositories:

<https://github.com/orgs/Platinum-Pedigree-Consortium/repositories>

Minor points:

1. Figure 1 legend “The pedigree dataset has been expanded...to include...G3 spouses (NA12879 and NA12886) - aren't the other datasets (200080 and 200100) the new ones? This made it sound like NA12879/NA12886 are new.

We agree with the reviewer that this was confusing so modified the Figure 1 legend as follows:

“The pedigree dataset has been expanded, for the first time, to include the fourth generation and G3 spouses (200080 and 200100).”

2. It would be helpful to briefly define the “auN” metric upon first use. Similarly for other metrics used to describe the assemblies - e.g. “quality value” and other jargon terms used throughout.

We now include a short glossary of the assembly metrics referenced in the ‘Evaluation of phased genome assemblies’ Methods section.

Added text to Supplementary Notes:

“8. Assembly quality terminology used in this study

Assembly quality value (QV) - QV is an assembly quality metric that reflects the number of k-mers present only in the assembly and not in short Illumina reads for a given sample. The proportion of assembly-only k-mers over the total number of k-mers in the assembly represents the error rate. This error rate is reported as a log-scaled probability of error for the consensus base calls (Phred quality score). For instance, Q30 corresponds to 99.90% assembly accuracy, Q40 to 99.99%, etc.

Misorientation - defined as a region in the assembly whose orientation is in reverse instead of direct orientation. Such errors can be easily seen using Strand-seq data, which has the capability to preserve the orientation of single-stranded DNA (Sanders et al. 2016).

Hamming error rate - represents total number of haplotype-phasing differences between a sample and a reference phasing.

Switch error rate - represents the number of haplotype-phasing switches between a sample and a reference phasing.

Assembly collapse - region in the *de novo* assembly where the assembly algorithm was not able to resolve the correct copy number of paralogous sequences.

Assembly misjoin - point in the *de novo* genome assembly where sequences from distinct parts of the genome are joined together.”

3. Line 234: “Some of the largest gains occur among SDs “ -> what are SDs? Should that be SVs?

Here we indeed meant to write SDs and not SVs. SDs is an acronym for segmental duplications, a term that has been first used in the introduction on line 78.

4. Line 245: what is a “morbid copy number variant”?

Morbid CNV is a term that was used as part of the morbidity map to define a region associated with instability and causing a disease if deleted and/or duplicated. We revised this to reflect that:

“This includes a rare inversion (~703 kbp) overlapping a disease-associated copy number variant region at Chromosome 15q25.2 20 (Supplementary Fig. 20) and an inverted duplication (~295 kbp) at Chromosome 16q11.2.”

5. Figure 2c mentions a strong paternal age effect but it also seems like there is a correlation with maternal age? It would be nice to annotate with Pearson r/p-value. If the maternal age effect is also significant it is worth pointing out.

We do observe a maternal age effect of 0.20 DNMs per additional year of maternal age, but it is not significant ($p=0.540$) given the sample size. We revised the figure legend to reflect this:

“**c**) A strong paternal age effect is observed for germline *de novo* SNVs but not for PZMs. There is no significant maternal age effect observed here for DNMs or PZMs.”

6. For Extended Data Fig. 3b: is there a significant correlation between the allele balance vs. fraction of children with the alternate allele for the PZMs?

We do find a significant correlation between allele balance and the fraction of children with the alternate allele using a linear model ($p=0.000462$). The slope of the resulting line is 0.867, which means that for every increase of 0.14 in AB, we expect to see transmission to one additional child.

We revised the legend to read:

“**b**) The mean allele balance (AB) of DNMs and PZMs across HiFi, Illumina, and ONT data plotted against the fraction of children who inherited a variant are significantly correlated for PZMs (two-sided t-test, $p=0.000462$), and about half of PZMs with $AB < 0.25$ are transmitted to at least one child.”

7. Line 277: “95% [of SNVs/indels] and 70% [of SVs], respectively, show evidence of transmission from G2-G3”. What is the expectation? It seems like this implies unsurprisingly that the SV calls are slightly less reliable?

The variant concordance test within the pedigree provides a level of detail that has proven

confusing for two-out-of-three reviewers. Every variant in the final truth set is pedigree consistent, and we further explore the technical aspects of our truth set in Kronenberg et al. (Kronenberg et al. 2024). To simplify the manuscript, and avoid confusion, we removed mention of the concordance rates of different variant classes (SNVs/indels/SVs).

We revised our text according to latest analyses and to improve clarity:

“We establish a variant truth set and identify a total of 5.95 million SNVs/indels and 35,662 SVs—all of which are pedigree consistent across the second and third generations (**Supplementary Table 5**) (Kronenberg et al. 2024). Of our 5.95 million small variants, 77% are supported by all three technologies and callers.”

8. Are the PRDM9 genotypes used for anything? It is mentioned briefly (line 295+) but there is no rationale given for why those were computed (to look for correlation with recombination patterns) and that section ends abruptly.

We agree with the reviewer that this partial result seems to be disconnected. We decided to remove this part of the manuscript in order to improve flow and clarity of this manuscript.

Text removed from manuscript:

~~Last, we characterized the PRDM9 genotypes for all individuals in the pedigree (Supplementary Table 10), comparing the results obtained from Verkko and hifiasm (UL) assemblies across the G1-G3 samples. Across the entire family, we define the alleles A, B, M10, and M19—all four from the PRDM9-A-type predicted binding site group.~~

9. Figure 2b: Can the difference in allelic balance for DNMs vs. PZMs be explained by thresholds for detection? E.g. a very common PZM might be hard to distinguish between germline DNM and by definition have to be lower.

The difference in allele balance (AB) between DNMs and PZMs certainly explains why PZMs are often omitted from *de novo* studies, where a standard detection limit is $AB > 0.25$, but in this study, we do not have a minimum AB required to discover a variant.

In order to distinguish between DNMs and PZMs, we do take AB into account, but we weight the mutational haplotype much more highly. In fact, there are some PZMs that have AB close to 0.5, but we are sure they are not germline in origin because they are not present on every read from a parental haplotype. However, because we expect half of all reads to be derived from a single parent, and a PZM to be present on a subset of those reads, PZMs should definitionally have a lower AB than germline mutations.

10. For the intragenerational recurrent mutations - although it is mentioned in discussion it might be worth mentioning in the main text that this could be due to parental mosaicism.

We added to the text the possibility of parental germline mosaicism:

“After both strict filtering and visual inspection of reads (**Methods**), we identified a high-confidence set of 32 loci (**Supplementary Table 11**): five showing intragenerational recurrence (observed DNMs in at least two G3 individuals) and 27 loci with intergenerational recurrence (observed DNMs in at least two generations). Because they are only observed in a single generation, the intragenerational DNMs may represent mosaicism in the parental germline, rather than recurrent mutational events.”

11. It would be helpful to mention in the methods if the Illumina data was PCR-free? Does the G4 data differ in either pcr+/- or read length compared to the G1-G3 data?

All of the Illumina sequencing was PCR-free and 150 bp paired-end.

Revised text:

“Illumina data generation

Illumina WGS on G1-G3 was generated as previously described. Illumina WGS on G4 and marry-in spouses for G3 were generated by the Northwest Genomics Center using the PCR-free TruSeq library prep kit and sequenced to approximately 30x on the NovaSeq 6000 with paired end 150 bp reads.”

12. Were DNM Y chromosome events similarly assessed for transmission to the next generation as the autosomal events were? I wasn't sure after reading the chrY section.

All Y-chromosomal DNMs were assessed for transmission to the next generation and concordance with the expected transmission through generations was confirmed for 100% of applicable DNMs. Specifically, all *de novo* SNVs (n=10) and indels (n=3) identified in G2-NA12877 were present in the assemblies of his four sons (G3: NA12882, NA12883, NA12884 and NA12886) and in the PacBio HiFi read data of his three grandsons (G4: 200101, 200102 and 200105). Similarly, *de novo* SNVs (n=10) and indels (n=3) identified in G3-NA12886 were present in the PacBio HiFi read data of his three sons (G4: 200101, 200102 and 200105). *De novo* SVs (n=3) from G2-NA12877 were present in the assemblies of his four sons (G3: NA12882, NA12883, NA12884 and NA12886). While the detection of SVs from the HiFi reads alone in the highly repetitive Yq12 region is challenging, the HiFi read data from the three G4 males (200101, 200102 and 200105) supported the presence of the *de novo* SVs identified from G2-NA12877 (n=3) and G3-NA12886 (n=1) (**Fig. 5b**).

We made the following changes to the Results section to make this clearer to the reader:

“The latter range from 2,416 to 4,839 bp in size, each affecting an entire *DYZ2* repeat unit(s), with an average of one SV per Y transmission. All applicable DNMs (SNVs: n=20/48, indels: n=6/9, SVs: n=4/5) are concordant with the expected transmission through generations (i.e., from G2 to G3-G4 and from G3-NA12866 to his three male descendants in G4) (**Fig. 5b**). Overall, 83% (52/63) of the DNMs ...”

REFEREE #2:

The characterization of mutations and recombinations in families via high quality assembled genomes is novel. These various sequencing technologies used by the authors enabled them to search for mutations in centromeres which are notoriously hard to analyze due to their repeat structure. The accumulation of mutations in the germline in the unique parts of genome are better characterized with much larger cohorts of trios and short reads, this study adds little to that characterization.

We thank the reviewer for taking the time to provide detailed feedback on our paper. The referee is correct that our results in the unique portions of the genome are generally consistent with much larger studies and the novelty of our work is access to the repetitive regions of the genome where we demonstrate that mutation rate (including SV *de novo* rate) is significantly greater.

Major points:

1. The authors are detecting mutations in repeated regions which is a formidable task, but I was hoping for a more nuanced analysis of the accumulation of mutations in repeated regions of the genome.

While our study provides some of the first family-based rates of mutation in VNTRs, centromeres, Y chromosome satellites, and segmental duplications, as pointed out by the referee, we are somewhat limited in our ability to analyze more subtle differences in substitution patterns between repeated and unique regions of the genome.

To compare the mutation spectra, we subset our DNMs into two groups by intersecting them with different genomic annotations. The first group is composed of 76 SNVs in SDs (n=52 DNMs, n=24 PZMs over 227.3 Mbp). To make the second group, we took our initial set of *de novo* SNVs and removed all SNVs in SDs, centromeres, acrocentric p-arms, simple repeats, and transposable elements (Alus, LINEs, and SINEs), leaving us with a total of 376 SNVs (n=314 DNMs, n=62 PZMs over 1.65 Gbp) mapping to truly unique regions of the genome. When we use chi-square tests to compare the single-base substitution spectrum between SNVs in SDs and SNVs in unique regions, no class of mutation rises to statistical significance. The largest difference we observe, however, is a depletion of *de novo* CpG>TpG mutations in SDs, but it is not statistically significant (chi-squared test, p=0.143). We do, however, observe a significant decrease in the *de novo* transition/transversion ratio in SDs compared to all DNM calls (chi-squared test, p=0.0109) and compared to unique DNM calls (chi-squared test, p=0.012), which is consistent with a recent population genetic-based analysis (Vollger et al. 2023). This decrease in Ti/Tv is not observed for LINE repeats (**Supplementary Fig. 51**).

Supplementary Figure 51: Mutation spectra in unique and repeated regions.

We examined all autosomal DNMs and PZMs, as well as subsets in LINEs, segmental duplications (SDs) and unique regions (autosomes without centromeres, acrocentric p-arms, SDs, transposable elements, or simple repeats) to compare the single-nucleotide substitution spectra. There were no significant enrichments of any mutational class in SDs or LINEs compared to unique sequence, but the transition/transversion (Ti/Tv) ratio is significantly lower in SDs compared to unique sequence (chi-squared test, $p=0.012$).

We added the analysis above to the Supplementary Note and added the results to the main text:

“In particular, we observe elevated rates of *de novo* SNVs in repetitive regions both for germline and postzygotic events, consistent with recent human population-based analyses^{7,52} and theoretical predictions⁵³. SD regions show an 88% increase (2.2 [95% CI: $1.53 - 2.86 \times 10^{-8}$] vs. 1.17×10^{-7}), which is driven by SDs with >95% identity. We also observe a significant decrease in the *de novo* transition/transversion ratio in SDs compared to all DNM calls (chi-squared test, $p=0.0109$) consistent with previous expectations⁷ (**Supplementary Note 7**).”

2. The *de novo* assembly of the genomes seems to be of high quality and state of the art, however, many of the *de novo* mutation analysis lack appropriate quality control and scientific context. For example, is there any correction done for the difference in false positives/negative rates of DNM calls between unique and repeated regions?

We agree with the reviewer that the quality control of both the *de novo* assemblies and the derived DNM calls is crucial. For this reason we took great care to QC both the assemblies and the DNM calls. We include a Supplementary Note to summarize QC steps taken to validate various DNM classes (see below).

We added the following text as a **Supplemental Note 9**:

“Evaluation of *de novo* SNV mutations

Autosomal SNVs: All autosomal *de novo* SNVs identified using alignment methods were validated using the same strategy regardless of genomic region, with the exception of SNVs in TRs, for which we applied a minimum allele balance of 0.05. To validate a DNM call, we examined read data from three orthogonal sequencing technologies: HiFi, ONT, and Illumina. For a given sequencing technology, we examined reads spanning a variant from every sample in the pedigree. Omitting any reads that fell below our mapping quality threshold (59 in HiFi and ONT, 0 in Illumina) or base-quality threshold (10 at the site of the mutation), we counted the number of reads that had the alternate allele, weighting alleles with a base quality >20 (high quality) more than reads with base quality <20 (low quality). Considering each sequencing technology individually, we determined that a variant was truly *de novo* if it was present in a child and absent from its parents, and inherited if a parent had reads with the alternate allele. For HiFi and Illumina, parents were required to have zero high-quality reads with the alternate allele, or up to one low-quality read. Because ONT has a higher error rate, we allowed a parent to have one high-quality read with the alternate allele, or up to two low-quality reads. We combined validations across technologies, determining that a variant was inherited if it looked inherited in any technology. We also excluded any variants that were observed in at least one high-quality HiFi read in any sample of the pedigree not directly descended from the *de novo* sample. True *de novo* events were required to be supported across at least two technologies (n=16 supported only by HiFi and Illumina; n=11 supported only by HiFi and ONT), but the majority of our final SNV callset (n=728/755) were supported by all three.

STRs/VNTRs (<50 bp): To assess candidate STR and VNTR DNMs, we took the following approach. Beyond the basic filtering criteria reported in the Methods section, which eliminated likely false positive DNMs, we validated a subset of DNMs based on a) transmission to a subsequent generation, b) *de novo* allele consistency with an orthogonal sequencing technology, and c) manual inspection using IGV and our own plotting utilities (available at the GitHub repository associated with the manuscript). For criterion a), we asked if candidate DNMs observed in two of the G3 individuals (2189 and 2216) were observed in those individuals' G4 children. If at least one of the G4 children inherited a TR allele matching the *de novo* allele observed in the focal G3 individual, we considered the DNM to be validated by transmission. Importantly, we could only apply validation criterion a) to two of the eight G3 individuals. For criterion b), we asked if candidate STR DNMs with allele lengths ≤120 bp were supported by sequencing evidence from an orthogonal technology (Element AVITI). Details of our orthogonal validation approach can be found in the Methods section (“Measuring concordance with orthogonal sequencing technology”). For the final criterion c), we manually inspected candidate VNTR DNMs using both IGV and our own plotting utilities; the latter were used to examine the distribution of CIGAR (insertion and deletion) operations in sequencing reads aligned to candidate DNM loci in all members of a given trio. Candidate VNTR DNMs that were flagged as false positives after manual inspection were removed from the final callset.

Centromeric SNVs: In order to validate centromeric SNVs, we started with a raw SNV callset of 1,789 variant calls generated by parsing CIGAR strings of child-to-parent alignments using the SVbyEye function 'cigar2ranges' (Porubsky et al. 2024). For each variant, we inferred the reference and alternate alleles by examining its location both in the assembly of the child, and in the parent who transmitted the centromere to the child. We aligned both parent and child HiFi and ONT reads to the child's assembly and counted the number of reads with either the reference or alternate allele. Evaluating each technology separately, we considered a variant to be true in HiFi if the child had at least one read (with any mapping quality and base quality) with the alternate allele, and the parent had no reads with the alternate allele. In ONT, we allowed the parent to have up to one read with the alternate allele, because ONT data are more error prone. Once we had evaluated a variant with each type of read data, we combined the results, determining that a variant was true if it had support in both platforms, false positive if it was not supported, and inherited if a parent appeared to have the alternate allele in either HiFi or ONT read data.

Chromosome Y SNVs/indels: The SNVs and indels across the MSY (i.e., excluding the PARs) were called using the G1-NA12889 Y assembly as a reference and all types of variants were then called from the G2 and G3 Y assemblies using Dipcall (Methods). Across the one G2 and four G3 males, Dipcall called a total of 5,609 candidate SNVs and 5,083 indels. The vast majority of the candidate variants (4,927 SNVs and 3,683 indels) were called from G3-NA12886, which has the lowest quality assembly, while for the other samples the number of candidate calls ranged from 102 to 274 (mean of 170 per Y assembly) for SNVs and from 299 to 470 (mean of 350 per Y assembly) for indels, respectively. In order to obtain a confident set of SNVs and indels, the following steps were taken: i) The *de novo* assemblies were assessed for misassemblies using Flagger and NucFreq. Any DNMs overlapping with flagged regions by both or either of the tools were filtered out. ii) For SNVs, the final filtered calls were supported by 100% of HiFi reads (i.e., no reads supported the reference allele in offspring or alternative allele in the father) and ONT reads mapped to both the reference and each individual assembly were checked for support. For indels (≤ 50 bp), homopolymer tracts were excluded from the analysis, while the rest of the calls were validated using the read data (HiFi, ONT, Illumina) as follows: Individual reads mapped to the reference (G1-NA12889 Y assembly) and covering the indel call plus 150 bp of flanking sequence were extracted from all samples using subseq (<https://github.com/EichlerLab/subseq>), followed by alignment using MAFFT. All alignments were manually checked and any calls where the HiFi data had two or more reads supporting a reference allele and one or more reads supporting an alternate allele were removed. All final SNV and indel calls were additionally supported (if unique mapping to the region was possible) by both Illumina and Element read data mapped to the reference. iii) For all applicable SNVs and indels (i.e., if male offspring was available for the male where the call was made), 100% concordance with the expected transmission through generations was confirmed. Specifically, all *de novo* SNVs (n=10) and indels (n=3) identified in G2-NA12877 were present in the assemblies of his four sons (G3: NA12882, NA12883, NA12884 and NA12886) and supported by the PacBio HiFi read data of his three grandsons (G4: 200101, 200102 and 200105). Similarly, SNVs (n=10) and indels (n=3) identified in G3-NA12886 were supported by the PacBio HiFi read data of his three sons (G4: 200101, 200102 and 200105). iv) Additionally, *de*

novo SNVs and indels were independently called from the reads mapped to a reference. For this, the HiFi, ONT and Illumina data were mapped to the G1-NA12889 chromosome Y assembly followed by variant calling using GATK in haploid mode on the aligned HiFi data. Each male was directly compared to his father, selecting variants unique to the son. The SNVs and indels were then validated by examining the father’s HiFi, ONT, and Illumina data, excluding any variants present in the parental reads. The final read-based callset was 100% concordant with those called from the assemblies.”

3. Further, in the calculation of mutation rates, unique parts of the genome are generally used to derive a denominator but in the case of repeats this assumption breaks down, could the authors expand on this?

To calculate the mutation rate for the alignment-based SNV calls, we estimate the size of the denominator by examining HiFi data aligned to every base in the reference genome. If a base has at least one HiFi read that unambiguously aligns from an individual and both its parents, we consider that coordinate to be accessible in that individual. Because we require read data to map unambiguously to a site in order for it to be “accessible,” we would not expect to misrepresent the number of callable bases in repetitive regions because those regions would not be called and therefore not accessed for mutation.

We revised the Methods section to make this more explicit:

“To determine where we were able to identify *de novo* variation in the genome, we assessed HiFi data for every trio. We first used GATK HaplotypeCaller v4.3.0.0 with the option “ERC_BP_RESOLUTION” to generate a genotype call at every site in the genome. Only sites where both parents were genotyped as homozygous reference (0/0) were considered callable, as sites with a parental alternate allele were excluded from our *de novo* discovery pipeline. We then examined the HiFi reads from a sample and its parents, restricting to only primary alignments with mapping quality of at least 59. For children, we only considered HiFi reads derived from blood, but we considered blood and cell line data for parents. We counted the number of reads with a minimum base quality score of 20 at every site in the genome, and then combined this information with our variant calls. A site was deemed callable if both parents and the child each had at least one high-quality read with a high-quality base call. We observed an average of 2.67 Gbp of accessible sequence across the autosomes (out of 2.90 Gbp total, standard deviation = 24.9 Mbp). For female children, callable Chromosome X was determined the same way, whereas for the male children, we only considered the mother’s HiFi data when examining the X chromosome and the father’s HiFi data when examining the Y chromosome. In addition, male sex chromosomes were not restricted to sites where both parents were genotyped as reference—each parent was allowed to carry an alternate allele.

We calculated the germline autosomal mutation rate for every sample by dividing the number of germline autosomal DNMs by twice the number of base pairs we determined to be callable. For postzygotic mutations, we used the same denominator. In females, the amount of callable sex chromosomes was defined as twice the number of callable bases on the X chromosome, and in males it was defined as the sum of the callable bases on the X and Y

chromosomes. For each feature-specific mutation rate (such as SDs), we intersected both a sample's *de novo* SNVs and the sample's callable regions with coordinates of the relevant feature. We then calculated the mutation rate by dividing the number of SNVs in the region by the amount of callable genomic sequence where alignments could be reliably made."

4. The authors are limited by the number of independent trios in this study, as all their analysis are based on single extended family resulting in anti-conservative statistical tests. The authors should use all the available individuals in analysis of DNMs and recombinations, I value the effort made by the authors to differentiate between the generations.

This is a fair point. Unfortunately, we do not have cell lines for G4, thus we were not able to generate UL-ONT data to create T2T assemblies. Therefore, assembly-based analyses for variant discovery were limited to G1-G3. In our initial experimental design we instead used G4 samples to confirm transmission of detected DNMs for two samples of G3 (NA12879 and NA12886) and therefore limit type I error. We have, however, generated additional Strand-seq data for the spouses of G3 individuals. With this data we were able to fully phase these individuals and, in turn, map recombination breakpoints in both G4 subfamilies (n=12). (For more details see the response below.)

In response, we revised the sentence in the discussion to read:

"Third, we limited DNM discovery to the first three generations of only one multigenerational family and used G4 for validation purposes of transmitted variants. We acknowledge that familial variation depends on the genetic background (Rahbari et al. 2016; Sasani et al. 2019; Steely et al. 2022) and, thus, many more families will be required to establish a reliable estimate of the mutation rate, especially for complex regions of the genome."

5. The decrease in the number of crossovers with parental age is extremely strong and unexpected. The family members in the previous generation determine the detection of the crossovers, therefore the p-values in this recombination analysis are anti-conservative as the analysis is based on a single extended family. These results are going against the grain and are incompatible with current recombination results based on sequence level information (PMID: 30679340), where there is a weak positive age effect for crossovers in the maternal germline and non-significant age-effect in the paternal one. The authors see a strong decrease in number of crossovers in both germlines despite the recombination processes in males and females are wildly different. This result raises questions like, are the assembly metrics of the phased genome assemblies correlated with number of detected crossovers? Further, what would be biological mechanisms to decrease the number of crossovers with age in both germlines?

We agree with the reviewer that our observation is rather unexpected and might be private to this family only since we did not explore other pedigrees to record such a pattern. As a result, we extended our recombination map for the two G4 families. For this we generated additional Strand-seq data for both G3 spouses as well as 10 G4 samples. We are missing two G4 samples for the second family due to cell culturing issues. With this data we were able to fully

phase G3 spouses, which allowed us to map recombination breakpoints in G4 samples phased using parental Illumina k-mers. In total, we detected 825 of recombination breakpoints across all G4 samples (with respect to T2T-CHM13). We now report these in **Supplementary Table 8**.

When analyzing the correlation between the number of recombination breakpoints and the parental age, we do not observe such a strong correlation as we originally observed in G2 and G3 individuals. Given this, we now report this correlation for G2-G3 and G4 families and avoid making a conclusion that this is a general trend in the human population. We, thus, revised the text and updated the Methods to reflect this new analysis as follows:

“Strand-seq analysis of G1 parents as well as G3 spouses (200080 and 200100) allows us to phase and determine parent-of-origin for G2 and G4 chromosomes adding 964 breakpoints²⁸. In total across the three generations (G2-G4; 22 transmissions), we identify 1,503 meiotic breakpoints (with respect to T2T-CHM13) (**Supplementary Fig. 26**), including 16 recombination “hotspots”, 11 of which are in line with previously reported increased recombination rates²⁹ (**Supplementary Table 8, Supplementary Fig. 27**).”

Supplementary Figure 26: Male and female recombination breakpoints per sample.

A barplot showing the total number of recombination breakpoints detected in each sample (G2-G4) colored by inherited homologs (red colors - maternal, blue colors - paternal). There are a total of 1,503 and 1,479 breakpoints with respect to T2T-CHM13 (top) and GRCh38 (bottom) reference genome, respectively (**Supplementary Table 8**).

“We observe a significant decrease in crossover events with advancing parental age for both male ($R = -0.86$; $p = 0.0014$; Pearson correlation) and female ($R = -0.66$; $p = 0.039$; Pearson correlation) germ lines (Extended Data Fig. 2e). However, this is not observed in two G4 subfamilies suggesting that this may be a family-specific feature (**Supplementary Fig. 31**).”

Supplementary figure 31: Recombination breakpoints and parental age.

Correlation between the number of recombination breaks (y-axis) and parental age (x-axis) shown separately for maternal (red) and paternal (blue) recombination breakpoints for two G4 subfamilies (G4 family 1 and G4 family 2).

6. The manuscript reads like that sections of the manuscript were written by independent authors and loosely coupled together; it would be beneficial to harmonize the reporting of false positives/negatives in the manuscript across the different variant types. Further, the manuscript is too long in its current form as the authors are repeating themselves. This leads to the fact that the interesting aspects of the manuscript (DNMs in the repeated regions) are overshadowed by lengthy technical text not approachable by the general scientist.

We significantly revised the main text—moving some of the technical aspects to the supplement and focusing on the more interesting biological observations. With this, we removed ~500 words from the results section. In addition, we worked to smooth some of the transitions between the sections. As a result, the manuscript is now reduced in length and should present a more harmonized and streamlined summary of the results.

Specific points:

L153: What does AuN stand for?

AuN is an estimate of assembly contiguity similar to “average” contig length measured by the contig N50 value—the contig length such that at least 50% of the assembly is assembled in contigs of this and longer length. AuN further takes into account a continuous distribution of contig lengths while N50 is single point in this distribution. As a result, AuN is less susceptible to outliers in contig length and represents a more reliable metric. More details about assembly contiguity measures can be found here: <https://lh3.github.io/2020/04/08/a-new-metric-on-assembly-contiguity>

We added the above explanation along with the reference to **Supplementary Note 8**.

L174: There is no e panel in figure 2.

We thank the reviewer for spotting this error. Indeed, this figure reference should refer to Extended data figure 1e.

We revised the text accordingly:

“Thus, by merging complete centromeres generated by both assemblers, we create a nonredundant list of 288 completely and accurately assembled centromeres (**Extended Data Fig. 1e**).”

L177: Need to explain the scale of the quality value and its interpretation to the general reader. This metric is okay for evaluating an assembly but of limited value for QC of DNMs as it suggests that roughly 3,000 false positives would be detected assuming QV of 60.

For calling DNMs from aligned reads, we do use one quality metric as an initial filter for variant calls made by GATK and DeepVariant. We remove any variant calls where the genotype quality (GQ) is <20. GQ works a little differently than traditional Phred-scaled quality scores, as it reports the difference in likelihood between the most likely and the second most likely genotype, but a GQ of 20 or greater means we are 99% sure we have the correct genotype assigned at a site. We check GQ for both parents and a child before selecting a variant for further validation. This genotype filter helps to remove about 25% of initial *de novo* candidates; looking at sample NA12887 as an example, it reduces the initial *de novo* callset from 32,633 SNVs to 24,708. This remaining callset is still full of false positives and is eventually whittled down to just 79 SNVs after subsequent validation steps. Our final validations are based on evaluating sequencing reads from three orthogonal technologies and confirming the presence of a *de novo* allele in the child and its absence in the parents.

L179: Is the misorientation rate on the contig level or base level?

The misorientation rate is reported at a base-pair level. It is reported as a size of regions genotyped by Strand-seq as misoriented with respect to genotyped regions per diploid genome. The reported value is the largest detected fraction of misoriented bases in G3 individuals.

To make this clearer, we modified the text as follows:

“We estimate the accuracy of the Verkko assemblies at quality value 54 on average (range: 47-58) (**Supplementary Fig. 5**). Using Strand-seq data confirms a low misorientation rate (<0.022% of bases) (**Supplementary Fig. 6**) and a high phasing accuracy with Hamming error rates <2% (**Supplementary Fig. 7**).”

L180: Is this for a pair of heterozygotic genotypes?

Here the Hamming error rate is calculated across all heterozygous variants for each chromosome. A graphical explanation of Hamming error rates from our previous publication:

Graphics taken from (Porubsky et al. 2017).

We added an explanation of this measure to the Supplemental Notes.

Extended figure 1d: This presentation of the assemblies' metrics could be improved.

We followed the reviewer's advice and prepared a simplified version of Extended figure 1d (see below). We updated the figure with this new panel. The updated figure provides the birds-eye view of the most important aspects, while the more specific metrics have been retained in Supplementary figure 3.

L227-228: The parent pair in G2 has 8 offspring and there are two individuals in G3 that have 6 offspring. Are the authors referring to a transmission to at least one offspring of a parent pair or

per trio? Given that the authors report a transmission rate of 95% it appears to be the former. In an ideal world you expect the probability of transmission to at least one child, to be $1 - 0.56 \approx 98.4\%$. This indicates that the SNV and SV truth sets are of low quality. It would be beneficial to describe to the general reader what to expect and the observed deficit in transmission. The qc metrics employed by the authors could be improved.

Thank you for highlighting a subtle and potentially confusing point. For this section we are only considering the G2 and G3 members and require that all genotypes of any variant must be consistent with the full transmission in all 10 individuals (2 in G2 and 8 in G3). This is consistent with a previous study by Eberle et al. (2017) and is described in greater detail in a subsequent paper now available on bioRxiv (Kronenberg et al. 2024). Thus, by definition, the transmission rate is 100% for these truth variants. This is a stringent requirement and many true variants are excluded by a single genotype error in one of the samples. For example, if the genotyping accuracy were 0.95 for a variant type, we would expect just 60% of the variants to pass this criterion (i.e., $0.95^{10} = 0.60$). Since several reviewers find this sentence/point confusing, we removed it and refer readers to our follow-up paper, where we dive into the analysis.

Revised:

“We establish a variant truth set and identify a total of 5.95 million SNVs/indels and 35,662 SVs—all of which are pedigree consistent across the second and third generations (**Supplementary Table 5, Supplementary Fig. 13, Methods, Data Availability**) (Kronenberg et al. 2024). Of the 5.95 million, 77% of small variants are supported by all three technologies and callers.”

L238: Missing transmission rate per trio.

We calculated the Mendelian error rate on a trio-by-trio basis from the raw small variant calls (DeepVariant). See table below, which was added to **Supplementary Table 5**:

G3 Child	Consistent variant #	Inconsistent variant #	Transmission %
NA12879	6538561	128315	98.1%
NA12881	6545740	134760	97.9%
NA12882	6558222	117957	98.2%
NA12883	6553508	117388	98.2%
NA12884	6558104	116644	98.3%
NA12885	6560770	126979	98.1%
NA12886	6530414	126628	98.1%
NA12887	6552445	120183	98.2%

We further reviewed each non-reference MEI using IGV on a trio-by-trio basis. These variants were supported with reads in IGV and were 100% consistent with Mendelian inheritance. Based on this, we slightly revised our MEI counts (see below):

“We identify 2,161 *Alu* insertions, 398 LINE-1 insertions, and 149 SINE-VNTR-*Alu* (SVA) retrotransposon insertions. Only *Alu* elements >260 bp are shown here. We identify 112 LINE-1 insertions of either full-length or near full-length (at least 5,500 bp) and 123 SVA insertions of at least 2,000 bp (**Supplementary Table 6**).”

L243: Missing transmission rate per trio.

We did not find any evidence of a simple inversion to be of *de novo* origin using both computational methods and visual inspection. Inversions still remain challenging to fully phase and genotype, however, for sites which we were able to assess we observe a high transmission rates. Now reported in Supplementary Table 7:

sample.id	n.total.sites	n.evaluated.sites	transmitted.sites	transmission.rate
NA12879	120	92	90	97.8
NA12881	120	95	95	100
NA12882	120	89	88	98.9
NA12883	120	92	89	96.7
NA12884	120	90	88	97.8
NA12885	120	98	96	98
NA12886	120	93	92	98.9
NA12887	120	100	98	98

We make this clearer in the revised text:

“Similarly, we provide a comprehensive census of mobile element insertions, including full-length elements capable of retrotransposition (**Supplementary Table 6, Supplementary Fig. 14, Supplementary Note 3**), and use Strand-seq data to identify 120 inversions segregating in a Mendelian fashion (21 were ambiguous) (**Supplementary Table 7, Supplementary Figs. 15-18, Methods**).”

L254: What is the correlation between the number of contigs and the number of crossovers? And other assembly metrics such as AuN?

We did not observe any obvious correlation between assembly contiguity and the number of reported recombination events (G2-G4). This further supports that our detected recombination breakpoints are not due to the breaks in the assembly.

Revision figure

Left: A scatterplot showing the relationship between number of recombination breakpoints (y-axis) and total number of contigs per phased assembly. **Right:** Shows relationship between number of recombination breakpoints (y-axis) and AuN statistics per phased assembly.

L258: Why did the authors exclude the G4 individuals from this analysis? Especially in the light of limited sample sizes.

To address this reviewer question, we decided to generate additional Strand-seq data for the spouses of G3 individuals as well as 10 samples for G4. We note that Strand-seq data for two G4 samples are missing due to cell line issues. Having Strand-seq data for G3 spouses allows us to fully phase these genomes, which is critical to detect meiotic recombination breakpoints in G4 samples.

See also response on correlation between number of crossovers versus parental age.

L269: Depending on the genetic length of the chromosome?

Given the extended recombination map to G4 individuals, we observe a mild correlation between the number of nonrecombinants and the chromosome size. As expected, the number of nonrecombinant alleles increases with decreasing chromosome length.

Revised text:

“We find that 15-20% of paternal and maternal homologs are transmitted without a detectable meiotic breakpoint (i.e., nonrecombinant chromosomes) while the remainder (80-85%) contain at least one recombination breakpoint (**Supplementary Fig. 28**).”

Supplementary Figure 28: Summary of observed recombinant and non-recombinant parental alleles.

a) Barplot showing the counts of nonrecombinant homologs per chromosome and across the complete recombination map of G2-G4. b) Fraction of recombinant and nonrecombinant homologs across G2-G4 individuals. Two G4 subfamilies are marked (family1 and family2). Note: Male chromosome X is not considered here.

L275-L284: The tests are probably anti-conservative as these tests are mainly based on offspring (G3) from single parent pair (G2).

To address this, we extended our available recombination maps by two G4 families. With such extended datasets we observed the nearly identical recombination ratios as well as their enrichment at the chromosome ends in males as was expected based on the previous research. We, however, do not confirm the decreasing number of crossovers with the increasing parental age. We reflect these observations in the revised text:

“In line with previous research, we observe a significant excess (two-sided t-test, $p=6 \times 10^{-6}$) of maternal recombination events with expected 1.4 maternal to paternal breakpoint ratio with chromosomes 8 and 10 showing the most significant maternal excess (z -score > 2.3 ; $p < 0.02$) (Supplementary Fig. 29). Paternal recombination is significantly biased towards the ends of human chromosomes with 55 paternal recombination events mapping within 2 Mbp of the

telomere compared to one in females creating a bimodal paternal distribution of inherited segment lengths (**Extended Data Fig. 2c, Supplementary Fig. 30, Methods**). We observe a significant decrease in crossover events with advancing parental age for both male ($R = -0.86$; $p = 0.0014$; Pearson correlation) and female ($R = -0.66$; $p = 0.039$; Pearson correlation) germ lines in G3 (**Extended Data Fig. 2d**). However, this is not observed in two G4 subfamilies suggesting that this may be a family-specific feature (**Supplementary Fig. 31**).”

Original Supplementary Figure 29:

Revised Supplementary Figure 29:

Revised Supplementary Figure 30:

L289: Is the difference significant? Especially when considering the trio as the sample unit?

We had a closer look at the distribution of refined recombination breakpoints in G3 (n=487). As reported in the main text, we were able to lower the median breakpoint resolution from 3,367 bp to 2,467 bp. However, this difference is not significant. This is because there are only about the same number of intervals with reduced (n=248) and expanded recombination intervals (n=191) as a result of T2T-CHM13 reference bias. The T2T-CHM13 reference is not able to take into account regions of homozygosity between inherited parental haplotypes.

Revised text:

“We initially narrowed G3 recombination breakpoints to ~3.4 kbp. We further refined 90.4% (487/539) of the recombination events to a median size of ~2.5 kbp using direct comparisons between phased genome assemblies of a parent and a child (**Supplementary Fig. 22, Methods**).”

Revision figure: Comparison of reference- and family-based recombination breakpoint intervals.

A) Summary of refined recombination breakpoints (n=487) using phased genome assemblies and multiple sequence alignment (MSA) between sequence extracted from the parental and inherited homolog in the child as opposed to breakpoints defined with respect to a single reference (REF: T2T-CHM13). **B**) The same distribution as in (A) showing each recombination breakpoint range as a dot. Result of the T-test (two-sided) p-value is shown on top. **C**) The same distribution as in (B) but shown here per G3 sample (n=8). Result of the T-test (two-sided) p-value is shown on top.

L320: What is the correlation between the AB of DNM using Illumina and Element? How many DNMs/PZM showed a significant AB difference between methods?

To begin, we first checked to make sure all 755 of our DNMs and PZMs were supported by Element data (examining reads with mapping quality >30, with base quality >20). There were 14 variants in locations where no Element reads could be aligned. Of the remaining 741 variants, 99.2% of DNMs (n=616/621) and 91.7% of PZMs (n=110/120) were supported in Element data. In order to test whether AB was concordant between Illumina and Element data, for every variant we performed a simple chi-square test; if the p-value was <0.05, we determined that AB was significantly different between platforms. We found a significant difference between Illumina and Element AB for 6.9% of DNMs (n=43) and 5.8% of PZMs (n=7).

We also repeated this analysis to compare Element and HiFi data and found no significant difference in AB between these two platforms for 4.0% of DNMs (n=25) and 7.5% of PZMs (n=9).

We revised the text as follows:

“By this criterion, we discover 755 *de novo* SNVs and 73 *de novo* indels across the autosomes of 10 individuals (n=2 G2; n=8 G3 individuals, Fig. 2a), as well as 27 *de novo* SNVs and 1 indel on the X chromosome. We further examined autosomal SNVs in Element data and found that 98.0% of all SNV calls were also supported by the AVITI sequencing platform (**Supplementary Fig. 32**).”

Supplementary Figure 32: Evaluation of single-nucleotide DNMs using Element data.

The allele balance of each autosomal SNV call in Illumina or HiFi plotted against Element. By chi-squared test, there is no significant difference in allele balance between Illumina and Element for 99.2% and 91.7% of DNMs and PZMs, respectively. Between HiFi and Element, there is no significant difference for 96.0% and 92.5% of DNMs and PZMs, respectively.

L322: Why did the authors exclude G4 individuals from this analysis? I realize some analyses are only possible for the different generations, but most of the analyses portrayed in Figure 2 could be done with the entire set.

In our experimental design, we set G4 individuals aside for validation purposes in order to provide validation for *de novo* variants observed in G3 such that we can track the inheritance of a subset of these in G4 children, thus providing unambiguous evidence of them being true positives. Further, we do not have ONT data for G4 individuals as cell lines are not available. Variants included in the Figure 2 analyses were all validated using ONT data, and such validations are not possible for G4 samples, so we would not be able to report DNMs with the same level of confidence.

We revised the text as follows:

“We partition variants into SNVs or indels based on length and then validate each variant by requiring orthogonal support with ONT and/or Illumina (i.e., present in child and absent in parents). Because we did not have corresponding G4 cell lines for ONT data generation, we restricted our DNM discovery to G2 and G3 individuals. By this criterion, we discover 755 *de novo* SNVs and 73 *de novo* indels across the autosomes of 10 individuals (n=2 G2; n=8 G3 individuals, Fig. 2a), as well as 27 *de novo* SNVs and 1 indel on the X chromosome.”

L327: It is not clear to me how the authors derive that a sequence variant that is inconsistent with a haplotype is necessarily a PZ mutation that occurred early in development. There are other plausible explanations for that, somatic mutations that occurred postnatally and rise in frequency with advancing age (e.g. clonal hematopoiesis) and/or genotype artifacts. Postnatal somatic mutations and artifacts would be expected to be in equal frequency on the paternal/maternal chromosomes of the proband.

It is true that we cannot unambiguously attribute every variant with an inconsistent haplotype to early development, especially without multiple tissue samples from every individual studied. We termed these mutations PZMs to be consistent with earlier papers (Acuna-Hidalgo et al. 2015; Rockweiler et al. 2023), but operationally a PZM is no different from a postnatal somatic mutation. Its higher allele balance suggests that it originated earlier in development when compared to somatic mutations with lower VAF.

We find that 65% of PZMs are transmitted to the next generation, indicating that the variant was present in at least two tissues: blood and germline. These transmitted PZMs can be assumed to arise in the first weeks following fertilization, before primordial germ cells differentiate. PZMs that did not transmit to the next generation could in fact be somatic mutations that occurred in blood, but we are unable to distinguish those mutations with our current data.

We took great care to eliminate genotype artifacts from our postzygotic callset. All reported PZMs are observed in HiFi and at least one of ONT or Illumina, with 55% of our PZM calls supported across all three sequencing platforms.

L329: How does the allelic balance of the transmitted PZ allele look like in the offspring? The allele balance should be around 50%? Do the offspring carrying the transmitted PZ allele share a haplotype? The authors could try to construct the phylogenetic trees of the early cell lineages in the G2 parents (similar to Figure 3 from PMID: 31492841 and Figure 2 in PMID: 30397338). 64.5% transmission to at least one child suggests 7% ($7\% \approx 1 - \exp(\log(0.645)/6)$) transmission rate which is much lower than the roughly 25% allelic balance in Figure 2b. This suggests that these post-zygotic mutations calls are generally somatic mutations or false positives.

In total, we see 40 PZMs transmitted 120 times. For each of 120 transmissions, we examined a variant's allelic balance (AB) across HiFi, Illumina, and ONT (if available) reads in the child. We observe 29 transmissions (across 13 PZMs) where the AB was consistent across all data types and significantly different from 0.5 (**Supplementary Fig. 45**). This smaller fraction of PZMs may represent recurrent genotyping errors. Of those 13 PZMs, however, 7 have other transmissions with AB of 0.5, leaving only 6 PZMs that deviate from expectation.

In addition, we assigned haplotypes to each transmitted PZM by examining informative SNPs in HiFi data, similar to how we phased our mutations. For each variant, we expected to see it on every read attributed to the parent who transmitted the mutation, and 87.5% ($n=105/120$) of transmissions conformed to this expectation. We observed 15 transmissions over 5 PZMs that were not present on every read from the parent. When we compare these results to our AB results, we find that these 5 PZMs also have an AB significantly less than 0.5 across every transmission. These 5 PZMs represent likely false positives in our dataset.

For the 35% of PZMs that did not transmit to the next generation, the average AB in HiFi reads was 0.18. For a variant at that AB in our G3 samples with six children, we can estimate the probability that it is not transmitted to the next generation as $(1-0.18)^6 = 30\%$ (and 20% for G2 samples with eight children). It is almost a guarantee that not every PZM is shared between blood and germline cells, so even if we could confirm that every mutation arose in the early rounds of development, a transmission rate of 65% does not greatly differ from what we would expect to observe.

We report the text and figure below as **Supplementary Note 4**.

Supplementary Figure 45: Allele balance of transmitted PZMs.

For each child that inherited a PZM from a parent, we calculated the mean allele balance (AB) of the variant across HiFi, Illumina, and ONT data if present (for the children of G2 individuals but not G3). We used a binomial test to check whether the AB was significantly different from 0.5. If all transmissions of a variant had AB less than 0.5 and were not present on every read from a parental haplotype, they were determined to be false positive events.

Revised text:

“Of the 311 *de novo* SNVs in G2 and G3 individuals with offspring, 97.1% of germline events transmit to the next generation, compared with 64.5% of postzygotic events (**Extended Data Fig. 3, Supplementary Note 4**).”

L357: Please state the p-value and reword the statement that they differ, and references to previous studies of mutational spectrum of PZ mutations would be appropriate here.

We revised the text to include the p-value and emphasize that there are no significant differences between the DNM and PZM spectra:

“In addition, we observe a significant parental age effect of 1.55 germline DNMs per additional year of paternal age when fitting with linear regression ($p=0.013$)—a signal absent from *de novo* SNVs designated as PZMs (**Fig. 2c**). While our small sample size does not provide sufficient power to detect significant differences between the *de novo* and postzygotic mutational spectra (**Supplementary Fig. 33a**), we do observe a novel depletion of CpG>TpG PZMs that does not yet reach statistical significance (chi-squared test, $p=0.0809$), and an enrichment of postzygotic T>A substitutions (chi-squared test, $p=0.147$) that has been previously observed (Sasani et al. 2019). Using this approach, we successfully...”

L362-L371: Given that the PZ mutations are generally not in the germline based on the transmission patterns, how are the authors correcting for this fact? Do the authors quantify difference in false positive/negative rate across the different regions? The fact that the PZ mutations generally do not transmit, and the authors see a greater fraction of PZ mutations in the SDs, this indicates that there is a higher false positive rate in the SDs. Furthermore, given that the SD and centromeres are composed of variable number of repeats this introduce a variability in the denominator, could the authors expand on this?

We respond to the three points raised above as follows:

1. PZMs don't transmit: We, in fact, observe that the majority of PZMs (65% at minimum) do transmit and therefore are present in the germline. The remaining 35% do not violate transmission expectations given their low AB (see above).

2. False positive rate: To identify false positive postzygotic calls, we evaluated 40 PZMs transmitted to the next generation. We determined a PZM was false if it met two criteria: the AB in children who inherited the allele was significantly different from 0.5 across all sequencing platforms, and the alternate allele in these children was not present on every sequenced haplotype from the parent who transmitted it. We found 5 PZMs that appeared to be false

positives, for an overall PZM false positive rate of 12.5%. In segmental duplications, the false positive rate was 25% (n=2/8).

3. Denominator: To calculate the denominator, we examine the HiFi data aligned to every base in the reference genome. Of course, the reference genome does not perfectly describe the genomes of our samples, and in some cases, our samples have sequence that is not represented in the reference genome. By using the reference genome to calculate a denominator for our mutation rate, we exclude these regions. However, seeing as we are not able to assess them with our alignment-based *de novo* calling pipeline, it is appropriate to leave these sequences out of our denominator, at least for read-based calls. In the case of our assembly-based mutation calls, the callable denominator is based on the total number of correctly assembled base pairs and should include these regions excluded from our alignment-based method.

L388: How about the transmission rate to a single offspring?

The sentence in paper this is referring to:

“We were able to genotype 7.68 million of these loci in every member of the pedigree and, of those, 7.17 million (93.4%) loci were completely Mendelian concordant across all trios.”

We interpret this as asking for a given trio, how many loci are concordant. We calculated transmission for each trio individually. We find that, on average, 98.8% of loci were Mendelian concordant in a given trio (range: 98.4-99.3%).

L404: Did the authors try to use the HiFi data to validate these STRs/VNTRs DNMs via adjacent het SNPs?

Both TRGT genotyping and our strategy for determining DNM parent-of-origin (POI) did leverage nearby heterozygous SNPs. TRGT uses small variants within TR-containing reads to assign those reads to either haplotype before inferring a diploid genotype. Thus, our TR DNM discovery strategy implicitly leverages nearby heterozygous SNPs.

Our POI strategy involved looking for heterozygous SNPs +/-500 kbp from the DNM in the child at which the alternate SNP allele could have been inherited from either the mother or father, but not both (e.g., the child was genotyped $\emptyset/1$, the mother $1/1$, and the father \emptyset/\emptyset). By examining long stretches of these “informative SNPs,” we could reliably determine the POI for most of the DNMs observed in G3.

To more formally address the reviewer’s question, we used Element sequencing data to compare orthogonal validation rates for STR DNMs with or without a confident POI. We find that phased and unphased STR DNMs are equally likely to orthogonally validate with Element (Fisher’s exact $p = 0.92$).

	STR DNM is supported by Element	STR DNM is not supported by element
STR DNM is confidently phased	46	15
STR DNM is not confidently phased	8	1

L423: Once again 76.2% transmission to at least one child of 6 is a severe under transmission from the expected 50% transmission to a single child. How does this look for homopolymers?

The reviewer is correct that the transmission rate for TR DNMs is below expectation (naively, we would expect a DNM to be transmitted to at least one of six children with probability $1 - 0.5^6$). We believe that several factors could be impacting our lower-than-expected transmission rate. For example, given an observation of a TR DNM in an individual, we assess transmission to a subsequent generation by identifying the children of that individual that possess a TR allele matching the allele length of the *de novo* allele observed in the individual. Since there is often sequencing “stutter” that impacts our ability to precisely define the length of a *de novo* allele, some true transmission events likely go undetected because we are unable to match the *de novo* allele to its inherited copies in the children. However, we don’t find that the transmission rate is substantially lower at homopolymers than at non-homopolymers (Fisher exact $p = 0.76$) but note that we can only assess transmission in two G3 samples.

	STR DNM is transmitted to G4	STR DNM is not transmitted to G4
DNM is homopolymer	87	26
STR DNM is not homopolymer	9	2

We made a note of this in the revised text as follows:

“Of the 128 *de novo* TR alleles observed in the two G3 individuals, 96 (75%) were transmitted to the next generation, which is significantly lower than the expectation reflecting the challenges that still remain in accurately characterizing *de novo* TRs.”

Figure3d: It is hard to see the lines in this plot and this visualization could be improved. Are the TR allele sizes consistent with flanking phase informative SNVs? For example, reads in NA12877 supporting the 248 allele are they linked with SNVs on the H1 haplotype of the NA12889?

Figure 3d has been revamped to improve line visibility and clarity; a color-blind friendly palette is now used to denote the generations.

The TR allele sizes are indeed fully consistent with flanking phase-informative SNVs. While the inheritance vectors already inform on this, it is useful to further illustrate this in the context of the SNVs in the proximity of this particular TR site. We added a supplemental note and figure to highlight this:

“*De novo* TR alleles are present at this locus in seven of eight G3 individuals; these *de novo* alleles transmit to four G4 individuals, with two expanding further upon transmission. Additionally, the spouse of a G3 individual (sample 200080) carries a distinct TR allele that undergoes a *de novo* contraction in subsequent transmissions. All allele transmissions are fully consistent with the inheritance vectors (**Supplementary Note 5**) and are supported by both HiFi and ONT reads.”

Supplementary Note 5:

“Haplotype analysis of flanking SNVs to validate recurrent tandem repeat allele transmission presented in Figure 3c,d:

To demonstrate that the flanking informative SNVs are consistent with the genotyped TR allele sizes at chr8:2376919-2377075 (T2T-CHM13) in the context of haplotypes across multiple samples, we extracted SNVs from a 4.5 kbp region upstream and downstream of this tandem repeat in all reads. Only SNVs with an allele frequency above 0.2 were kept, and their positions were normalized across all samples, categorizing each SNV in each read as observed, unobserved, or not spanned. Per haplotype we then condensed the SNVs in the read data into a single representative vector (**Supplementary Fig. 46**).

Condensing the haplotypes involves computing posterior probabilities for each SNV on a given haplotype using a simple Bayesian model, where the presence of allele 1 is modeled as a binomial distribution with a Beta prior ($\alpha = 0.5$, $\beta = 0.5$). The posterior mean is calculated based on the observed counts of allele 1 and allele 0 across all reads, while accounting for uncovered positions where reads do not span the SNV of interest. We then generate a binary representative vector for each haplotype by thresholding the posterior mean at 0.5, assigning a value of 1 if the posterior mean suggests the presence of allele 1 with a probability of 50% or higher, and 0 otherwise.”

Supplementary Figure 46: Transmission of flanking SNVs at recurrent TR locus.

a) Sites with informative SNVs within reads corresponding to each haplotype can be condensed into a representative SNV vector for that haplotype. b) The sample-haplotype stratified SNV matrix at chr8:2376919-2377075 (T2T-CHM13) across generations 1 to 3.

L479: Most of the centromeres are not completely sequenced and assembled, it would be useful to the reader to state the fraction of centromeres that were assembled.

We state the number of completely assembled centromeres (i.e., 44.7%) in the main text and Supplementary Figure 4:

“Notably, we successfully sequenced and assembled 288 centromeres (44.7%) across the three generations, which required application of both Verkko and hifiasm (UL), as each assembler preferentially assembled different human centromeres (**Extended Data Fig. 1e**, **Supplementary Fig. 4, Methods**).”

Note from the supplement:

“Verkko, for example, assembled 175/644 centromeres (27.2%) accurately, while hifiasm (UL) assembled 161/644 centromeres (25.0%) accurately. Only 48/644 centromeres (7.5%) were completely and accurately assembled by both Verkko and hifiasm (UL). Thus, by merging complete centromeres generated by both assemblers, we create a nonredundant list of 288 completely and accurately assembled centromeres.”

L482: What is the transmission rate here? This is especially important here as the event indicated in Figure 4d could be a somatic cell-line duplication in G1 NA12891 which would appear as a germline deletion in G2 NA12788. It would be beneficial to have the genotypes for all the individuals as a supplementary table or figure for these 18 events.

We assessed the transmission rate, defined as the rate with which a *de novo* SV is transmitted to the next generation, for all centromeres when possible. Of the 18 *de novo* SVs in centromeres, only 8 could be assessed for transmission, as they had offspring that had been sequenced. Of these 8, we found that 100% of them were transmitted to the next generation. The remaining 10 *de novo* SVs could not be assessed for transmission because their offspring had not been sequenced in this study. We updated **Supplemental Table 10**, which provides the contig ID, position of the SV event, validation status, and genotypes for all 18 *de novo* SV events.

We revised the main text to now reflect this additional information:

“Comparing these assembled centromeres between parent and child (Methods), we identify 18 (12%) *de novo* SVs validated by both ONT and HiFi data with roughly an equivalent number of insertions and deletions (Fig. 4b). All *de novo* SVs (n=8) that had a child sequenced as part of this study confirmed transmission of the SV to the next generation (**Supplemental Table 10**).”

L485-487: This sentence is cryptic, it would also be nice to have a schematic view of the terminology used here for the general reader (HOR arrays, HOR cassettes and pericentromeric flanking sequence).

Thank you for this suggestion. We now include a schematic of the centromere in Extended Data Figure 4a and indicate the different sequence elements that comprise it, including α -satellite HORs, monomeric/divergent α -satellite, α -satellite HOR arrays (active and inactive), and pericentromeric sequences.

Extended Data Figure 4. Changes in centromere sequence, structure, and DNA methylation patterns across generations. a) Schematic of the generalized organization of human centromeres and their flanking sequence. Major components and their structures are shown. HOR, higher-order repeat. Not drawn to scale.

L509-L512: What is the false positive/negative rate of the SNVs detected within these alpha-satellite HORs? As before the denominator is probably tricky to derive within these repeats.

As suggested, we revisited the *de novo* SNVs called within the centromeres, requiring validation by both ONT and PacBio HiFi data. We detected 16 SNVs that were confirmed as *de novo* with both technologies, including 5 in the α -satellite HORs and 11 mapping to the flanking sequences. Thus, we estimate a high false positive rate of ~82.2% for any of the 90 candidate *de novo* SNVs. The majority of false positives were due to the presence of a small portion of the reads (typically 5-10%) containing the SNV, while the remaining reads (90-95%) did not contain the SNV. This suggests that a subpopulation of cells containing the SNV may have been the source of the SNV in the child, or it could suggest that this region is particularly prone to mutagenesis, and the same mutation arose during cell culture (as detected with ONT data), mimicking what was observed in the child. Additional experiments will be required to investigate these low allele-balance variants. However, to be consistent with all other *de novo* SNV validations, including autosomal and the Y chromosome, we are requiring multi-platform support and report only the 16 *de novo* SNVs. When we combine this count with our read-based discovery of *de novo* SNVs (where we also required multiplatform support), we assumed no overlap between the callable base pairs between the read-based and assembly discovery methods and summed both to compute an overall centromere rate of 5.70×10^{-8} (C.I. $2.15 \times 10^{-8} - 9.26 \times 10^{-8}$). We regard this as a conservative estimate. We revised the corresponding text as follows and revised Figure 2d:

“We identify 16 SNV DNMs in centromeres, including five within the α -satellite HOR arrays, revealing a DNM rate of 1.01×10^{-7} mutations/bp/generation (95% C.I. = $5.89 \times 10^{-9} - 1.90 \times 10^{-7}$). This rate is comparable to the rate from our read-based mapping approach, which identified 18 centromeric SNVs, albeit over more than 10 times the amount of sequence, resulting in a DNM rate of 4.21×10^{-8} mutations/bp/generation (95% C.I. = $1.98 \times 10^{-8} - 6.44 \times 10^{-8}$) (**Fig. 2d, Supplementary Table 10**). Combined, we estimate a significantly higher SNV DNM rate for centromeres of 5.70×10^{-8} (two-sided t-test, $p=0.02$). While discovery of these DNMs still remains challenging, we believe this a conservative estimate because we required validation of all events by ONT and HiFi sequencing platforms.”

Revised Fig. 2 panel d.

L548-L555: How many DNMs here are transmitted to the next male? They seem to generally being transmitted to G2 based on Figure 5 but it would be nice to get a quantification of that. There is no comparison to previous estimates of the Y de novo mutation rate, it would be advisable to restrict to the regions of the Y that are accessible to short reads to see if the estimate of the mutation rate is comparable to previous ones.

A similar question about the transmission of Y-chromosomal DNMs was raised by Reviewer 1's minor comment 12 (please see above for full details). To summarize, all applicable (i.e., if male offspring was available) Y-chromosomal DNMs were 100% concordant with the expected transmission through generations. Specifically, all *de novo* SNVs (n=10), indels (n=3), and SVs (n=3) identified in G2-NA12877 were present in the assemblies of his four sons (G3: NA12882, NA12883, NA12884 and NA12886) and supported by the PacBio HiFi read data of his three grandsons (G4: 200101, 200102 and 200105). Similarly, *de novo* SNVs (n=10), indels (n=3), and SVs (n=1) identified in G3-NA12886 were supported by the PacBio HiFi read data of his three sons (G4: 200101, 200102 and 200105).

Our *de novo* SNV rate for the ~22 Mbp euchromatic regions accessible with Illumina technology were comparable to previous estimates (1.81 × 10⁻⁸ [95% CI: 0 - 4.89 × 10⁻⁸] mutations per bp per generation in this study vs. 2.87 × 10⁻⁸ [95% CI: 2.68 - 3.08 × 10⁻⁸] mutations per bp per generation from Helgason et al., 2015).

To clarify these points, we added the following sentences to the Results "Y chromosome mutations" section:

"The latter range from 2,416 to 4,839 bp in size, each affecting an entire *DYZ2* repeat unit(s), with an average of one SV per Y transmission. All applicable DNMs (SNVs: n=20/48, indels: n=6/9, SVs: n=4/5) are concordant with the expected transmission through generations (i.e.,

from G2 to G3-G4 and from G3-NA12866 to his three male descendants in G4) (**Fig. 5b**). Overall, 83% (52/63) of the DNMs ...”

We revised the text as follows:

“We thus estimate the *de novo* SNV rate of 1.99×10^{-7} (95% CI = $1.59 - 2.39 \times 10^{-7}$) for the entire MSY combining both read- and assembly-based approaches. While this estimate is an order of magnitude higher than previously reported for Y euchromatic regions, if we restrict to the ~22 Mbp euchromatic regions accessible with Illumina technology, the rates are comparable to previous estimates (1.81×10^{-8} [95% CI: 0 - 4.89×10^{-8}] mutations per bp per generation in this study vs. 2.87×10^{-8} [95% CI: $2.68 - 3.08 \times 10^{-8}$] mutations per bp per generation from (Helgason et al. 2015)) (**Supplementary Table 13**).”

L586: Missing quantification for this statement, and this statement should be reworded according to the level of evidence from the quantitative analysis.

We agree with the reviewer that this statement was not fully supported by the visual presentation of the data. To alleviate this, we now add a new figure panel d in Supplementary Figure 37 to show the correlation between the number of *de novo* SVs ($n=37$, for four SVs we were not able to determine homolog of origin) occurring on maternally and paternally inherited homologs with respect to corresponding parental age. Pearson’s correlation does not meet the level of significance; however, the age-related trend is obvious from the data.

Revised text:

“Overall, 68% (28/41) of events originate in the paternal germline with a trend toward an increase in SVs with paternal age, although it does not reach statistical significance (Supplementary Fig. 37).”

Supplementary Figure 37: Summary of detected *de novo* SVs (n=41).

a) A barplot showing the total number of *de novo* SVs per G3 sample as stacked insertion (INS) and deletion (DEL) counts. **b**) A barplot showing the total number of bases affected by *de novo* SVs per G3 sample as stacked insertion (INS) and deletion (DEL) base-pair counts. **c**) A barplot showing the percentage of *de novo* SVs inherited from paternal (blue) or maternal (red) homologs. The gray bar shows *de novo* SVs where inheritance could not be reliably determined. **d**) Correlation between the number of *de novo* SVs (y-axis) and parental age (x-axis) shown separately for maternally (red) and

paternally (blue) inherited homologs. e) Distance distribution of *de novo* TRs from detected cross-over in a given sample and chromosome. Empty points mark *de novo* TRs where no crossover was detected for a given chromosome in a given sample.

L595: Most of the double strand breaks (DSB) formed during meiosis will not be resolved as crossovers, the lack of correlation between crossover location and *de novo* SVs in this limited number of meiosis is inconclusive. The interpretation should reflect that.

Discovery of *de novo* SVs genome-wide with this level of haplotype resolution including recombination breakpoints is relatively unprecedented. While the number of such events is low, NAHR is often invoked as the primary mechanism for their formation. The fact that NONE of the 27 events have recombination breakpoints mapping at the site of a *de novo* VNTR we believe is an important observation. Nevertheless, we acknowledge the possibility of other mechanisms associated with double-stranded breaks but not recombination as well as more families need to be assessed.

We revised the text as follows:

“We find, however, that none of the 27 euchromatic *de novo* SVs coincide with detected crossover positions (Supplementary Fig. 37d). In fact, the average distance was often many megabase pairs apart, arguing against NAHR between homologs as the primary mechanism for their origin, although we cannot preclude other mechanisms associated with double-strand breaks not involving recombination. A larger number of *de novo* SVs events in many more families will need to be assessed.”

L603-606: Is this consistent with the PZ SNV mutations transmitted to the offspring? The early mutations transmitted to the offspring determine the early cell lineages of parent.

This is indeed potentially analogous to the PZ SNV mutations that are transmitted to the next generation. We revised the text to reflect this possibility:

“We also find this insertion present at a low frequency (~11% of overlapping reads) in the parent (G2-NA12878) but not in the grandparental transmitting haplotype. This is consistent with a germline mosaic event potentially arising in G2 postzygotically, similar to transmitted SNV PZMs (Fig. 5d, Supplementary Fig. 39).”

L631-634: All the claims should be revisited after quantification of the false positive/negative rates.

For PZMs, we used allele balance and haplotypes to determine that five out of our evaluated 40 PZMs were in fact false positive calls, with a false positive rate (FPR) of 12.5%. Compared to the previous study of this family (Sasani et al. 2019), there are no previously identified PZMs that we did not recover, yielding a false negative rate (FNR) of 0%. As such, we can revise our estimate of the PZM rate (μ) to be $\mu \cdot (1 - \text{FPR}) / (1 - \text{FNR})$, or $2.23 \times 10^{-9} \cdot (1 - 0.125) / (1 - 0.0) = 1.95 \times 10^{-9}$.

For germline SNVs, we used Element data for a final round of validation. We determined that a DNM was false positive if it was not supported in 1 of 4 sequencing technologies (HiFi, ONT, Illumina, Element) and it had AB < 0.1 in at least one other sequencing technology. For example, a variant with AB of 0.39 in HiFi, 0.05 in ONT, 0.2 in Illumina, and 0 in Element was considered a false positive event. In total, we observed 8 such false positives out of 626 DNMs, for a FPR of 1.28%. Like for PZMs, we compared our callset to that of Sasani et al. (2019). After validating their variant calls with our pipeline, there were only four previously discovered DNMs that were absent from our callset. We can calculate our FNR as (false negative)/(true positive + false negative), giving an FNR of 0.64%. We can revise our estimate of the *de novo* mutation rate (μ) to be $\mu \cdot (1 - \text{FPR}) / (1 - \text{FNR})$, or $1.17 \times 10^{-8} \cdot (1 - 0.0128) / (1 - 0.0064) = 1.16 \times 10^{-8}$.

We can combine these germline and postzygotic rates for an overall *de novo* SNV rate of 1.355×10^{-8} mutations per base pair per generation.

In addition, we further validated our set of centromeric SNVs, eliminating two-thirds of our initial callset as false positives. As such, our centromeric mutation rate has been revised to 1.01×10^{-7} and is no longer significantly different from the overall autosomal mutation rate.

Revised text:

“We find that the rate of *de novo* SNVs varies by more than an order of magnitude depending on the genomic context. In particular, we observe elevated rates of *de novo* SNVs in repetitive regions both for germline and postzygotic events, consistent with recent human population-based analyses^{7,52} and theoretical predictions⁵³. SD regions show an 88% increase (2.2 [95% CI: $1.53 - 2.86 \times 10^{-8}$] vs. 1.17×10^{-7}), which is driven by SDs with >95% identity. We also observe a significant decrease in the *de novo* transition/transversion ratio in SDs compared to all DNM calls (chi-squared test, $p=0.0109$) consistent with previous expectations⁷ (**Supplementary Note 7**). Although the number of validated SNV DNMs is still rather modest, we currently estimate that satellite DNA in the Yq12 heterochromatic region is at least 30 times more mutable than autosomal euchromatin (3.68×10^{-7} mutations per bp per generation).”

Using this approach, we successfully assay 91.9% of the autosomal genome (2.66 Gbp) with an overall SNV mutation rate of 1.39×10^{-8} SNVs/bp/generation (95% CI: $1.22 - 1.56 \times 10^{-8}$) (**Supplementary Fig. 33b, Supplementary Note 4**).”

L646-648: Missing CIs for the estimate presented here, and it would be cleaner to have this comparison in the results.

We thank the reviewer for this comment and we added the CIs to the discussion section as suggested. Also, as suggested in the comment above for L548-L555, we added a comparison to previous Y-chromosomal mutation rate to the Results section (see details above):

“While our SNV DNM rate estimate for Y euchromatic regions is comparable to previous pedigree-based work (~22 Mbp, 1.81×10^{-8} [95% CI: 0 - 4.89×10^{-8}] mutations per bp per generation in this study compared to 2.87×10^{-8} [95% CI: 2.68 - 3.08×10^{-8}] mutations per bp per generation from⁴³), the SNV estimate for Yq12 is >20× higher.”

L650-655: I am confused as most of the PZ mutations identified by the authors did not transmit (100%-64.5%) despite the numerous offspring and here they are attributing them all to the germline. In a previous analysis by the authors [PMID: 31549960] and using monozygotic twins [PMID 33414551] a much lower fraction of pre-PCGS mutations is detected.

We see that the majority of PZMs (64.5%) do in fact transmit to the next generation. As we outline above, based on their low allele balance, we do not expect 20-30% of PZMs to transmit to the next generation, assuming that these variants are present in the germline as well as the blood. However, we do not claim that untransmitted PZMs are present in the germline, nor do we claim that they are pre-PCGS mutations.

In the previous analysis referenced (Sasani et al. 2019), approximately 10% of DNM calls are determined to be postzygotic in origin (referred to as gonosomal variants), compared to 17% of DNM calls in this analysis. While this is a marked increase, we are able to construct our haplotypes with long-read data, which allowed us to phase more than 90% of our variants and improved our ability to detect PZMs. The other analysis (Jonsson et al. 2021) found that fewer than 5% of DNMs are postzygotic in origin, using Strelka2 as their variant caller on Illumina data. Strelka2 is a very specific variant caller that identifies far fewer mutations than our HiFi variant callers (Noyes et al. 2022). Further, we see an enrichment of PZMs in repetitive regions of the genome, many of which cannot be accurately examined with Illumina data. We also do not restrict our PZMs to pre-PCGS mutations, which will also inflate our number of PZMs.

We revised the text slightly to reflect the additional power to classify PZMs:

“In addition to germline events, we classified nearly twice the number of *de novo* SNVs as PZMs (12.9 PZM per transmission or 17%) compared to even the highest previous estimate (6-10%)^{14,49}. Previous studies have distinguished between *de novo* postzygotic and germline SNVs using allele balance thresholds or by identifying incomplete linkage to nearby SNVs across three generations. Long-read sequencing provides a third approach, allowing us to assign nearly every *de novo* SNV to a parental haplotype and distinguish mosaic events by the presence of three distinct long-range haplotypes and as a result correctly classify PZMs with higher allele balances.”

L657: Is this reference appropriate? The reference is describing mitotic errors, rather than mutations. There are numerous studies looking at this developmental epoch through lens of colonies/laser dissection which the authors could refer to. Further, is the rate of mitotic divisions abnormally high in the cleavage state compared to somatic cell divisions?

In mammalian embryogenesis, there appears to be a period of accelerated cell divisions. Unlike in other species, where these accelerated divisions occur right after fertilization, in mammals the

acceleration happens during gastrulation (Mac Auley et al. 1993, Snow 1977, Farrel and O'Farrel 2014).

We revised the citation to include two papers that show elevated mutation rates in early embryogenesis:

“Early cell divisions of human embryos are frequently error prone (Added references) with an accelerated rate of cell division; these properties may contribute to the high fraction of PZMs with high (>25%) allele balance (38% are estimated to have high allele balance and 83% of these (n=20/24) are transmitted to the next generation).”

Added references:

Park, S., Mali, N.M., Kim, R. *et al.* Clonal dynamics in early human embryogenesis inferred from somatic mutation. *Nature* 597, 393–397 (2021).

Ju, Y., Martincorena, I., Gerstung, M. *et al.* Somatic mutations reveal asymmetric cellular dynamics in the early human embryo. *Nature* 543, 714–718 (2017).

REFEREE #3:

This paper describes telomere-to-telomere assembly of a four-generations pedigree, with the goal of studying de novo mutations and recombination rates. Although this pedigree has been analyzed before, the previous studies were limited by short-read sequencing; thus only small de novo mutations were profiled in previous studies. This work uniquely enables the analysis of highly repetitive regions of the genome, such as segmental duplications and centromeres, which more than doubled the number of discovered de novo mutations. High quality phased assemblies also enabled generation of high-resolution recombination maps.

Overall, this paper highlighted important new biological insights on de novo mutations and recombination, and generated a highly valuable resource for many future studies, as the authors are making sequencing data, assemblies and variant calls publicly available. This work employs state of the art assembly, variant calling and QC techniques, and convincingly demonstrates the high quality of the produced resources. I only have a few small requests below.

We appreciate the overall positive response to the work.

Comments:

1. It seems that manual curation was required for generating the final call set for de novo SVs and centromeric variants. It is understandable that the fully automated generation of this benchmark may not be practical. I am curious how many variants were selected for manual screening and what was the rate of rejected variants? Did the analysis reveal any patterns of systematic errors?

Indeed, manual curation was critical to identifying true *de novo* germline SVs. We considered almost a thousand candidate events with only 50 (41 in G3) events ultimately passing our stringent criteria. While it is difficult to generalize, both the type of event (VNTRs vs. centromeres), the caller, the reference, and whether *de novo* variants were defined by read-based versus assembly-based approaches seemed to have an effect on the false positive and negative rates. Because a similar request was made by Referee #2, we combined these details regarding SV validation and curation by both type and class of *de novo* SNV and SV into a **Supplementary Note 10** (see below):

“Evaluation of SV DNMs

Centromeric SVs: We initially identified 29 candidate SVs in the centromeres (7 between G1 and G2, and 22 between G2 and G3), based on whole-contig alignments between parent and child—note not using a standard human genome reference. We manually checked each of these by aligning the raw ONT and PacBio HiFi reads to the relevant assemblies and assessing for the presence of the SV. We found that 11/29 of these candidate SVs were not supported by the underlying raw reads—false positive rate of >37% for assembly-based calls. In almost all 11 cases, we found that a small portion of the raw reads also contained the SV, indicating that there was a subpopulation of cells with the SV and a subpopulation without the SV—thus a potentially somatic event. During the genome assembly process, the structure that was most supported by the reads was included in the final assembly, which meant that, in some cases, a

false *de novo* SV was included. We were careful in our analysis to check every SV we detected for long-read sequencing support, and the final set of 18 *de novo* centromeric SVs passed this quality control. A subset of these (n=8) where transmission could be assessed, were all confirmed to be transmitted to the next generation. We now include a list of all true *de novo* centromeric SVs and their validation metrics in the **Supplementary Table 10**.

VNTR SVs: In the case of *de novo* SVs, we followed a similar procedure. We first collected candidate *de novo* SVs reported based on phased genome assemblies and read-based variant calling. In the case of read-based SV calls from Sawfish, we searched for support in phased genome assemblies by constructing a multiple sequence alignment (MSA) of each deemed *de novo* SV. Out of a total of 658 *de novo* SV candidates, we marked 12 variants as likely valid (false positive rate >98%). All of these were marked as various VNTRs and were subsequently vetted by TRGT callset. We evaluated candidate SV DNMs at TR loci by visually inspecting the distribution of CIGAR operations in HiFi reads aligned to each locus (for more information about the process we used to extract CIGAR information from high-quality reads, see Methods section entitled “Measuring concordance with orthogonal sequencing technology”). For each member of a trio, we plotted the net number of CIGAR operations in every read aligned to a VNTR locus harboring a candidate DNM. We included examples of the resulting images in the Platinum-Pedigree-Consortium GitHub repository associated with the manuscript (https://github.com/Platinum-Pedigree-Consortium/ppc-trs/tree/main/hq_sv_images). By examining the distributions of CIGAR operations in each member of the trio, we evaluated whether the child’s sequencing reads contained evidence for a *de novo* allele length absent from both parents.

Chromosome Y SVs: The SVs across the MSY were called from G2 and G3 Y assemblies using Dipcall with the G1-NA12889 Y assembly as a reference (**Methods**). Across the one G2 and four G3 males, Dipcall called a total of 70 candidate SVs. Similar to the SNVs and indels, a higher number of raw SV calls were made from G3-NA12886 Y assembly (n=30), while 16 candidate SVs were called from G2-NA12877 and 7 to 10 SVs from the remaining three samples (G3-NA12882, G3-NA12883 and G3-NA12884). Any *de novo* variants overlapping with flagged regions by Flagger and/or NucFreq were filtered out. For all SV calls, HiFi read depth for reference and alternative alleles was visualized and SVs in regions showing high levels of read depth variation coinciding with clusters of SNVs with >10% of reads supporting an alternative allele removed. Additionally, HiFi and ONT reads mapped to both the reference and individual assemblies were checked for support. For all applicable SVs (i.e., if male offspring was available), 100% concordance with the expected transmission through generations was confirmed. Specifically, all *de novo* SVs (n=3) identified in G2-NA12877 were present in the assemblies of his four sons (G3: NA12882, NA12883, NA12884 and NA12886) and supported by the PacBio HiFi read data of his three grandsons (G4: 200101, 200102 and 200105). Similarly, the *de novo* SV identified in G3-NA12886 was supported by the PacBio HiFi read data of his three sons (G4: 200101, 200102 and 200105).

Other SVs: All predicted *de novo* SVs outside of centromeres and VNTRs were evaluated by Verkko as well as hifiasm (UL) assemblies. We did this by extracting a sequence around the SV by adding two times the size of the SV on each side. We extracted the sequence from a G3 individual and corresponding G2 parents. Next, we constructed the MSA and visually checked if the predicted SV is visible in both Verkko and hifiasm (UL) assemblies. For assembly-based *de novo* SV candidates, we considered the PGGB (pangenome SV callset) and PAV callsets with a reported initial number of 84 and 75 *de novo* candidates, respectively. Of these, all but one assembly-based *de novo* candidate were rejected. Thus, >99% were false positives. The only *de novo* variant reported by phased assemblies is the SVA insertion described in this paper. In the case of read-based *de novo* SV candidates, we considered the Sawfish callset of the initial number of 658 of putative *de novo* SVs. See VNTR section above for more details. With respect to GRCh38, we also considered other read-based *de novo* SV candidates reported by Sniffles (n=77) and pbsv (n=45). However, none of these passed our assembly-based validation. We note that our strict requirement of having all *de novo* SVs fully assembled might lead to false negative calls and, as such, our reported numbers likely represent a lower bound. The complete list of detected *de novo* SVs is reported in **Supplementary Table 10**.

Indeed, we observed systematic errors in both hifiasm and Verkko assemblies. Most of these were false insertions, which in the case of hifiasm were often composed of a long homopolymer stretch (**Supplementary Fig. 52**). We, however, note that hifiasm and Verkko are still in active development and some of these errors are being addressed in newer software versions.

Supplementary Figure 52: Examples of false *de novo* insertions in phased genome assemblies.

a) Multiple sequence alignments for Verkko (left) and hifiasm (right) assemblies. The top two rows represent both haplotypes for the child (G3-NA12879) while the other four rows represent G2 parental haplotypes (NA12877-father,

NA12878-mother). False insertion is visible as a piece of DNA present only in the child. **b)** Long-read (PacBio HiFi) alignments to the child's (G3-NA12879) assemblies. Coverage of the most frequent base at each position is shown as a black line. Coverage of the second most frequent base is shown as the red line. False insertion in haplotype 2 of G3-NA12879 is visible as low read support visible as clear dip in the coverage of HiFi reads of the false insertion. **c)** Multiple sequence alignments for Verkko (left) and hifiasm (right) assemblies. The top two rows represent child's (G3-NA12879) haplotypes while the other four rows represent G2 parental haplotypes. False insertion is visible as a piece of DNA present only in haplotype 1 of the child and being composed of only G's.

2. The authors used multiple tools and techniques to generate initial de novo variant candidates. For example, to generate an initial set of SVs, multiple reference-based tools, as well as linear assemblies and pangenome graphs analysis were used. It would be interesting to see if the different variant calling methods were in general in good agreement. Was there any evidence that some real variants may be uniquely captured by just one or a few methods only?

On average, each SV call is supported by 2.5 (average) of the four methods, and 61% of SV calls are supported by more than one method. The agreement between methods varied by type: deletions had the highest overlap supported by 2.9 (average) callers, insertions supported by 2.3 (average) callers, and inversions supported by 1.1 (average) callers. These details are further elaborated in Kronenberg et al., 2024 (Kronenberg et al. 2024). We also added a new supplementary figure so that readers can visualize the agreement between SV calling methods and reference it in the text:

“We establish a variant truth set and identify a total of 5.95 million SNVs/indels and 35,662 SVs—all of which are pedigree consistent across the second and third generations (**Supplementary Table 5, Supplementary Fig. 13, Methods**) (Kronenberg et al. 2024).”

Supplementary Figure 13. UpSet diagram illustrating the agreement among different SV callers.

Each overlap represents the intersection of structural variant (SV) callsets. The bars on the right indicate the size of each SV callset, while the central bar chart displays the counts of overlapping sets. On average, each SV call is supported by 2.5 (average) of the four methods, and 61% of SV calls are supported by more than one method. The agreement between methods varied by type: deletions had the highest overlap supported by 2.9 (average) callers, insertions supported by 2.3 (average) callers, and inversions supported by 1.1 (average) callers.

REFERENCES:

- Acuna-Hidalgo, Rocio, Tan Bo, Michael P. Kwint, Maartje van de Vorst, Michele Pinelli, Joris A. Veltman, Alexander Hoischen, Lisenka E. L. M. Vissers, and Christian Gilissen. 2015. "Post-Zygotic Point Mutations Are an Underrecognized Source of De Novo Genomic Variation." *American Journal of Human Genetics* 97 (1): 67–74.
- Ahn, B. Y., and D. M. Livingston. 1986. "Mitotic Gene Conversion Lengths, Coconversion Patterns, and the Incidence of Reciprocal Recombination in a *Saccharomyces Cerevisiae* Plasmid System." *Molecular and Cellular Biology* 6 (11): 3685–93.
- Altemose, Nicolas, Glennis A. Logsdon, Andrey V. Bzikadze, Pragya Sidhwani, Sasha A. Langley, Gina V. Caldas, Savannah J. Hoyt, et al. 2022. "Complete Genomic and Epigenetic Maps of Human Centromeres." *Science* 376 (6588): eabl4178.
- Bhérer, Claude, Christopher L. Campbell, and Adam Auton. 2017. "Refined Genetic Maps Reveal Sexual Dimorphism in Human Meiotic Recombination at Multiple Scales." *Nature Communications* 8 (1): 14994.
- Broman, K. W., J. C. Murray, V. C. Sheffield, R. L. White, and J. L. Weber. 1998. "Comprehensive Human Genetic Maps: Individual and Sex-Specific Variation in Recombination." *American Journal of Human Genetics* 63 (3): 861–69.
- Conrad, Donald F., Jonathan E. M. Keebler, Mark A. DePristo, Sarah J. Lindsay, Yujun Zhang, Ferran Casals, Youssef Idaghdour, et al. 2011. "Variation in Genome-Wide Mutation Rates within and between Human Families." *Nature Genetics* 43 (7): 712–14.
- Eberle, Michael A., Epameinondas Fritzilas, Peter Krusche, Morten Källberg, Benjamin L. Moore, Mitchell A. Bekritsky, Zamin Iqbal, et al. 2017. "A Reference Data Set of 5.4 Million Phased Human Variants Validated by Genetic Inheritance from Sequencing a Three-Generation 17-Member Pedigree." *Genome Research* 27 (1): 157–64.
- Eslami Rasekh, Marzieh, Yözen Hernández, Samantha D. Drinan, Juan I. Fuxman Bass, and Gary Benson. 2021. "Genome-Wide Characterization of Human Minisatellite VNTRs: Population-Specific Alleles and Gene Expression Differences." *Nucleic Acids Research* 49 (8): 4308–24.
- Gershman, Ariel, Michael E. G. Sauria, Xavi Guitart, Mitchell R. Vollger, Paul W. Hook, Savannah J. Hoyt, Miten Jain, et al. 2022. "Epigenetic Patterns in a Complete Human Genome." *Science* 376 (6588): eabj5089.
- Halldorsson, Bjarni V., Marteinn T. Hardarson, Birte Kehr, Unnur Styrkarsdottir, Arnaldur Gylfason, Gudmar Thorleifsson, Florian Zink, et al. 2016. "The Rate of Meiotic Gene Conversion Varies by Sex and Age." *Nature Genetics* 48 (11): 1377–84.
- Helgason, Agnar, Axel W. Einarsson, Valdís B. Guðmundsdóttir, Ásgeir Sigurðsson, Ellen D. Gunnarsdóttir, Anuradha Jagadeesan, S. Sunna Ebenesersdóttir, Augustine Kong, and Kári Stefánsson. 2015. "The Y-Chromosome Point Mutation Rate in Humans." *Nature Genetics* 47 (5): 453–57.
- Jeffreys, A. J., N. J. Royle, V. Wilson, and Z. Wong. 1988. "Spontaneous Mutation Rates to New Length Alleles at Tandem-Repetitive Hypervariable Loci in Human DNA." *Nature* 332 (6161): 278–81.
- Jonsson, Hakon, Erna Magnusdottir, Hannes P. Eggertsson, Olafur A. Stefansson, Gudny A. Arnadottir, Ogmundur Eiriksson, Florian Zink, et al. 2021. "Differences between Germline Genomes of Monozygotic Twins." *Nature Genetics* 53 (1): 27–34.
- Kong, Augustine, Daniel F. Gudbjartsson, Jesus Sainz, Gudrun M. Jonsdottir, Sigurjon A. Gudjonsson, Bjorgvin Richardsson, Sigrun Sigurdardottir, et al. 2002. "A High-Resolution Recombination Map of the Human Genome." *Nature Genetics* 31 (3): 241–47.
- Kristmundsdottir, Snaedis, Hakon Jonsson, Marteinn T. Hardarson, Gunnar Palsson, Doruk Beyter, Hannes P. Eggertsson, Arnaldur Gylfason, et al. 2023. "Sequence Variants Affecting the Genome-Wide Rate of Germline Microsatellite Mutations." *Nature*

- Communications* 14 (1): 3855.
- Kronenberg, Zev, Cillian Nolan, David Porubsky, Tom Mokveld, William J. Rowell, Sangjin Lee, Egor Dolzhenko, et al. 2024. "The Platinum Pedigree: A Long-Read Benchmark for Genetic Variants." *bioRxiv*. <https://doi.org/10.1101/2024.10.02.616333>.
- Mitra, Ileena, Bonnie Huang, Nima Mousavi, Nichole Ma, Michael Lamkin, Richard Yanicky, Sharona Shleizer-Burko, Kirk E. Lohmueller, and Melissa Gymrek. 2021. "Patterns of de Novo Tandem Repeat Mutations and Their Role in Autism." *Nature* 589 (7841): 246–50.
- Noyes, Michelle D., William T. Harvey, David Porubsky, Arvis Sulovari, Ruiyang Li, Nicholas R. Rose, Peter A. Audano, et al. 2022. "Familial Long-Read Sequencing Increases Yield of de Novo Mutations." *American Journal of Human Genetics* 109 (4): 631–46.
- Porubsky, David, Shilpa Garg, Ashley D. Sanders, Jan O. Korb, Victor Guryev, Peter M. Lansdorp, and Tobias Marschall. 2017. "Dense and Accurate Whole-Chromosome Haplotyping of Individual Genomes." *Nature Communications* 8 (1): 1293.
- Porubsky, David, Xavi Guitart, Dongahn Yoo, Philip C. Dishuck, William T. Harvey, and Evan E. Eichler. 2024. "SVbyEye: A Visual Tool to Characterize Structural Variation among Whole-Genome Assemblies." *bioRxiv*. <https://doi.org/10.1101/2024.09.11.612418>.
- Rahbari, Raheleh, Arthur Wuster, Sarah J. Lindsay, Robert J. Hardwick, Ludmil B. Alexandrov, Saeed Al Turki, Anna Dominiczak, et al. 2016. "Timing, Rates and Spectra of Human Germline Mutation." *Nature Genetics* 48 (2): 126–33.
- Robinson, James T., Helga Thorvaldsdóttir, Wendy Winckler, Mitchell Guttman, Eric S. Lander, Gad Getz, and Jill P. Mesirov. 2011. "Integrative Genomics Viewer." *Nature Biotechnology* 29 (1): 24–26.
- Rockweiler, Nicole B., Avinash Ramu, Liina Nagirnaja, Wing H. Wong, Michiel J. Noordam, Casey W. Drubin, Ni Huang, et al. 2023. "The Origins and Functional Effects of Postzygotic Mutations throughout the Human Life Span." *Science (New York, N.Y.)* 380 (6641): eabn7113.
- Sanders, Ashley D., Mark Hills, David Porubský, Victor Guryev, Ester Falconer, and Peter M. Lansdorp. 2016. "Characterizing Polymorphic Inversions in Human Genomes by Single-Cell Sequencing." *Genome Research* 26 (11): 1575–87.
- Sasani, Thomas A., Brent S. Pedersen, Ziyue Gao, Lisa Baird, Molly Przeworski, Lynn B. Jorde, and Aaron R. Quinlan. 2019. "Large, Three-Generation Human Families Reveal Post-Zygotic Mosaicism and Variability in Germline Mutation Accumulation." *eLife* 8 (September). <https://doi.org/10.7554/eLife.46922>.
- Smit, A. F. A., R. Hubley, and P. Green. 2013. "RepeatMasker Open-4.0 [Internet]." 2013. <http://www.repeatmasker.org>.
- Steely, Cody J., W. Scott Watkins, Lisa Baird, and Lynn B. Jorde. 2022. "The Mutational Dynamics of Short Tandem Repeats in Large, Multigenerational Families." *Genome Biology* 23 (1): 253.
- Sudmant, Peter H., Tobias Rausch, Eugene J. Gardner, Robert E. Handsaker, Alexej Abyzov, John Huddleston, Yan Zhang, et al. 2015. "An Integrated Map of Structural Variation in 2,504 Human Genomes." *Nature* 526 (7571): 75–81.
- Vollger, Mitchell R., Philip C. Dishuck, William T. Harvey, William S. DeWitt, Xavi Guitart, Michael E. Goldberg, Allison N. Rozanski, et al. 2023. "Increased Mutation and Gene Conversion within Human Segmental Duplications." *Nature* 617 (7960): 325–34.
- Williams, Amy L., Giulio Genovese, Thomas Dyer, Nicolas Altemose, Katherine Truax, Goo Jun, Nick Patterson, et al. 2015. "Non-Crossover Gene Conversions Show Strong GC Bias and Unexpected Clustering in Humans." *eLife* 4 (March). <https://doi.org/10.7554/eLife.04637>.

RESPONSES TO REFEREES:

REFEREE #1:

The authors have done an impressive job responding to the comments. I appreciate the new detailed analysis of VNTR vs. STR mutation rates. The revised Figure 3a makes the point nicely that a small number of VNTRs with large units have very high mutation rates, and the observed “trough” in mutation rates is a really cool finding. The cell line artifact is also a nice addition and highlights the benefits of using the blood-derived data. Some minor comments on the cell line part:

1. It wasn't clear how the mutations from Conrad et al. were classified as cell line artifact vs. false positives. Did you use data from the current study to do that classification?

The classification was performed previously by Conrad et al. (2011) and, in this study, we compared our callset to those classified as artefacts or false positives. In order to distinguish true germline DNMs from somatic or cell line DNMs, Conrad et al. used two orthogonal validation approaches. The first one was a nested PCR amplification of putative DNMs followed by Illumina sequencing of pooled PCR products. The second experiment was based on hybridization capture of putative DNMs using Agilent SureSelect technology followed by SOLID sequencing (Conrad et al. 2011).

We added a note to the supplement to make this clearer:

“Note: In order to distinguish true germline DNMs from somatic or cell line DNMs, Conrad et al. (2011) used two orthogonal validation approaches. The first one was a nested PCR amplification of putative DNMs followed by Illumina sequencing of pooled PCR products. The second experiment was based on hybridization capture of putative DNMs using Agilent SureSelect technology followed by SOLID sequencing (Conrad et al. 2011).”

2. “Note, because we are using DNA derived from primary material (peripheral blood), the number of SNVs arising from cell line artifacts is dramatically reduced” -> aren't cell line artifacts eliminated altogether, not just reduced? Of course there are other sources of error but I wouldn't classify them as cell line artifacts.

We agree with the reviewer that SNVs resulting from cell line artifacts should be eliminated as we are using HiFi data generated from blood DNA. There are still, of course, somatic mutations but those are not “artifacts” of cell culture but mutations that arise during cell division.

We revised the sentence as follows:

“Note, because we are using DNA derived from primary material (peripheral blood), unlike previous studies, we are eliminating *de novo* variants arising from cell line artifacts leaving only those of germline or somatic origin.”

Below are some minor comments on statistics and figure legends:

1. L245-246: indicate if p-values are one or two-sided. I assume two-sided

Yes, indeed the original p-values were calculated in a two-sided test. However, based on Reviewer #2 comments, we recalculated the significance levels, where appropriate, using Poisson generalized linear model and reported the p-values accordingly. Significance remains.

2. Extended data figure 2d, Figure 2c, Supp Figure 31: I could not find how the gray shaded regions, which I assume represent confidence intervals, are defined.

Gray shaded bars in Extended Data Fig. 2d and Supplementary Fig. 31 were calculated using the R function `stat_poly_line` (R package `ggpmisc`) and represent 95% confidence intervals of the regression line fitted using linear model function.

We added extra information on this to the relevant figure legends:

“Gray areas represent 95% confidence intervals of the regression line fitted using linear model function.”

3. Figure 2b, Extended data figure 2c, supp figure 22, supp figure 33c: boxplot elements are not defined

In Figure 2b, Supplementary Fig. 22, and Supplementary Fig. 33c, boxplot elements define a median line in the middle and the lower and upper hinges correspond to the first and third quartiles (the 25th and 75th percentiles), respectively. The whiskers extend from the hinge to the largest or smallest value but no further than $1.5 \times \text{IQR}$ from the hinge. Lastly, the histogram in the Extended Data Fig. 2c was constructed using a bin size 50.

We added the following information to the legends of the relevant figure panels:

“Boxes represent interquartile range (IQR), including median line; whiskers extend to $25\% - 1.5 \times \text{IQR}$ and $75\% + 1.5 \times \text{IQR}$; outliers are shown as dots.”

4. Figure 2d, supp fig 33c: I could not find the significance levels denoted by different numbers of *'s.

We added this extra information to the figure legends:

A significant difference from autosomal DNM or PZM rate was determined by a two-sided t-test, one asterisk (*) indicates $p < 0.05$, two asterisks (**) $p < 0.001$.

5. For some of the tests comparing maternal vs. paternal statistics a t-test is used (comparing maternal vs. paternal recombination events) whereas elsewhere a Wilcoxon signed rank test is

used (number of DNMs on maternal vs. paternal haplotypes). Wilcoxon signed rank test seems more appropriate since it compares matched samples and has fewer assumptions.

This is a good point. For consistency we recalculated the p-value reported for comparison of the number of maternal and paternal recombination breakpoints using Wilcoxon signed-rank test and updated the main text and the relevant figure panel in Supplementary Fig. 29 (see below).

Main text:

“In line with previous research, we observe a significant excess (Wilcoxon signed-rank test, $p=6.4\times 10^{-5}$) of maternal recombination events with expected 1.4 maternal to paternal breakpoint ratio²⁹ (Supplementary Fig. 29).”

Figure panel legend:

“a) Distribution of total maternal and paternal recombination breakpoints for each G2-G4 sample ($n=22$). The p-value (Wilcoxon signed-rank test) is shown on top. Boxes represent IQR, including median line; whiskers extend to $25\% - 1.5 \times \text{IQR}$ and $75\% + 1.5 \times \text{IQR}$.”

6. There are many places where rates are given for mutations/bp/generation with a 95% CI. In the legend of Figure 3a it is described how the CI is constructed (Poisson confidence intervals (computed using a chi-square distribution)). Is this how the other ones were computed? This info should be stated in the text, perhaps just the first time it is used if it is the same for all of the subsequent CIs.

We were previously using different methods, but to standardize our results, we recalculated all confidence intervals using the same method (Poisson confidence intervals computed using a chi-square distribution).

Accordingly, we updated confidence intervals in the text, figures, and Supplementary Table 10. The following text has changed slightly:

L483-492

We identify 16 SNV DNMs in centromeres, including five within the α -satellite HOR arrays, revealing a DNM rate of 1.01×10^{-7} mutations/bp/generation (95% CI: $5.75\times 10^{-8} - 1.63\times 10^{-7}$).

This rate is comparable to the rate from our read-based mapping approach, which identified 14 centromeric SNVs, albeit over more than 10 times the amount of sequence, resulting in a DNM rate of 3.27×10^{-8} mutations/bp/generation (95% CI: $1.79 \times 10^{-8} - 5.51 \times 10^{-8}$) (Fig. 2d, Supplementary Table 10). Combined, we estimate a significantly higher SNV DNM rate for centromeres of 4.94×10^{-8} (two-sided t-test, $p=0.017$). While discovery of these DNMs still remains challenging, we believe this a conservative estimate because we required validation of all events by both ONT and HiFi sequencing platforms.

L623-634

SD regions show an 88% increase (2.2 [95% CI: $1.64 - 2.88 \times 10^{-8}$] vs. 1.17×10^{-7}), which is driven by SDs with >95% identity.

Confidence intervals have also been changed in the Results section.

7. Supp Figure 1: define the red dots

Here, the red dots define the median value. We added this information into the figure legend.

8. Supp Figure 42: the legend doesn't indicate what red vs. black mean in the bar plots

Here, we use red bars to highlight values reported for germline and putative cell line artifacts from other mutation classes reported by Conrad et al. (2011).

We added the following description in the figure legend:

Red bars highlight values reported for germline DNMs and somatic/cell line DNMs in comparison to other mutation classes reported by Conrad et al. (2011).

REFEREE #2:

The authors have addressed my concerns about the unexpected decrease in number of crossovers with advancing parental age, although the presentation and statistical rigor could be improved. There is a clear parent pair effect in this set which is probably of technical nature. This strongly suggest the p-values reported in the main text (L245-246) and in Extended Data Figure 2d are anti-conservative and a mixed effect model would be more appropriate with random effect for the parent pair.

As suggested by the referee, we tested different regression models, including the one suggested by the reviewer. A mixed-effect model seems appropriate here given the visual variance among the parental pairs, particularly singling out the G4 families. As such, we tested a variety of model architectures beyond a basic fixed-effects multiple regression that fits the number of recombination breakpoints to parental age and sex. However, both linear and Poisson generalized linear models (GLMs), including family or generation as a random effect (either intercept alone or both intercept and slope), do not significantly improve fit beyond a simpler fixed-effects model agnostic of family (either using AIC or ANOVA to assess fit of nested models). The AIC values are 292 and 296 for linear GLMs modeling family intercept alone and intercept and slope as random effects, respectively; AIC=286 for a Poisson GLM modeling the family intercept as a random effect, while a Poisson GLM modeling family intercept and slope as random effects did not converge. Arguably, this observation may be a consequence of low sample size or small number of parental pairs, as mixed-effects models can lack power or struggle to converge when either parameter is small.

Under the best fit model, we find a significant decrease in recombination breakpoints as a function of parental age and sex ($p=7.17 \times 10^{-3}$ and 1.22×10^{-9} for parental age and sex, respectively; Poisson GLM with a log link, AIC = 284.2). Allowing the parental age slope to vary between families with an interaction term yielded no significant family-specific effects (AIC=290.3, 289.4; $p>0.05$ for Poisson GLMs modeling interactions among parental age, homolog, and family; G4 families modeled separately and together, respectively).

We updated Extended Data Fig. 2d and Supplementary Fig. 31 with the best fit model (i.e., Poisson generalized linear model (GLM) with link log) and revised the main text with these new results but also caution that larger sample sizes of many more families will be required to conclusively determine if there are family-specific effects.

Extended Data Figure 2. Recombination breakpoint map of CEPH 1463.

d) Significant association between the number of recombination breaks (y-axis) and parental age (x-axis) shown separately for maternal (red) and paternal (blue) recombination breakpoints (G2-G3) detected with respect to T2T-CHM13. Regression lines were fitted using Poisson GLM with a log link ($p=2.02 \times 10^{-3}$, 7.88×10^{-4} for parental age and sex effects, respectively).

Supplementary Figure 31: Recombination breakpoints and parental age.

Significant association between the number of recombination breaks (y-axis) and parental age (x-axis) shown separately for maternal (red) and paternal (blue) recombination breakpoints detected with respect to T2T-CHM13 for all G2-G4 individuals. Regression lines were fitted using Poisson generalized linear model (GLM) with a log link ($p=7.17 \times 10^{-3}$ and 1.22×10^{-9} for parental age and sex, respectively).

Revised text:

“In G2-G3 we observed a significant decrease in crossover events with advancing parental age for both male and female germlines (Extended Data Fig. 2d). We modeled this observation using the same architecture (a Poisson generalized linear model (GLM) with link log), including G4 subfamilies, and continue to observe the same significant decrease in recombination breakpoints as a function of parental age and sex ($p=7.17 \times 10^{-3}$ and 1.22×10^{-9} for parental age and sex, respectively; Poisson GLM with a log link, AIC = 284.2) (Supplementary Fig. 31). We consider this observation preliminary due to low sample size.”

The authors have improved the analysis of the false positive rate across the different DNM types and genomic regions. More specifically by estimating and reporting the transmission rate for the different genomic regions and types of variants, it has enabled the reader to gauge the accuracy of their DNM calls. This work by the authors has clearly resulted in better estimates of

the mutation rates and improved presentation of their results. However, the quantification and presentation of the post-zygotic mutation contribution should be improved. The authors quote that 17% of DNMs are post-zygotic per transmission in origin in the abstract, there are numerous issues with this statement. To my understanding, this statement is based on a simple aggregate across the G2 and G3 generations, tallying the number of mutations that are fully linked with (germline/pre-zygotic) or not (post-zygotic) with one of the parental haplotypes. However, most of the individuals (6/8) in the G3 generation do not have sequenced children, therefore the authors have generally no information whether these PZM variants in G3 will be transmitted to the next generation, i.e. G4. Further, a PZM transmitted to multiple offspring should be weighted differently than one transmitted to a single offspring. This is especially important in the case of a PZM with no evidence of transmission which could just be confined to a somatic cell type and hence irrelevant for estimation of germline mutation rates.

Based on Supplementary Figure 45 there seems to be transmission of 24 PZM variants from G2 to G3 resulting in 61 transmitted PZM-DNMs to the G3 offspring or on average 7.6 PZM mutations are transmitted from G2 to G3. In contrast the G3 have received 506 germline mutations from G2 (based on Supplementary Table 10) resulting in 63 DNMs on average, then the fraction of PZM in G3 received from G2 is roughly 11% ($7.6/(63+7.6)$) based on these data. This roughly halves the statistic reported by the authors and is more in line with other reported estimates. However, it is hard to generalize this estimate from the authors as this is based on developmental mutations identified in two individuals and there is considerable variability in the contribution of these early developmental mutations to the different cell types (PMID:34433963; 33414551; 38605218).

Further there are clear QC issues with the PZMs (Supplementary Figure 45), false positive calls seem to be more frequent in the G3 generation and are present in nearly all of the individuals in high frequency (~25%). This indicates that the false positive variants are biased towards more frequent variants, which hinders accurate quantification of PZMs contribution to the germline for individuals that do not have children.

In other words, this indicates that it does not suffice to restrict to mutations that are partially linked to long reads to claim that they will be accurately transmitted to the next generation based on their AB. These artifacts would probably have been evident by reporting the grand parental haplotypes of the G3 individuals at the PZM positions in Supplementary Table 10.

The reviewer raises some interesting points. However, we believe there is a misunderstanding. First, we never include PZMs in our reported estimates of the germline mutation rate—the germline rate is always calculated exclusively using DNMs that we determined to arise in parental gametes; 98% of those that could be tested transmit faithfully to the next generation. Second, transmission is not a requirement to validate a PZM, but in our case it is a good proxy for confirming a variant's presence in multiple tissues. As we previously stated, it is true that we cannot confidently assert that any PZM that is not transmitted to the next generation is actually present in multiple tissues and not confined to the blood. However, even if every PZM is present in the germline, we expect 20-30% to not be transmitted to the next generation due to

characteristically lower allele balance, and therefore incomplete distribution across progenitor germ cells.

Nevertheless, as suggested by the reviewer, to further validate our PZMs, we implemented another strategy that we could apply to all variants, even those in samples without sequenced children. We defined the haplotype on which a PZM arose using tagging SNPs (tSNPs), and then refined that set of tSNPs until we had a list of alleles that uniquely identify the haplotype in a sample's parent and, if applicable, grandparent. We examined every HiFi read, regardless of quality, from that inherited haplotype across the generations, checking for the presence of the alternate allele (Supplementary Fig. 45a). In total, we found six variants that may have been inherited from the previous generation and therefore not true PZMs.

We removed those variants from our callset and then applied our second validation strategy based on transmission (Supplementary Fig. 45b). In this case, we removed any variants that, upon transmission to the next generation, still displayed incomplete linkage to a haplotype. This step removed four more variants, for a total of 10 variants now excluded from our postzygotic callset resulting in a false positive rate of 8.4%. Note, we no longer report these or include them in any further analyses. In summary, we now report 119 PZMs, compared to our previous estimate of 129 PZMs. Accordingly, we readjusted all figures, rates, and analyses in the text. Thus, we find that PZMs make up 16% of all *de novo* SNV calls ($n=119$ PZMs/745 *de novo* SNVs). In the four samples with sequenced children, we find that PZMs account for 12% of all SNVs transmitted to the next generation ($n=33$ PZMs/275 transmitted SNVs).

We added the results of this validation process as part of Supplementary Fig. 45, which we believe may have been previously confusing. Panel a shows the “backward validation process,” displaying all HiFi reads in the parent and grandparent from which that haplotype was inherited. If that haplotype could not be uniquely defined by tSNPs, there is no data for previous generations. Unphased reads are shown in gray, reads assigned to the PZM haplotype are in blue, and reads phased to the other haplotype are in red. Any reads with the PZM allele are shown in yellow. The six PZMs that failed this validation test are highlighted in pink—you can see that in most cases, a parent has a read with the alternate allele.

Panel b displays the “forward validation process,” showing high-quality HiFi reads in each offspring who inherited a PZM allele. Transmissions are clustered by PZM, and highlighted in orange are the four PZMs that failed this validation—you can see that the PZM allele is incompletely linked to the inherited haplotype across multiple transmissions. Note that for simplicity, only HiFi data are shown in these plots, but ONT and Illumina data are also considered when assessing inheritance and transmission.

Supplementary Figure 45: Phased haplotypes and allele counts.

a) Phased HiFi read counts for a *de novo* sample (bottom row), the parent from which they inherited the *de novo* haplotype (middle row), and the grandparent from which they inherited that haplotype (top row). Each column corresponds to a PZM, and missing read data indicates that a haplotype could not be uniquely assigned to a parent or grandparent. Reads assigned to the *de novo* haplotype are shown above the x-axis in blue, reads from the other haplotype are below the x-axis in red, and unphased reads are below the x-axis in gray. Reads with the alternate allele are shown in yellow. Variants in boxes are false positives—a pink highlight indicates that the variant failed backward validation by examining ancestors, and an orange highlight indicates that it failed forward validation by transmission. **b)** PZM transmissions to the next generation are shown. Transmissions are grouped by mutation, and one bar is shown for every transmission event. Lettered events correspond to the false positives shown in part (a).

Finally, as a result, we changed the abstract and the discussion to match these new findings:

Abstract:

“From this family, we estimate a range of 98-206 DNMs per transmission, including 74.5 *de novo* single-nucleotide variants (SNVs), 7.4 non-tandem repeat indels, 65.3 *de novo* indels or

structural variants (SVs) originating from tandem repeats, and 4.4 centromeric *de novo* SVs and SNVs. Among males, we find 12.4 *de novo* Y chromosome events per generation. Short tandem repeats and variable number tandem repeats are the most mutable with 32 loci exhibiting recurrent mutation through the generations. We accurately assemble 288 centromeres and six Y chromosomes across the generations, documenting *de novo* SVs, and demonstrate that the DNM rate varies by an order of magnitude depending on repeat content, length, and sequence identity. We show a strong paternal bias (75-81%) for all forms of germline DNM, yet we estimate that 16% of *de novo* SNVs are postzygotic in origin with no paternal bias.”

Discussion:

“Previous studies predicted that 6-10% of DNMs are not germline in origin, but instead arise sometime after fertilization, giving rise to a mosaic variant^{14,55}. These studies distinguished between *de novo* postzygotic and germline SNVs using allele balance thresholds⁵⁵ or by identifying incomplete linkage to nearby SNVs across three generations¹⁴, but long-read data can substantially increase the sensitivity of these approaches. We assign nearly every *de novo* SNV to a parental haplotype, defining a PZM by its incomplete linkage to that haplotype. We classify 16% of *de novo* SNVs as postzygotic in origin (n=119 PZMs/745 *de novo* SNVs). Because all sequencing data in this study are derived from blood, we cannot demonstrate that every PZM is present in multiple tissues, but we can use transmission to the next generation as a proxy, as it reveals the mutation is also present in germ cells. In the four samples with sequenced children, we find that PZMs account for 12% of all SNVs transmitted to the next generation (n=33 PZMs/275 transmitted SNVs), an increase over previous estimates. Early cell divisions of human embryos are frequently error prone^{56,57} with an accelerated rate of cell division and these properties may contribute to the large fraction of PZMs with high (>25%) allele balance (AB; 49% [n=58/119] are estimated to have high AB and 86% of high-AB PZMs in individuals with sequenced children [n=24/28] are transmitted to the next generation). Such events would previously have been classified as germline but, consistent with PZM expectations, we find no paternal bias associated with these *de novo* variants (**Fig. 2b**).”

Specific Comments:

- L321-328: This PZM rate should be revisited in the context of transmitted mutations.
- L327: Further given the high frequency false positives in G3 the enrichment in SDs should be revisited how many of the FPs (3)

We are choosing to report the mutation rate for all PZMs, not just transmitted ones. This is similar for previous studies, which did not typically have a third generation for testing transmission. Every PZM is not expected to be in germ cells, and the PZMs that are in germ cells have low frequency and may not make it into the next generation by random chance, so restricting our callset to transmitted calls would be a meaningful underestimation of the actual mutation rate. Nevertheless, we try to make this distinction clearer below.

We report a PZM rate of 2.04×10^{-9} SNVs/bp/generation (95% CI: $1.68 - 2.47 \times 10^{-9}$) across the autosomes, with a 3.9-fold enrichment of PZMs in SDs (95% CI: $4.84 \times 10^{-9} - 1.25 \times 10^{-8}$ SNVs/bp/generation, two-sided t-test, p=0.049). If we restrict our callset to only PZMs that could

be validated by transmission to the next generation, (n=33 PZMs across 4 samples), we calculate a PZM rate of 1.54×10^{-9} SNVs/bp/generation (95% CI: 1.07×10^{-9} – 2.17×10^{-9}) with a 2.69-fold enrichment of in SDs (95% CI: 1.15×10^{-9} – 1.08×10^{-8} SNVs/bp/generation) that does not reach significance due to the small sample size (two-sided t-test, $p=0.218$).

We rewrote the Results section on small variants to better reflect the nature of PZMs and our validation approach. We also make it clear what the numbers are for both transmitted and untransmitted variants (see text in bold below).

Revised text:

De novo SNVs and small indels. To discover small variants, we examined HiFi reads aligned to T2T-CHM13 for variant discovery, then leveraged orthogonal ONT and Illumina data to confirm that a variant is in fact present in a sample and absent from both its parents (**Methods**). This strategy reduces bias introduced by any one sequencing platform, but it restricts DNM discovery to G2 and G3 individuals, as we did not have corresponding G4 cell lines for ONT data generation. Our *de novo* callset included 755 SNVs and 73 indels across the autosomes of 10 individuals (n=2 G2; n=8 G3 individuals, **Fig. 2a**), as well as 27 SNVs and 1 indel on Chromosome X.

To further characterize autosomal DNMs, we used flanking SNVs from long-read data to construct haplotypes, phase variants, and trace a mutation back either to a parental gamete or the early embryo. We determined that a mutation occurred somatically, and likely early in embryonic development, if it met one of two criteria: it was incompletely linked to a parental haplotype (n=122), or, if it could not be phased, it had an allele balance significantly less than 0.5 across all three sequencing platforms (n=7). We further validated each postzygotic mutation (PZM) by tracing its haplotype backward across generations and forward for the four individuals with sequenced offspring (**Supplementary Note 4**). **Of the 62 PZMs in these four samples, 64.5% (n=40) are transmitted to the next generation, compared to 97.2% of germline SNVs (n=242/249) and 100% of indels (Extended Data Fig. 3). We found that 10 PZMs failed these haplotype-based validations, resulting in a final callset of 119 PZMs, accounting for 16% of total autosomal SNVs (745 *de novo* SNVs).** Previous Illumina-based analysis of this family¹⁴ identified 605 *de novo* SNVs of either germline (G2 and G3) or postzygotic (only G2) origin, 92.4% (n=559) of which were represented in our final callset, while all but four of the absent variants failed validation with long-read data. Not only were we able to identify an additional 72 PZMs in G3 for the first time, but we also identified a total of 186 novel DNMs, a 6.1% and 21% increase in germline SNV and indel discovery, respectively.

We find that 81.4% of germline small DNMs originate on paternal haplotypes (4.38:1 paternal:maternal ratio, Wilcoxon signed-rank test, $p < 2 \times 10^{-16}$), with a significant parental age effect of 1.55 germline DNMs per additional year of paternal age when fitting with linear regression (two-sided t-test, $p=0.013$). In contrast, PZMs show no significant difference with respect to parental origin (1.38:1 paternal:maternal ratio, Wilcoxon signed-rank test, $p=0.09$) and no parental age effects (Fig. 2c). While our small sample size does not provide sufficient power to detect significant differences between the *de novo* and postzygotic mutational spectra (Supplementary Fig. 33a), we do observe a novel depletion of CpG>TpG PZMs (chi-squared

test, $p=0.17$) and an enrichment of postzygotic T>A substitutions (chi-squared test, $p=0.268$) that has been previously observed¹⁴.

Using this approach, we successfully assay 91.9% of the autosomal genome (2.66 Gbp) (**Supplementary Fig. 33b, Supplementary Note 4**). Excluding all variants classified as postzygotic, we find that the parental germline contributes 1.17×10^{-8} SNVs/bp/generation (95% CI: $1.08 - 1.27 \times 10^{-8}$). *De novo* SNVs are significantly enriched in repetitive sequences, as much as 2.8-fold in centromeres (95% CI: $1.79 - 5.51 \times 10^{-8}$ SNVs/bp/generation, two-sided t-test, $p=0.017$) and 1.9-fold in SDs (95% CI: $1.64 - 2.88 \times 10^{-8}$ SNVs/bp/generation, two-sided t-test, $p=0.0066$) (**Fig. 2d, Supplementary Fig. 33c, Supplementary Table 10**). We observe a lower PZM rate of 2.04×10^{-9} SNVs/bp/generation (95% CI: $1.68 - 2.47 \times 10^{-9}$) across the autosomes, yet we see 3.9-fold enrichment of PZMs in SDs (95% CI: $4.84 \times 10^{-9} - 1.25 \times 10^{-8}$ SNVs/bp/generation, two-sided t-test, $p=0.049$). Among PZMs transmitted to the next generation ($n=33$ PZMs across 4 samples), we observe a 2.69-fold enrichment in SDs (95% CI: $1.15 \times 10^{-9} - 1.08 \times 10^{-8}$ SNVs/bp/generation) that does not reach significance due to the small sample size (two-sided t-test, $p=0.218$).

- L635-648: This should be revisited and here the authors should acknowledge the limitation of their setup to assess the quality of the PZM calls especially in the SD regions as they can only assess transmission of 40 PZMs.

We added clarification on the role of transmission in validating our PZMs and now report the fraction of PZM calls in both the whole callset (16%, $n=119$ PZMs/745), and in the smaller callset of transmitted variants (12%, $n=33/275$).

Based on our new analysis, the referee is correct that there is a higher false positive rate in the SDs compared to unique regions. Through our validation process, we eliminated five false positive PZM calls that were present in SDs (false positive rate of 20.8%). Our final callset contains 19 PZMs in SDs, including four that were validated by transmission to the next generation. Among all PZMs, we observe a 3.9-fold enrichment of PZMs in SDs (95% CI: $4.84 \times 10^{-9} - 1.25 \times 10^{-8}$ SNVs/bp/generation, two-sided t-test, $p=0.049$) and for PZMs transmitted to the next generation, ($n=33$ PZMs across 4 samples), we observe a 2.69-fold enrichment in SDs (95% CI: $1.15 \times 10^{-9} - 1.08 \times 10^{-8}$ SNVs/bp/generation) that does not reach significance due to the small sample size (two-sided t-test, $p=0.218$).

The PZM rate in SDs was initially reported in the Results section (L331-334), not in L635-648. Consequently, we removed false positive calls from the rates reported in L331-334 and included the PZM enrichment in SDs for only PZMs validated by transmission. We added the following sentence, as outlined above:

“Among PZMs transmitted to the next generation ($n=33$ PZMs across 4 samples), we observe a 2.69-fold enrichment in SDs (95% CI: $1.15 \times 10^{-9} - 1.08 \times 10^{-8}$ SNVs/bp/generation) that does not reach significance due to the small sample size (two-sided t-test, $p=0.218$).“

We also excluded PZMs from our comparison between assembly- and alignment-based SNV calls in the centromeres, see revised text above (L483-492).

Finally, we modified the Discussion section (see below) to better reflect the abilities and limitations of our PZM callset. Note that we do not refer to PZMs in L635-641, and these modifications apply to L643-650.

Revised text:

“Previous studies predicted that 6-10% of DNMs are not germline in origin, but instead arise sometime after fertilization, giving rise to a mosaic variant^{14,55}. These studies distinguished between *de novo* postzygotic and germline SNVs using allele balance thresholds⁵⁵ or by identifying incomplete linkage to nearby SNVs across three generations¹⁴, but long-read data can substantially increase the sensitivity of these approaches. We assign nearly every *de novo* SNV to a parental haplotype, defining a PZM by its incomplete linkage to that haplotype. We classify 16% of *de novo* SNVs as postzygotic in origin (n=119 PZMs/745 *de novo* SNVs). Because all sequencing data in this study are derived from blood, we cannot demonstrate that every PZM is present in multiple tissues, but we can use transmission to the next generation as a proxy, as it reveals the mutation is also present in germ cells. In the four samples with sequenced children, we find that PZMs account for 12% of all SNVs transmitted to the next generation (n=33 PZMs/275 transmitted SNVs), an increase over previous estimates. Early cell divisions of human embryos are frequently error prone^{56,57} with an accelerated rate of cell division and these properties may contribute to the large fraction of PZMs with high (>25%) allele balance (AB; 49% [n=58/119] are estimated to have high AB and 86% of high-AB PZMs in individuals with sequenced children [n=24/28] are transmitted to the next generation). Such events would previously have been classified as germline but, consistent with PZM expectations, we find no paternal bias associated with these *de novo* variants (Fig. 2b).”

- Supplementary Table 10: It would be useful to get the AB of the parents from all sequencing technologies for the SNVs/Indels in Supplementary Table 10. Further, the grand parental origin of the children’s haplotypes for the carriers and non-carriers of the DNM allele would help in the QC of the PZM transmissions.

We updated Supplementary Table 10 to include the AB of each *de novo* SNV for a sample and both parents. To calculate AB, we consider only “high-quality” reads, meaning that the base quality is >20 at the site of the mutation, and for HiFi and ONT reads, the mapping quality is >59. If a sample has “NA” for allele balance, it means there were no high-quality reads at the mutation site. A snapshot of the full table with headers is shown below.

Table containing a list of all alignment detected de novo SNVs classified as germline and postzygotic. Allele balance (AB) is given for each sample and their parents (father/mother) based on high-quality reads (base quality > 20, map quality > 59 for HiFi and ONT).													
sample	chromosome	T2T-CHM13v2.0 position	reference allele	alternate allele	variant origin	inheritance	children with allele	HiFi AB	ONT AB	Illumina AB	Parental HiFi AB	Parental ONT AB	Parental Illumina AB
NA12877	chr1	40670042	G	A	germline	maternal	NA12887;NA12884	0.5	0.54023	0.51351	0.00.0	0.00.0	0.00.0
NA12877	chr1	77958305	C	G	germline	paternal	NA12886;NA12881;NA12879;NA12885;NA12883	0.38296	0.49495	0.33333	0.00.0	0.00.0	0.00.0
NA12877	chr1	77958326	G	A	germline	paternal	NA12886;NA12881;NA12879;NA12885;NA12883	0.4	0.53191	0.37037	0.00.0	0.00.0	0.00.0
NA12877	chr1	79030358	A	C	postzygotic	maternal	NA12882	0.32743	0.5042	0.14286	0.00.0	0.00.0	0.00.0
NA12877	chr1	98389568	A	C	germline	paternal	NA12879;NA12885;NA12883	0.48872	0.44	0.74286	0.00.0	0.00.0	0.00.0

We also assessed (see Supplementary Figure 45) grandparental status of the PZM as discussed above.

REFEREE #3:

I thank the authors for answering my requests in detail. I have no other critical points. I have one more suggestion that I think may be helpful for the readers. The new supplementary figure 13 now shows an upset plot with all SVs called by different methods. It could be helpful to see a similar plot, but only for the curated SV variant set. This way a reader can see which tools were better at detecting the true variants. I leave this to the discretion of the authors and do not need to re-review the manuscript.

Supplemental Figure 13 was already showing the curated variant SV set. We revised the figure legend for clarity:

Supplementary Figure 13. The agreement among different SV callers in the curated truth set.

The UpSet plot depicts the intersection of pedigree consistent structural variant (SV) callsets, excluding tandem repeats. The bars on the right indicate the size of each SV callset, while the central bar chart displays the counts of overlapping sets. On average, each SV call is supported by 2.5 (average) of the four methods, and 61% of SV calls are supported by more than one method. The agreement between methods varied by type: deletions had the highest overlap supported by 2.9 (average) callers, insertions supported by 2.3 (average) callers, and inversions supported by 1.1 (average) callers.

February 19, 2025

Dear Gf kqt,

We have revised the manuscript in response to the last remaining comments from the referee. As suggested, by Referee #2, we qualified our observations by adding a note in the abstract that the 16% include germline mosaic mutations and thus can be transmitted to next generation. We maintain that germline mosaic mutations do count in our analysis and that is why we stressed the term postzygotic mutation but now qualify this more clearly in the text. Per the referee's request, we have also made a point in the results that there is no biological mechanism to expect recombination to decrease in both parental germlines and that this observation runs counter to previous studies based on lower resolution population-based surveys. Again, our observation is significant, but we caution this is based only on one family so more families will need to be analyzed. Finally, with respect to the technical point raised by the referee: We opted to use a log link function for the Poisson regression because we observed a negative association between parental age and number of recombination breakpoints. A linear link function would result in a negative count of recombination breakpoints past a certain parental age, which would be largely uninterpretable.

In addition, in response to editorial comments we have formatted the manuscript to match *Nature* style and reduced the main text to 4,665 words (+217-word abstract); additional point-by-point responses below. With respect to data access, 23 samples are now available as part of AWS Open Data program (we provide detailed instructions how to access these as part of their S3 bucket through a GitHub: <https://github.com/Platinum-Pedigree-Consortium/Platinum-Pedigree-Datasets>). The remaining five samples were approved for restricted access via dbGaP under Accession ID: phs003793.v1.p1. We were granted approval (February 12) by NCBI/NIH (communications have been painstakingly slow especially this last month) to start uploading data for the remaining samples to dbGaP. We will do so in coming days/weeks to make sure all data are publicly available ahead of publication. We have adjusted the data access section of the manuscript to reflect these updates.

I wish to participate in transparent peer review. We thank you for a thorough and prompt review process and look forward to publication in *Nature*.